



# Sediment fluxes dominate glacial-interglacial changes in ocean carbon inventory: results from factorial simulations over the past 780,000 years

Markus Adloff[1,2], Aurich Jeltsch-Thömmes[1,2], Frerk Pöppelmeier[1,2], Thomas F. Stocker[1,2], and Fortunat Joos[1,2]

[1]Climate and Environmental Physics, Physics Institute, University of Bern, Switzerland
[2]Oeschger Centre for Climate Change Research, University of Bern, Switzerland

**Correspondence:** Markus Adloff (markus.adloff@unibe.ch)

**Abstract.** Atmospheric $CO_2$ concentrations changed over ice age cycles due to net exchange fluxes between land, ocean, ocean sediments, atmosphere, and the lithosphere. Marine sediment and ice cores preserved biogeochemical evidence of these carbon transfers, which resulted from sensitivities of the various carbon reservoirs to climate forcing, many of which remain poorly understood. Numerical studies proved the potential of several physical and biogeochemical processes to impact atmospheric

$CO_2$ under steady-state glacial conditions. Yet, it is unclear how much they affected carbon cycling during transient changes of repeated glacial cycles, and what role burial and release of sedimentary organic and inorganic carbon and nutrients played. Addressing this uncertainty, we produced a simulation ensemble of various physical and biogeochemical carbon cycle forcings over the repeated glacial inceptions and terminations of the last 780 kyr with the Bern3D Earth system model of intermediate complexity including dynamic marine sediments. This ensemble allows for assessing transient carbon cycle changes due to

these different forcings and gaining a process-based understanding of the associated carbon fluxes and isotopic shifts in a continuously perturbed Earth system. We present results of the simulated Earth system dynamics in the non-equilibrium glacial cycles and a comparison with multiple proxy time series. In our simulations the ocean inventory changed by 200-1400 GtC and the atmospheric inventory by 1-150 GtC over the last deglaciation. DIC changes differ by a factor of up to 28 between simulations with and without interactive sediments, while $CO_2$ changes in the atmosphere are at most four times larger when

interactive sediments are simulated. Simulations with interactive sediments show no clear correlations between DIC or nutrient concentrations and atmospheric $CO_2$ change, highlighting the likely need for multi-proxy analyses to understand global carbon cycle changes during glacial cycles in practice. Starting transient simulations with an interglacial geologic carbon cycle balance causes isotopic drifts that require several 100 kyr to overcome, and needs to be considered when designing spin-up strategies.

# 1 Introduction

During the Quaternary, Earth's carbon cycle repeatedly shifted between low atmospheric $CO_2$ concentrations during glacial periods and elevated concentrations during interglacials in orbitally-paced cycles (Petit et al., 1999; Siegenthaler et al., 2005;



Lüthi et al., 2008). The reconstructed evolution of atmospheric $CO_2$ concentrations from Antarctic ice cores aligns closely with temperature and, with a delay, ice sheet extent, suggesting a close coupling of climate and the carbon cycle (e.g. Shackleton, 25 2000; Bereiter et al., 2015). Yet, simulating atmospheric $CO_2$ changes that are consistent with recnstructed $CO_2$ and other proxy data is challenging because the observed carbon cycle changes were the result of complex Earth system responses to climate forcing (Schmittner, 2008).

Changing ocean chemistry is often attributed an important role in these cycles because of the considerable size of the marine carbon reservoir (Broecker, 1982a) and because reconstructions imply smaller carbon stocks in vegetation and soils 30 at the Last Glacial Maximum than during the current warm period (Lindgren et al., 2018; Jeltsch-Thömmes et al., 2019). A multitude of physical and biogeochemical processes have been assessed for their contribution to changes in the marine carbon storage on these timescales (e.g. Sigman et al., 2010; Fischer et al., 2010), and their relative importance for the $CO_2$ difference between the last glacial maximum and the late Holocene have been tested in numerical simulations with dynamic ocean models (e.g. Brovkin et al., 2012; Menviel et al., 2012). Changes in ocean circulation and $CO_2$ solubility due to lower 35 temperatures contributed to the lower glacial atmospheric $CO_2$ concentration (Broecker, 1982a; Smith et al., 1999; Brovkin et al., 2007; Sigman et al., 2010; Fischer et al., 2010), while increased salinity and surface ocean dissolved inorganic carbon (DIC) concentrations due to lowered sea levels added $CO_2$ back to the glacial atmosphere (Weiss, 1974; Broecker, 1982a; Brovkin et al., 2007). Furthermore, reduced carbon outgassing from the Southern Ocean due to a greater extent of sea ice isolating the surface ocean from the atmosphere, and enhanced stratification due to brine rejection during sea ice formation are 40 other physical processes suggested to have affected the glacial carbon cycle (Stephens and Keeling, 2000; Bouttes et al., 2010).

Biogeochemical processes that lead to increased ocean carbon storage include reduced organic carbon remineralization rates in the colder ocean, as well as increased nutrient supply from emerged shelves (phosphate), enhanced dust deposition (iron, silica) and supply from the deep Southern Ocean, which would have counteracted the effect of low temperatures on surface ocean productivity and increased export production (Broecker, 1982b; Martin, 1990; Pollock, 1997; Deutsch et al., 2004). In a 45 closed atmosphere-ocean system, the combination of these processes result in increased marine carbon storage during glacials, but not necessarily in open systems (i.e. considering dynamic land and lithospheric carbon reservoirs) (e.g. Buchanan et al., 2016; Kemppinen et al., 2019).

Little is known about the changes of carbon stored as organic and inorganic carbon in a fourth reservoir: marine sediments and the lithosphere. It has since long been assumed that changing sedimentary carbonate burial played a relevant role in glacial-50 interglacial carbon cycle changes by altering seawater carbonate chemistry, particularly on continental shelves which would have emerged from the ocean during glacial sea level low stands and provided new reef habitats and carbonate deposition environments during deglaciations and interglacials (e.g. Broecker, 1982b; Opdyke and Walker, 1992; Ridgwell et al., 2003; Brovkin et al., 2007; Menviel and Joos, 2012). Additionally, carbonate burial changes in the open ocean have been considered as amplifiers of marine carbon uptake (e.g. Archer and Maier-Reimer, 1994; Kohfeld and Ridgwell, 2009; Schneider et al., 55 2013; Roth et al., 2014; Kerr et al., 2017; Kobayashi et al., 2021). Consistently, reconstructions of marine burial changes over the last glacial cycle suggest a reduction in globally-averaged inorganic carbon burial (Cartapanis et al., 2018; Wood et al., 2023) during the last glacial period, but increased organic (Cartapanis et al., 2016) sedimentary carbon burial. The extents of



both changes are uncertain due to the spatial heterogeneity of sedimentary burial and the inherently local nature of marine sediment cores, but possibly of comparable magnitude to terrestrial carbon stock changes (Cartapanis et al., 2016, 2018).

Previous model simulations, that included organic carbon burial, showed that interactive sediments greatly affect atmospheric $CO_2$ and carbon isotope amplitudes through the burial-nutrient feedback (Tschumi et al., 2011; Roth et al., 2014; Jeltsch-Thömmes et al., 2019; Jeltsch-Thömmes and Joos, 2023). Dynamic sedimentary adjustment and imbalances in weathering-burial fluxes also increase the equilibration time of atmospheric $CO_2$ by a factor of up to 20 to several tens of thousands of years and the resulting $\delta^{13}C$ perturbations take hundreds of thousands of years to recover (Roth et al., 2014; Jeltsch-Thömmes

et al., 2019; Jeltsch-Thömmes and Joos, 2023). These findings demonstrate that organic and inorganic sedimentary changes and imbalances between the weathering and burial fluxes to the consolidated sediments and lithosphere need to be considered when quantifying carbon reservoir changes of the ocean, atmosphere and land and interpreting the reconstructed changes in $CO_2$, isotopes, and nutrients over glacial cycles.

    Model-based estimates of carbon and carbon isotope inventory differences between glacial and interglacial periods are com-

plicated by temporal carbon cycle imbalances during the continuously evolving climate of glacial cycles. This is particularly challenging when simulating marine sediments and the input of elements by weathering and volcanic outgassing and the loss of elements by burial in reactive sediments and the lithosphere, because of long-lasting re-equilibration and memory effects in carbon and nutrient fluxes and particularly isotopic changes (Tschumi et al., 2011; Jeltsch-Thömmes and Joos, 2020). Importantly, sedimentary fluxes and related weathering-burial imbalances never reached true equilibrium during glacial cycles,

which implies the possibility for memory effects that span several glacial cycles.

    Transient simulations of a whole glacial cycle with a fully dynamic marine and sedimentary carbon cycle showed that time lags in the carbon cycle response to orbital forcing add constraints for the identification of the processes that caused glacial $CO_2$ changes (Menviel et al., 2012). In particular, imbalances between marine carbon burial and continental weathering and the long marine residence time of phosphate delay the $CO_2$ increase during the temperature rise of deglaciations. Transient

simulations of more than one glacial cycle showed that reconstructed atmospheric $CO_2$ and benthic marine $\delta^{13}C$ changes over the last 400 kyr could be reasonably well simulated with a combinations of physical (radiative and ocean volume changes) and biogeochemical processes (carbonate chemistry and land carbon changes, temperature-dependent remineralization depth, additional nutrient supply during glacials) if shallow water carbonate burial changed too (Ganopolski and Brovkin, 2017). These burial changes were partially prescribed, which begs the question how well the considered processes can explain glacial-

interglacial atmospheric $CO_2$ change if sedimentary burial and surface ocean carbonate chemistry are entirely dynamically simulated. Simulations of glacial-interglacial cycles beyond the Mid-Brunhes transition (∼430 ka) were run with a box model (Köhler and Munhoven, 2020) and purely physical models (Stein et al., 2020) which are unable to capture transient and spatially heterogeneous interactive sediments. CLIMBER-2, a fully coupled intermediate-complexity Earth system model, was run stepwise over the last 3 Myr, but the results were not analysed for the carbon cycle dynamics (Willeit et al., 2019).

Here we extend factorial simulations of multiple simplified physical and biochemical forcings in a marine sediment and isotope-enabled intermediate complexity Earth system model over the last 780 kyr to understand how various processes affect carbon fluxes in response to continuously varying climate, and specifically how sediments affect the marine and atmospheric



carbon cycle across repeated glacial cycles. To avoid biases resulting from steady state assumptions, we simulated the last eight glacial cycles fully transiently, so that all carbon stores at the beginning of the last glacial cycle are achieved dynamically rather

than being prescribed. Specifically, we present two sets of simulations with and without interactive sediments to distinguish the role of interactive sediments in the carbon cycle changes caused by these different forcings over repeated glacial cycles.

## 2    Methods

### 2.1    Bern3D v2.0

We simulated the Earth system's transition through the last 780 kyr of glacial cycles with the intermediate complexity Earth

system model Bern3D v2.0s, which has a resolution of 41×40 in the horizontal and 32 logarithmically spaced ocean depth layers. The model combines modules for 3D physical ocean dynamics, marine biogeochemistry, marine interactive sediments, and atmospheric energy-moisture balance. The geostrophic-frictional balance ocean circulation is calculated explicitly (Edwards et al., 1998; Müller et al., 2006), and parameterizations are included to represent the effects of dia- and isopycnal diffusion and eddy-induced transport (Griffies, 1998). The NCEP/NCAR monthly wind stress climatology (Kalnay et al., 1996) is used to

prescribe wind stress at the ocean surface. Atmosphere-ocean gas exchange and carbonate chemistry are simulated according to the OCMIP-2 protocols (Najjar et al., 1999; Orr et al., 1999, 2017; Wanninkhof, 2014; Orr and Epitalon, 2015), and gas transfer velocities are scaled with wind speed (Krakauer et al., 2006). The global mean sea-air gas exchange was then reduced by 19% to achieve agreement with radiocarbon distribution estimates (Müller et al., 2008).

    The physical ocean component transports tracers through the ocean by advection, convection, and diffusion. Euphotic zone

production depends on temperature, light, sea ice cover, and nutrient (phosphate, iron, silica) availability, with a full description of the model biogeochemistry in Parekh et al. (2008); Tschumi et al. (2011) and of carbon isotope dynamics in Jeltsch-Thömmes et al. (2019). In our setup, a fraction of the particulate organic matter formed in the surface ocean is instantly remineralized following an oxygen concentration dependent version of the globally-uniform Martin curve (Battaglia and Joos, 2018) and particulate inorganic carbon and opal dissolution occurs according to globally-uniform e-folding profiles. The re-

maining solid particles reaching the sediment-ocean interface enter reactive sediments, where they are preserved, remineralized or redissolved depending on dynamically calculated porewater chemistry, and mixed by bioturbation (Tschumi et al., 2011). The sediment model includes 10 layers and computes fluxes of carbon, nutrients, alkalinity, and isotopes between the ocean, reactive sediments, and the lithosphere. Loss fluxes to the lithosphere are compensated for at equilibrium by a corresponding input flux from weathering. The ocean model includes diagnostic tracers for preformed DIC and phosphate, which track the

fractions of DIC and phosphate that are not incorporated into marine organic carbon during surface ocean production.

### 2.2    Model spin-up with interglacial boundary conditions

We spun up the model with pre-industrial boundary conditions in three stages, sequentially coupling all modules. First, we forced the ocean circulation and then the atmosphere-ocean carbon cycle as a closed system with pre-industrial climatic condi-



tions and prescribed $CO_2$. In the next step, the sediment module is coupled and terrestrial solute supply (phosphate, alkalinity,

DIC, $DI^{13}C$ and Si) to the ocean is prescribed to balance loss to the lithosphere over 50 kyr. Then, this weathering input flux

is diagnosed and kept constant thereafter. Until this stage, atmospheric $CO_2$ and $\delta^{13}C$ were prescribed. The spun up model for

the pre-industrial was then run for 2000 years as an open system (freely evolving $CO_2$ and $\delta^{13}C$) with radiative forcing that

varied linearly from PI to the slightly different MIS19 conditions, the starting point of our experiments. To avoid large drifts in

carbon isotopes and alkalinity (Jeltsch-Thömmes and Joos, 2023) in the simulations with the forcings that perturbed the carbon

cycle the most (PO4, REMI, LAND, CO2T, BGC, ALL), we ran the fully-interactive model with the respective forcing for

three glacial cycles before starting our experiments.

## 2.3 Experimental design

We designed seven simplified forcings (table 1) to simulate the effects of Earth system changes that have been identified

as glacial-interglacial carbon cycle drivers (similar to the study of long-term circulation changes in Adloff et al. (2023)):

expanded Antarctic Bottom Water (AABW), here simulated by increased wind stess over the Southern Ocean, reduced sea-air

gas exchange in the Southern ocean, reduced solar radiation due to increased dust and aerosol fluxes, enhanced supply of

nutrients, reduced particulate organic carbon remineralization, lower rain ratio of particulate inorganic carbon (PIC, $CaCO_3$) to

particulate organic carbon (POC), and reduced terrestrial carbon storage during glacial times. We produced timeseries of these

forcings by defining a maximum forcing for the LGM, a minimum for the Holocene and then modulating this amplitude by

reconstructed relative changes in the temporal evolution of either Antarctic ice core $\delta D$ (Jouzel et al., 2007) or benthic $\delta^{18}O$

(Lisiecki and Raymo, 2005) for each year. The choice of the isotope record for calculating the instantaneous forcing depends

on whether we expect the forcing to evolve synchronously with temperature like $\delta D$ or have a time lag similar to $\delta^{18}O$.



**Table 1.** Forcing scenarios. Simulations are run in two configurations: the standard setup with interactive sediments and a closed-system setup without sediments (except PO4).

| ID | Description | LGM-PI amplitude | Modulating proxy |
|---|---|---|---|
| BASE | orbital changes<br>+ radiative effect of greenhouse gasses<br>+ ice sheet albedo | | $CO_2$, $CH_4$, $\delta^{18}O$ |
| SOWI | BASE + Wind stress strength<br>over Southern Ocean (>48 °S) | -40% | $\delta D$ |
| KGAS | BASE + gas transfer velocity<br>in Southern Ocean | -40% | $\delta D$ |
| AERO | BASE + Radiative forcing<br>from dust particles | -2.5 W/m$^2$ | $\delta^{18}O$ |
| PHYS | BASE + all physical<br>forcings combined | | |
| LAND | BASE + land C storage | -500 PgC | $\delta^{18}O$ |
| REMI | BASE + linear glacial<br>remineralization profile in upper 2000m | linear | $\delta D$ |
| PIPO | BASE + PIC:POC changes | -0.33 | $\delta D$ |
| PO4 | BASE + marine $PO_4$ reservoir | +30% | $\delta^{18}O$ |
| BGC | BASE + all biogeochemical<br>forcings combined | | |
| ALL | BASE + all forcings combined | | |
| CO2T | BASE + restoring reconstructed<br>atm. $CO_2$ concentrations | -90 ppm | $CO_2$ |




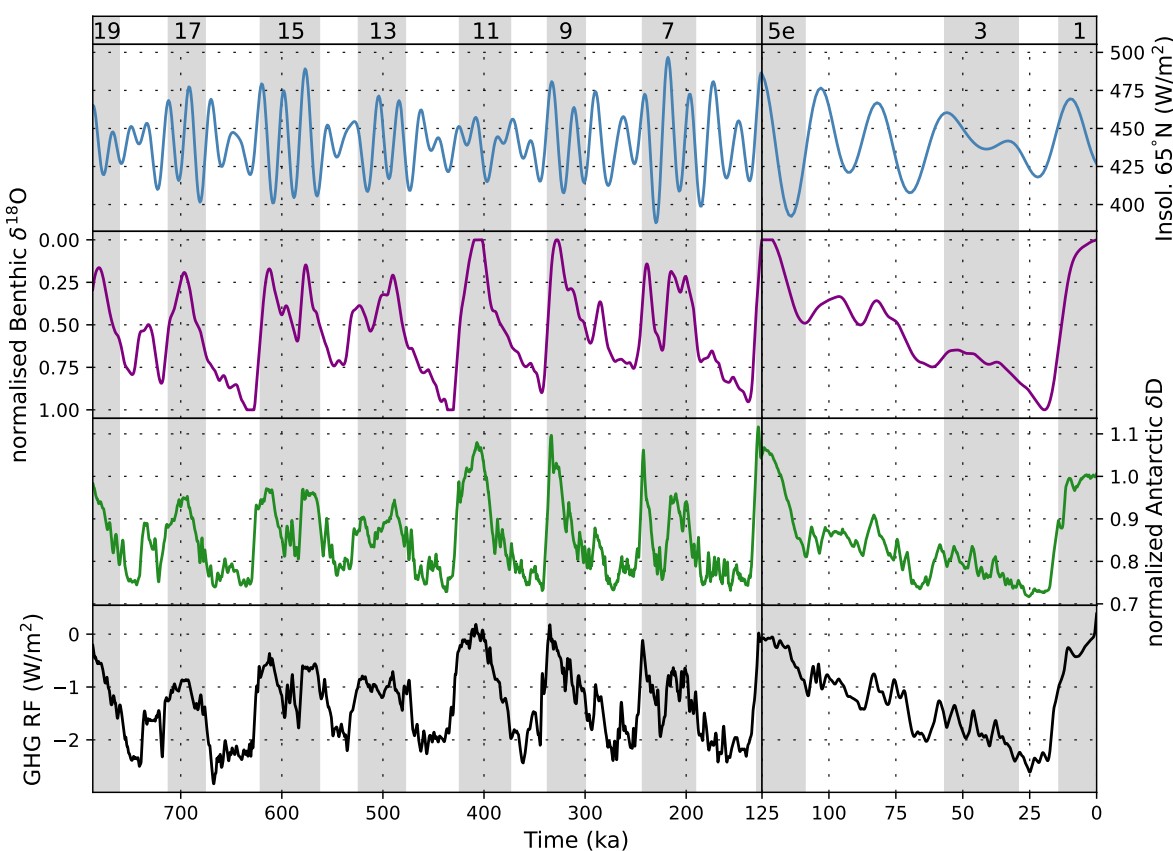

**Figure 1.** Forcing timeseries. Insolation changes are calculated according to Berger (1978); Berger and Loutre (1991). The $\delta^{18}O$ forcing is the LR04 stack (Lisiecki and Raymo, 2005), smoothed by averaging over a 10000-year moving window and normalized to the LGM-PI difference. The $\delta D$ forcing is from Jouzel et al. (2007) and normalized to the LGM-PI difference. The radiative forcing of $CO_2$ and $CH_4$ is calculated from Bereiter et al. (2015); Loulergue et al. (2008); Joos and Spahni (2008) following Etminan et al. (2016).



The application of the standard forcing in simulation BASE results in younger deep water masses in the Atlantic and Pacific during the LGM than at the PI (Pöppelmeier et al., 2020). To achieve an older glacial deep ocean, we reduced the wind stress

south of 48 °S by a maximum of 40% temporally changing proportionally to the $\delta$D change. As a result, the simulated deep ocean age is $\sim$100 years older in the LGM than in the PI, close to published model estimates (Schmittner, 2003). Changing wind stress only affects the circulation, not the piston velocity of gas exchange, which is forced by a wind-speed climatology. For an independent assessment of the effect of wind changes on sea-air gas exchange, we added a simulation in which we decreased the piston velocity in the Southern Ocean by a maximum of 40%, also following the evolution of $\delta$D. Finally, we tested

a negative radiative forcing due to increased dust loads in the glacial atmosphere (e.g. Claquin et al., 2003) by reducing the total radiative forcing by a maximum of 2.5 W/m$^2$ during the LGM, modulated by the $\delta^{18}$O record. In terms of biogeochemical forcings, we added a terrestrial carbon sink/source which removes/emits 500 PgC during deglaciation/ice age inception (Jeltsch-Thömmes et al., 2019) and increased the marine phosphate inventory by 30% during the LGM. The timeseries of both forcings are proportional to $\delta^{18}$O changes. We also reduced the speed of aerobic organic matter remineralization in the ocean

by transitioning between the standard, pre-industrial Bern3D particle profile (Martin scaling) during interglacials and a linear profile in the first 2000 m of the water column, following the $\delta$D record. Finally, we reduced the PIC:POC rain ratio by 30% in the LGM and modulated the forcing timeseries with the $\delta$D record.

We performed one 'base' run with orbital and radiative forcing only, one model run for each individual biogeochemical forcing added to the base forcing and combinations of forcings to study non-linear effects of forcing combinations. In addition

we performed one run in which we let the model dynamically adjust external alkalinity fluxes to restore the reconstructed atmospheric CO$_2$ curve. Alkalinity changes, e.g. due to changes in shallow carbonate deposition or terrestrial weathering, are an effective lever for atmospheric CO$_2$ change (e.g. Brovkin et al., 2007), and this additional run shows the long-term changes in marine biochemistry if these processes were the dominant drivers of glacial-interglacial atmospheric CO$_2$ change. All simulations were started from the pre-industrial spin-up adjusted for MIS 19 radiative forcing. Climate change occurs

only in response to the prescribed radiative forcing and not the simulated atmospheric CO$_2$ concentrations. To this set of 12 primary simulations, we added sensitivity experiments to assess biases induced by our experiment design: 1) We repeated the simulation ALL, including all forcings, with CO$_2$-sensitive climate to study the impact of the CO$_2$-climate feedback. 2) We repeated five simulations (REMI, PIPO, BGC, ALL, CO2T) starting from the transiently achieved MIS 15 Earth system state in their primary versions (215 kyr into the simulation) because they showed initial drifts in DIC and isotopes. We discuss the

relevance of initial conditions and imbalances of the geologic carbon cycle at the end of the manuscript.

For the discussion of the simulations, we quantify the factorial effect of the simulated processes on different carbon cycle metrics. In simulation BASE, only the standard forcing is active (see table 1), hence the factorial effect of the standard forcing forcing is equal to the simulated change:

$fBASE = $ BASE



In the simulations that combine the standard forcing with one other forcing, the factorial effect of the additional forcing is the difference between the respective simulation and BASE:

$$fFORC = \mathrm{FORC} - fBASE$$

In simulations PHYS, BGC and ALL several forcings are combined. We use these simulations to determine non-linearities by calculating the difference between the results of these simulations and the linear addition of the individual effects of the active
forcings:

$$nlPHY = \mathrm{PHYS} - (fBASE + fKGAS + fSOWI + fAERO)$$
$$nlBGC = \mathrm{BGC} - (fBASE + fREMI + fPO4 + fPIPO + fLAND)$$
$$nlADD = \mathrm{ALL} - \mathrm{PHY} - \mathrm{BGC} + \mathrm{BASE}$$
$$nlTOT = nlPHY + nlBGC + nlADD$$


    The impacts of ocean-sediment interactions and associated weathering-burial fluxes are quantified as differences between simulations with and without sediment module.

## 3   Results







**Figure 2.** Transient variations of atmospheric $CO_2$ concentrations as simulated in PHYS, PO4, REMI, and LAND and reconstructed by Bereiter et al. (2015). Shown is the deviation from the pre-industrial value. Gray shading indicates uneven Marine Isotope stages as indicated at the top of the figure. Dashed lines denote runs without sediment module (not available for PO4). The same plots for the other simulations are shown in S8.

The simulated amplitudes in $CO_2$ changes and their timings vary largely between simulations. The effect of dynamic sediments on the simulated atmospheric $CO_2$ strongly depends on the type of forcing. Interactive sediments have a negligible effect on atmospheric $CO_2$ changes in our simulations if only physical forcings are considered, but amplify glacial-interglacial $CO_2$ change under biochemical forcings.

The processes that cause the different model responses to the prescribed forcings have largely been described in other studies (e.g. Tschumi et al., 2008, 2011; Menviel and Joos, 2012; Menviel et al., 2012; Jeltsch-Thömmes et al., 2019; Jeltsch-



Thömmes and Joos, 2020), and are summarised in the SI to our manuscript, including a detailed analysis of the sedimentary changes simulated under the various forcings we tested. It is important to note that our simulations are designed to constrain the potential and plausibility of major contributions of the tested processes to the observed glacial-interglacial atmospheric $CO_2$ changes, rather than reproducing a full, realistic scenario. In their design, we prescribed the amplitudes and temporal evolution of each simulated process in a simplified way, while their true relative importance for glacial-interglacial carbon

cycle changes remains still unknown and might have varied spatially and temporarily. We therefore do not expect that any single simulation presented in our study captures all features of the reconstructed carbon cycle changes over glacial-interglacial cycles. Instead, we investigate the isolated processes, which occurred simultaneously in reality, and quantify their effects during eight consecutive glacial cycles. Comparing our results to proxy records, we can identify the dominant processes behind specific patterns in the reconstructions and the remaining challenges in reconciling the many carbon cycle reconstructions that are now

available.

## 4 Comparison with carbon cycle reconstructions





**Table 2.** Summary Model-Data comparison. Shown are the factorial effects of the tested processes and their non-linearites in comparison with reconstructed LGM - Holocene differences in various proxy systems. The direction of each arrow indicates whether a difference is positive (pointing upwards, teal-coloured) or negative (pointing downwards, brown-coloured). The width of the arrows shows the size of the difference relative to the reconstruction in the uppermost row "Data". For POM export and global preformed nutrient concentrations, only qualitative reconstructions exist. Hence, the arrows showing simulated effects are normed by the biggest effect by any simulated process.

| | $\Delta[CO_2]$ (ppm) | $\Delta$pH | $\Delta POM_{export}$ (g/m$^2$/yr) | | | $\Delta[PO_{4,reg}]$ (mmol/m$^3$) | $\Delta\delta^{13}$C (‰) | | | $\Delta[CO_3^{2-}]$ ($\mu$mol/kg) |
|---|---|---|---|---|---|---|---|---|---|---|
| loc. | global | Eq. Atl. | Iber. Marg. | Eq. Atl. | polar SO | global | shall. NA | deep NA | deep Pac. | deep Pac. |
| time | PI - 18 ka | 22 - 3 ka | 22 ka - PI | | | 21 ka - PI | 20 - 8 ka | | 22 ka - PI | 22 - 7 ka |
| Data | ↑ | ↑ | ↑ | ↑ | ↓ | ↑ | ↑ | ↓ | ↓ | ↓ |
| **Factorial Results** | | | | | | | | | | |
| $fBASE$ | | | | | ↓ | ↓ | | | | |
| $fKGAS$ | | o | | | o | | | | | |
| $fSOWI$ | | | | | o | ↑ | | | | |
| $fAERO$ | | | ↓ | | | | | ↓ | | |
| $fREMI$ | ↑ | ↑ | ↓ | ↓ | | ↑ | ↑ | ↑ | | ↑ |
| $fPO4$ | ↑ | ↑ | | | | ↑ | ↑ | ↑ | ↑ | ↑ |
| $fPIPO$ | | | | | | | | | | ↑ |
| $fLAND$ | ↓ | ↓ | | | | | ↓ | | | ↓ |
| $fCO2T$ | ↑ | ↑ | | | | | | | | ↑ |
| **Non-Linearities** | | | | | | | | | | |
| $nlPHYS$ | | o | | | | o | | | o | |
| $nlBGC$ | | | | | | ↓ | | | | |
| $nlADD$ | | | ↑ | | | ↓ | | ↓ | ↑ | |
| $nlTOT$ | ↓ | ↓ | ↑ | | | ↓ | | ↓ | ↑ | |
| section | 4.1 | 4.1 | 4.2 | 4.2 | 4.2 | 4.3 | 4.4 | 4.4 | 4.4 | 4.5 |

Table 2 provides a summary of the data-consistency of the factorial effects of each process we tested for selected carbon cycle proxies, and the non-linearities that arise when they are combined. The first row shows the reconstructed direction of LGM - Holocene differences, and the next lines show the direction and relative size (compared to the proxy signal) of changes induced by the various tested processes. The last three rows show the direction and relative size of non-linearities caused





by three different combinations of the processes above. For many of the considered proxies, these non-linearities are larger than without interactive sediments (not shown) but still small compared to the effect of individual processes, especially when combining only physical processes. For some proxies, the non-linearities are not negligible but still smaller than the biggest effect of an individual process. Hence, in most cases, proxy changes provide a first-order constraint on the plausibility of

large changes in individual processes. $fBASE$, the effect of temperature changes due to orbital, albedo and greenhouse gas changes, moves almost all proxy systems in the reconstructed direction (the directions of the arrows match), but almost never to the reconstructed extent (the widths of the arrows do not match). The only exception is strongly reduced export production in the polar Southern Ocean at the LGM, which in our model is predominantly driven by surface cooling and sea ice expansion regardless of which other processes were also changed. The mismatches in amplitudes of $fBASE$ and reconstructed changes

show that other processes must have affected the glacial carbon cycle, at least in our model. In the following sections, we expand on these model-data comparisons and discuss atmospheric $CO_2$, export production, the biological pump, sedimentary fluxes and carbon isotopes in more detail (see also section numbers at the bottom of table 2).

## 4.1 Atmospheric $CO_2$

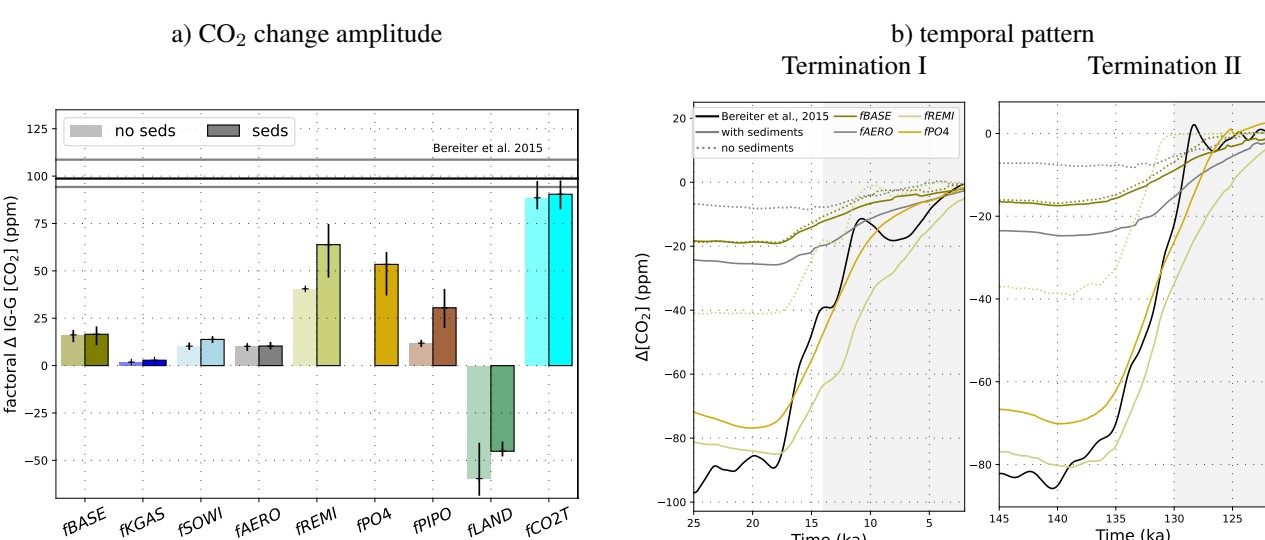

**Figure 3.** Atmospheric $CO_2$ changes across deglaciations. a) shows the factorial contributions to the mean glacial-interglacial $CO_2$ amplitude over the last five glacial terminations, as well as the range between their minimum and maximum. Light colors indicate results without interactive sediments, full colors indicate results with interactive sediments. The mean, minimum and maximum amplitudes over the last five deglaciations in the ice core record (Bereiter et al., 2015) are shown by the black and gray horizontal lines. b) shows selected factors transiently over the last two terminations.

By design, $CO_2$ restoring causes marine carbon uptake and release that shape atmospheric $CO_2$ in line with observations

(Fig. 3, S5 sum of fBASE and fCO2T, S8), so here we focus on the $CO_2$ changes simulated by the other forcings. PO4, REMI and PIPO produce the most consistent $CO_2$ difference between the LGM and PI with regard to the reconstructions (Fig. 3a). The timing of the lowest $CO_2$ values occurs during the coldest interval of glacial cycles, the glacial maxima, in all simulations



except LAND. However, in all simulations $CO_2$ keeps increasing throughout interglacials when temperature is almost constant. In simulations PO4 and CO2T, this is a consequence of the proxy records used to scale the respective forcing - benthic $\delta^{18}O$ for

PO4 and $CO_2$ for CO2T, which are not constant during interglacials (Fig. S17). In simulations AERO, REMI and PIPO, $CO_2$ lags the forcing by several thousand years during deglaciations and throughout interglacials (Fig. S17 and S18). In AERO, the lag is caused by the hysteresis behaviour of AMOC and exists in simulations with and without interactive sediments. In REMI and PIPO, the time lags are caused by weathering-burial imbalances, particularly a large build-up of alkalinity and DIC during the glacial phase which is only gradually reduced by enhanced $CaCO_3$ burial during deglaciation (Fig. S22). The different

correlations between temperature and $CO_2$ in glacials and interglacials is also a feature of the ice core records (Brovkin et al., 2016), though the timing of peak interglacial $CO_2$ concentrations and transient features of the atmospheric $CO_2$ increase during deglaciations in the records is not necessarily reproduced in our idealised simulations with simplified forcings (Fig 3b, right panel). In most simulations, the weathering-burial disequilibrium, which builds up over the glacial phase, amplifies the glacial $CO_2$ drawdown.

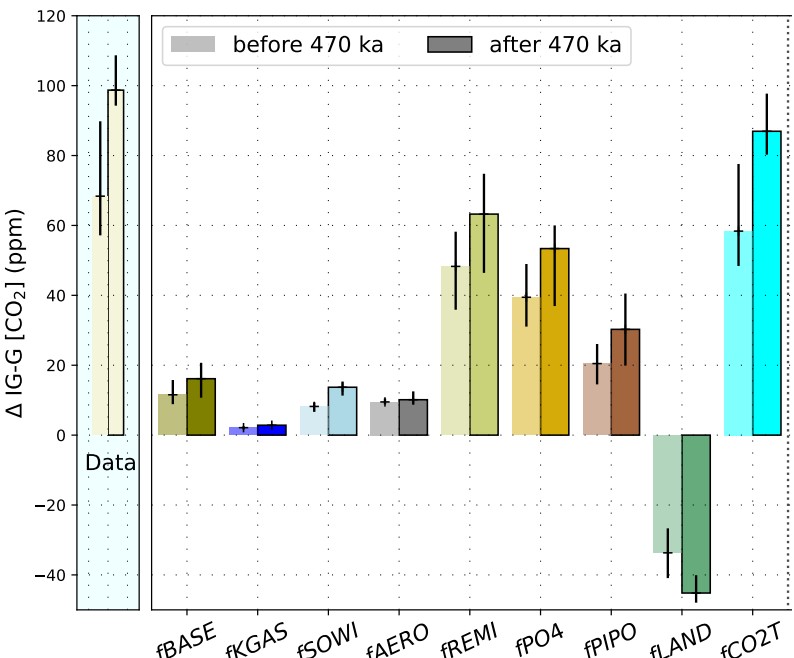

**Figure 4.** Difference of the glacial-interglacial atmospheric $CO_2$ amplitude before and after the MBT in our simulations compared to that in the reconstructed $CO_2$ record (Bereiter et al., 2015). For the results of the simulations without interactive sediments see Fig. S16

Burial-weathering imbalances also affect the difference between the glacial-interglacial $CO_2$ amplitudes of the last five glacial terminations compared to the glacial cycles before. Since maximum temperatures of interglacials were larger during the last four interglacials, the solubility of $CO_2$ in seawater was lower, contributing to higher atmospheric $CO_2$ levels. This creates a small direct response to the forced interglacial warming after the MBT under the standard forcing (fBASE, ~5 ppm, Fig.4). Wind-driven circulation changes in the Southern Ocean (SOWI) have a stronger effect, because in colder interglacials,





the deep Pacific remains more isolated than during warm interglacials, preventing the release of an additional 5 ppm in the earlier glacial cycles. Reduced remineralization rates (REMI) and increased nutrient supply during colder interglacials have the strongest individual effect on carbon storage differences across glacial cycles, each reducing the glacial-interglacial $CO_2$ difference by 15 ppm in the early, colder glacial cycles. As already noted in the discussion of absolute $CO_2$ concentrations, weathering-burial imbalances play a minor role for the simulated physical effects on the glacial-interglacial $CO_2$ amplitude

but amplify the biochemical effects. In fact, only by including interactive sediments does our model simulate a shift in the MBT glacial-interglacial $CO_2$ amplitude comparable to the observations (Fig. 4). In most simulations, sedimentary $CaCO_3$ deposition is reduced during glacials and marine alkalinity and DIC build up as a consequence. A large fraction of the glacial DIC pool is eventually incorporated into $CaCO_3$ and deposited during deglaciations, simultaneously drawing down alkalinity and thus releasing $CO_2$ to the atmosphere. One effect of interactive sediments is thus the larger of $CaCO_3$ burial and alkalinity

draw-down during bigger temperature rises at the onset of warm interglacials, which in our simulations is amplified by changes in PIC:POC (PIPO) and the remineralization profile (REMI). Another effect responsible for the larger $CO_2$ amplitude changes under biochemical forcings is the reduction of sedimentary organic carbon burial rates during interglacials under increased nutrient supply (PO4) or a flattened remineralization profile (REMI) during glacial phases. During deglaciations, sedimentary POC deposited during glacials is remineralized, which raises atmospheric $CO_2$ in addition to the previously described alkalinity

changes. The larger the temperature rise during deglaciation, the more POC is remineralized.

In summary, most of the simulated processes cause atmospheric $CO_2$ increases during deglaciations, and the good matches with various features of reconstructed glacial-interglacial $CO_2$ changes are produced by biochemical changes. Variations of the remineralization profile (simulation REMI) cause the largest deglacial $CO_2$ change but at a slower pace than reconstructed. Nutrient supply changes in simulation PO4 cause the fastest deglacial $CO_2$ rise but with a lower amplitude than REMI. All

features of the reconstructed $CO_2$ curve can artifically be achieved by changing external alkalinity fluxes (CO2T). However, the $CO_2$ changes in these simulations are caused by different mechanisms which also leave traces in the proxy record.

## 4.2   Marine DIC and the surface carbonate system



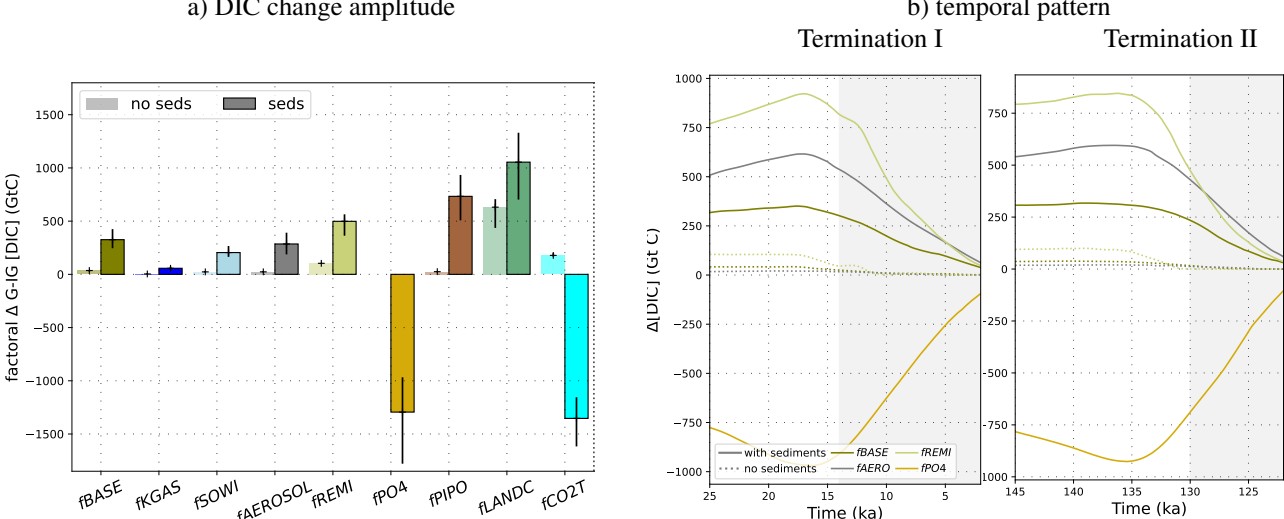

**Figure 5.** Factorial DIC concentration changes for each process across deglaciations. In a),factorial contribution of each process to the mean glacial-interglacial $CO_2$ amplitude over the last five glacial terminations, as well as the range between their minimum and maximum. Light colours show the without dynamic sediments, and full colours show the contributions with dynamic sediments. In b), the absolute temporal DIC evolution across two terminations is shown for selected simulations with and without interactive sediments.

The simulated $CO_2$ changes are the result of changes in carbon uptake in the surface ocean. All forcings strongly affect air-sea carbon fluxes in the Southern Ocean, by changing SST, sea ice, circulation, productivity and/or the carbonate system in

the surface ocean. It has long been recognised that the Southern Ocean played an important role in glacial-interglacial changes in the partitioning of carbon between atmosphere and ocean (Wenk and Siegenthaler, 1985; Fischer et al., 2010). In addition to changes in the Southern Ocean, in our simulations surface carbonate system shifts in the non-polar oceans are required to achieve glacial-interglacial $CO_2$ differences that are consistent with reconstructions. In simulations REMI, PO4 and CO2T, which produce the most data-consistent atmospheric $CO_2$ changes, such carbonate shifts occur: Surface ocean pH increases

towards the glacial maximum, which is consistent with reconstructions (Fig. S22, Hönisch and Hemming, 2005; Shao et al., 2019). It is noteworthy that in simulation REMI with interactive sediments, surface ocean pH changes align closely with those in simulation CO2T, in which the model was forced to reproduce reconstructed atmospheric $CO_2$ changes. This alignment is visible throughout the glaciation up until the onset of the deglaciation, where pH changes in REMI lack behind those in CO2T. During the deglaciation, pH changes in simulation PO4 show the same speed as in CO2T, even though the amplitude is lower.

This mirrors our results from the comparison of the simulated atmospheric $CO_2$ curves, which showed that REMI produced the most data-consistent $CO_2$ changes during the glaciation but failed to reproduce the quick $CO_2$ release during the deglaciation. The spatial patterns of marine carbon uptake and release in our simulations are thus driven by combinations of physical and chemical processes. The net carbon fluxes into the ocean during glaciation and out of the ocean during deglaciation are roughly of equal magnitude in the Southern Ocean (>40 °C S) as in the rest of the ocean combined, but of opposite sign, under most

forcings and with and without interactive sediments (Fig. S26).





Due to interactive sediments in our simulations, increased uptake or release of carbon in the surface ocean does not linearly correlate with DIC changes because marine carbon storage is also affected by changes in the deposition and dissolution fluxes of particulate carbon at the ocean-sediment interface. Interactive sediments affect marine carbon, alkalinity and nutrient concentrations in two important ways: Firstly, sediments form a large transient reservoir which can store and release large amounts of carbon and nutrients for hundreds to tens of thousands of years. Secondly, sedimentary mass accumulation, dissolution and remineralization rates control lithological burial, the only permanent sink for carbon and nutrients in our simulations and the only mechanism to cause imbalances between supply from weathering and sedimentary burial. Fluxes into and out of the sediments respond to environmental change, in some cases on the timescale of water mass replacement or regional productivity changes. Consequently, the simulated DIC change is much larger than the simulated change in atmospheric carbon under physical and biochemical forcings. In our simulations the ocean inventory changed by 200-1400 GtC and the atmospheric inventory by 1-150 GtC over the last deglaciation. DIC changes differ by a factor of up to 28 between simulations with and without interactive sediments, while $CO_2$ changes in the atmosphere are maximally four times larger when interactive sediments are considered (Figs 5, S2, S3, S4, S6, S10, S12, S13). Furthermore, the onset of the deglacial $CO_2$ rise in simulations with sediments does not always coincide with the onset of the deglacial DIC change, as is the case in simulations without sediments. This is simulated e.g. for terminations I and II in LAND and CO2T (Fig. S20), and for terminations I, II, III and IV in PO4 and ALL (Fig. S19). This suggests that reconstructed DIC changes, e.g., based on radiocarbon (Sarnthein et al., 2013) may be uncertain and do not necessarily imply a comparable $CO_2$ drawdown from the atmosphere. In simulations PO4 and CO2T, interactive sediments further produces the counter-intuitive result of reduced marine carbon storage during glacial maxima, because in these simulations excess carbon is stored in sediments as carbonate and organic carbon. In the following, we constrain these processes further with additional proxy data.

### 4.3 Export production changes during glacial-interglacial cycles

Export of biogenic matter from the surface ocean exerts a strong control on the carbonate system in the surface ocean and thus the sign and magnitude of the $CO_2$ flux across the sea-air interface (Volk and Hoffert, 1985). Decreasing temperatures, sea ice expansion, and curtailed nutrient supply due to a weakened circulation during glacials reduce export production in simulations with only physical forcings. In simulation PO4, the prescribed enhanced external nutrient input during glacial phases increases export production, while in simulation REMI, slowing remineralization rates in the surface ocean has the opposite effect by reducing the amount of nutrients in the surface. The lower PIC:POC ratio of simulation PIPO reduces export production of $CaCO_3$.





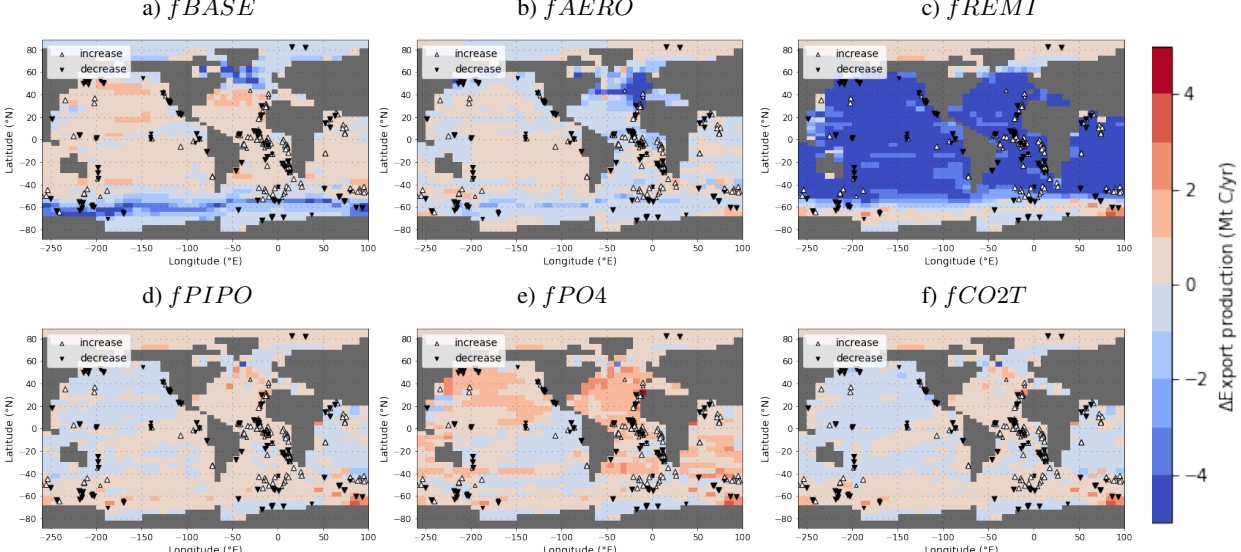

**Figure 6.** POC export production differences between the LGM and the PI in simulations BASE, PHYS, REMI, PIPO, PO4 and CO2T with interactive sediments. The filled triangles show the reconstructed sign and qualitative magnitude of change based on Kohfeld et al. (2005): black triangles show reconstructed decreases, white triangles show increases. Big triangles show big reconstructed changes, small triangles show small reconstructed changes.

Reviewing changes in various productivity proxies, Kohfeld et al. (2005) reconstructed spatial patterns of productivity
changes between the Holocene and the LGM. Specifically, they found lower productivity in the polar oceans and increased
productivity in the subpolar Southern Ocean and the tropical and subtropical Atlantic during the LGM. The direction of pro-
ductivity changes in the Southern Ocean and upwelling areas in our simulations align with the reconstructions in most simula-
tions, particularly with additional cooling and circulation slowdown with the physical forcings (Fig 6). These changes induced
by the physical forcing reduce productivity in the polar oceans and, to a lesser extent increase productivity elsewhere as nu-
trients are redistributed. AMOC weakening reduces productivity in the North Atlantic. Consequently, simulations that contain
a strong AMOC weakening (AERO, PHYS) fail to reproduce the reconstructed strong productivity increases in the tropical
and North Atlantic. Such a North Atlantic productivity increase is only simulated when AMOC changes are small or external
nutrient supply into the North Atlantic is enhanced during glacials (simulation PO4). Increased external nutrient input during
glacial phases causes large productivity increases in most of the extrapolar ocean, which is consistent with the reconstructions.
The comparison of magnitudes of productivity change between our simulations and reconstructions is more uncertain than
comparing the sign of change. Therefore, it must be noted that the standard forcing in BASE also produces small productivity
increases in most extrapolar oceans. Yet, the resulting changes in surface ocean carbonate chemistry are far to small to cause
data-consistent $CO_2$ changes, which are better reproduced in simulation PO4. Deepening the mean remineralization depth in
simulation REMI produces the biggest mismatch between simulations and reconstructions of export production, although it
captures the changes in atmospheric $CO_2$ well (see previous section). This mismatch indicates that the process behind the





simulated carbonate system changes that produced the data-consistent $CO_2$ changes, large reductions in export production, is not compatible with productivity proxies and did likely not play a dominant role in the real world glacial carbon cycle changes.

## 4.4 The efficiency of the soft tissue pump during glacial-interglacial cycles

The interplay between export production and ocean circulation influences the strength of the soft tissue pump, i.e. the degree of nutrient depletion in the euphotic zone and the magnitude of the vertical nutrient and DIC gradients (Volk and Hoffert, 1985). Two approaches have been developed to quantify the efficiency of the soft tissue pump: nutrient utilization in the surface ocean Sarmiento (2006) and the fraction of regenerated nutrients in the ocean's interior (Ito and Follows, 2005). No direct proxy exists for these metrics across glacial cycles, but several proxies have been used to constrain their evolution indirectly (see reviews in Galbraith and Skinner, 2020; Sigman and Hain, 2024). Nutrient concentrations have been inferred directly from reconstructed $\delta^{13}C$ of DIC and dissolved cadmium concentrations based on modern day correlations, and indirectly from export production reconstructions (e.g. Anderson et al., 2002). Hertzberg et al. (2016) inferred the vertical DIC gradient from the reconstructed vertical $\delta^{13}C$ gradient, and (Yu et al., 2019) from the reconstructed vertical $CO_3^{2-}$ gradient. These studies suggested that the biological pump was more efficient during the last glacial maximum, i.e. that nutrient utilization was greater and that vertical gradients of DIC were larger in the Southern Ocean, South Pacific and Atlantic, Eastern Equatorial Pacific and North Atlantic. Tools have also been developed to infer regional to global changes in the soft tissue pump efficiency. The deep ocean oxygen concentrations provide constraints on the amount of deep ocean regenerated DIC (Galbraith and Jaccard, 2015) and Vollmer et al. (2022) inferred preformed phosphate concentrations from reconstructed AOU. These studies concluded that regenerated DIC was increased and preformed phosphate decreased in the deep ocean during the last glacial maximum, inferring a net increased efficiency of the global soft tissue pump.



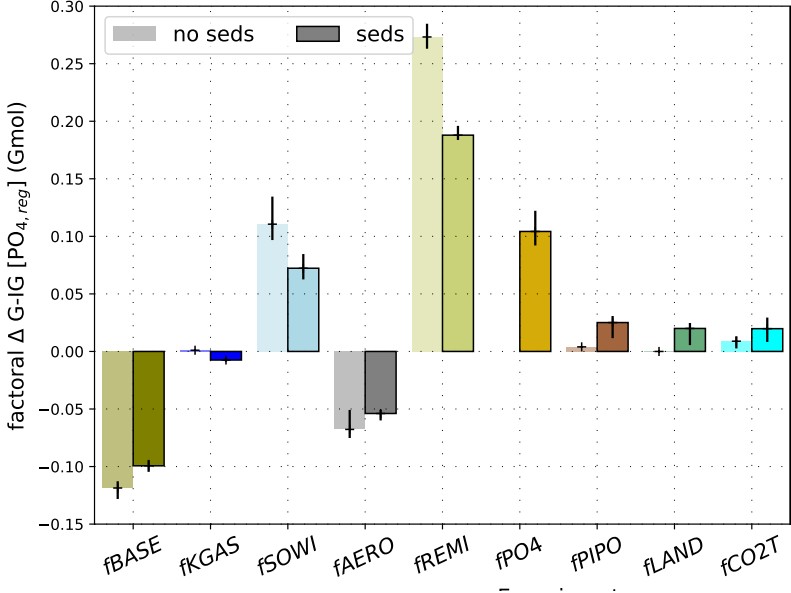

**Figure 7.** Glacial-Interglacial amplitudes of $[PO_{4,reg}]$ changes in all simulations. For BASE, the total amplitude is given, for the other simulations, the additional effect of each forcing to the standard forcing in BASE is shown. Error bars indicate the range of the amplitude across the last five deglaciations.

There are thus indirect proxy constraints for an increased efficiency of the soft tissue pump at the last glacial maximum.
We set up one simulation, REMI, with reduced remineralization rates during periods of lower $\delta D$ in the ice core record, i.e.
colder climate, which directly increases the soft tissue pump efficiency. As previously discussed, the resulting increased soft
tissue pump efficiency in the glacial ocean has a large potential to draw down atmospheric $CO_2$ and cause the reconstructed
accumulation of respired carbon in the glacial ocean, consistent with other modelling studies (e.g. Menviel et al., 2012; Morée
et al., 2021). Despite this large potential to cause $CO_2$ changes, the soft tissue pump changes in REMI and SOWI do not
produce proxy-consistent changes in export production (see above), $\delta^{13}C$, $[CO_3^{2-}]$ and $CaCO_3$ burial (see table 2 and the next
sections). A scenario of several simultaneously occurring processes with different effects on $CO_2$ and nutrient concentrations
over glacial cycles seems thus more likely than a dominance of soft tissue pump changes. Our simulations also show that a
small contribution of soft tissue pump efficiency changes to atmospheric $CO_2$ changes is not necessarily in contradiction with
the proxy record, as processes that alter the total inventories of nutrients and carbon in the ocean can affect nutrient distributions
independently of atmospheric $CO_2$.

With and without interactive sediments, the soft tissue pump efficiency increase in simulation REMI results in increased
nutrient utilization in the surface ocean and reduced preformed nutrients in the ocean's interior during glacial phases (Figs 7,
S23, S24). Although these metrics by definition do not scale linearly when we consider supply-burial imbalances of phosphate
(Fig S24, S23), both metrics suggest that the simulated soft tissue pump efficiency increases during glaciation in REMI. How-
ever, the simulated accumulation of regenerated nutrients in the deep ocean is smaller with interactive sediments, even though
the simulated glacial-interglacial $CO_2$ changes are larger. Without interactive sediments, the vertical nutrient gradient during



glacial phases is steepened primarily because a larger fraction of regenerated phosphate, released from decaying organic matter, accumulates in the ocean's interior. The accumulation of nutrients in the deep ocean is lower in the simulation with interactive

sediments because the resulting depletion of $O_2$ in the deep ocean increases organic carbon burial, and thus phosphate and carbon removal from the ocean. In simulations in which atmospheric $CO_2$ changes are dominated by processes other than changes in the soft tissue pump efficiency, changing vertical nutrient gradients can also anti-correlate with atmospheric $CO_2$, even in simulations with a constant marine nutrient inventory and no interactive sediments. For example, this is the case with the standard forcing (BASE) and alkalinity supply changes (CO2T): nutrient utilization and regenerated nutrient concentra-

tions are lower at glacial maxima (Fig. 7, S24) because of reduced export production in a colder ocean with increased sea ice cover. In simulation CO2T, marine alkalinity changes have large effects on atmospheric $CO_2$, while having a marginal effect on marine nutrient distributions (S24) by changing the strength of the carbonate pump.

## 4.5    $\delta^{13}$C in the atmosphere and ocean







**Figure 8.** $\delta^{13}$C over the last recent glacial cycle in various reservoirs. Solid lines are simulations AERO, CO2T, LAND, and PO4. Dotted lines are reconstructions from the Pacific cores ODP 677 and ODP 847, combined and smoothed by Köhler and Bintanja (2008), regional stacks of sediment cores in the North and South Atlantic (Barth et al., 2018), and Talos Dome ice Eggleston et al. (2016). Results for each individual forcing in an open and closed system are displayed in Fig. S30, S31, S32, S33, S34, S35.

Ice cores preserve the $\delta^{13}$C signature of atmospheric $CO_2$ (Friedli et al., 1984), which showed large fluctuations during the
last glacial cycle (Fig. 9), such as fluctuations of ∼0.5 ‰ during MIS 4 (71-57 ka, Fig. 8) and during the last deglaciation
(∼ 18-8 ka). They also show a long-term trend of lower atmospheric $\delta^{13}$C during the Eemian than the Holocene (Schneider
et al., 2013; Eggleston et al., 2016). None of the forcings that we applied here produce all of the reconstructed features. In our
simulations, the $\delta^{13}$C signature of atmospheric $CO_2$ is influenced by changes of carbon exchange with the ocean (and land in
the case of LAND, BGC and ALL) and the spatial pattern of $\delta^{13}$C in the surface ocean, with any signal diluted by the 4-box
land biosphere. With and without interactive sediments, the cooling induced by the standard forcing in simulation BASE leaves





atmospheric $\delta^{13}$C virtually invariant except for minor positive excursions of $\sim$0.05 ‰ as the deep ocean is ventilated during deglaciations (Fig. S30). The largest $\delta^{13}$C variability (up to $\pm$0.5 ‰) is produced by changing organic matter remineralization (REMI) or organic carbon storage (PO4 and LAND). Yet, the gradual trend of reconstructed atmospheric $\delta^{13}$C over the last glacial cycle (+$\sim$0.5 ‰ from inception to LGM) is only achieved in simulation PO4, the forcing with the biggest effect on

sedimentary organic carbon storage (Fig. 8). LAND causes similar long-term changes but of the opposite sign. No simulation captures the large millenial-scale fluctuations in the reconstructions (Fig. 9). Given our smoothed forcing and the absence of freshwater forcings, our simulations do not contain realistic millenial-scale circulation changes, which would likely be required to simulate these fluctuations (Tschumi et al., 2011; Menviel et al., 2015). However, unlike in PO4, REMI and LAND, the maximum $\delta^{13}$C change produced by circulation and temperature changes (PHYS) is only of roughly half the amplitude

of the $\delta^{13}$C swing during the deglaciation. It is established that a complex combination of processes is required to explain the atmospheric $\delta^{13}$C record (e.g. Menviel et al., 2015) but a simulation over the last glacial period or the deglaciation accurately reproducing the reconstructions has not yet been achieved, and reconciling reconstructed with simulated atmospheric $\delta^{13}$C remains a major challenge for future work.

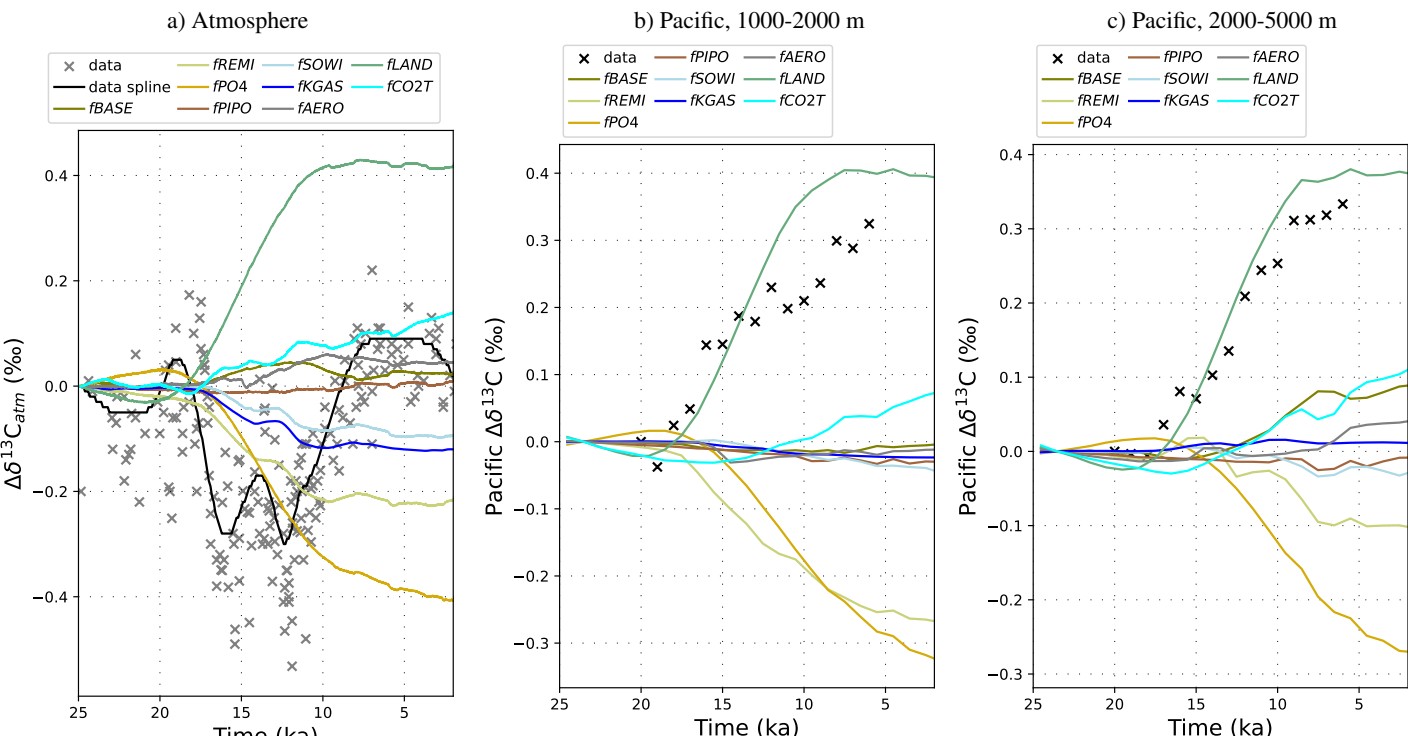

**Figure 9.** $\delta^{13}$C over the last deglaciation in (a) the atmosphere, (b) intermediate and (c) deep Pacific ocean. Lines are from simulations with an open system. Crosses are reconstructions from Schmitt et al. (2012), Eggleston et al. (2016) and Peterson and Lisiecki (2018). Results for the closed system are displayed in Fig. S29.



The $\delta^{13}$C signature of marine DIC in a given location is influenced by processes which affect the whole marine carbon

reservoir (e.g. changes in the size and composition of marine carbon input or output fluxes), as well as by changes in water mass distribution and isotopic fractionation during sea-air gas exchange and primary production, which change the isotopic composition of local DIC (Jeltsch-Thömmes et al., 2019; Jeltsch-Thömmes and Joos, 2023). Changes to the whole marine carbon reservoir are the dominant cause of DIC changes in LAND and CO2T. In simulation LAND, $\delta^{13}$C changes are driven by the simulated release of isotopically light land carbon (-24 ‰) during glacial inceptions and throughout the glacial, resulting

in $\delta^{13}$C minima in all reservoirs during glacial maxima and large $\delta^{13}$C increases during deglaciation in response to land carbon uptake (Fig. 8, 9). Equally a $\delta^{13}$C$_{DIC}$ decrease during the second half of the glacial phase, albeit of lower magnitude, occurred in simulation CO2T, driven by lower CO$_2$ and a consequent shift in fractionation that causes increased incorporation of isotopically heavy carbon into biogenic export and burial and therefore removal of isotopically heavy carbon from the ocean (Fig. S5). Compared to LAND and CO2T, deep ocean $\delta^{13}$C changes were inverted in simulations REMI and PO4: deep ocean

$\delta^{13}$C increased from glacial inception towards the glacial maximum (Fig. 8, S31) as more POC is buried, removing isotopically light C from the system. In these simulations, during terminations (and in simulation PO4 also during interglacials), a decrease of $\delta^{13}$C is simulated in all dynamic carbon reservoirs as CaCO$_3$ burial peaks temporarily and POC burial fluxes return to inter-glacial levels (Jeltsch-Thömmes et al., 2019). In addition, reduced POC fluxes to the sediments and increased ventilation of the deep ocean result in remineralization events of previously deposited sedimentary organic matter. While these tendencies

of deep ocean $\delta^{13}$C occur globally in simulation PO4, they are restricted to the Atlantic, particularly the North Atlantic, in simulation REMI.

Because the sensitivities of $\delta^{13}$C$_{CO_2}$ and $\delta^{13}$C$_{DIC}$ are different, and $\delta^{13}$C$_{DIC}$ varies between ocean basins, the forcings which best reproduce reconstructed $\delta^{13}$C changes also vary between atmosphere and ocean, and specific water masses. In the North Atlantic, nutrient input (PO4) and deepening of the remineralization depth (REMI) result in $\delta^{13}$C changes at intermediate

depth that are similar to observations (green curve in Fig. 8 and Fig. S31). Yet, under these forcings deep ocean $\delta^{13}$C outside the North Atlantic does not evolve as reconstructed. Deep ocean $\delta^{13}$C in the Pacific and South Atlantic are best reproduced by alkalinity nudging (CO2T) and land carbon release (LAND, blue and yellow lines in Fig. 8). LAND and CO2T are also the only simulations with a persistent $\delta^{13}$C increase in the deep and abyssal Pacific during the last deglaciation, with LAND roughly reproducing the reconstructed amplitude (Fig. 9). The $\delta^{13}$C record of the deep North Atlantic is best reproduced by

additional radiative cooling in simulation AERO, due to the resulting AMOC shoaling (purple curve in Fig. 8). This indicates that different processes were likely the dominant controls on $\delta^{13}$C regionally, even if they were not necessarily the dominant drivers of atmospheric CO$_2$. In our simulations, the simulated $\delta^{13}$C at intermediate depth in the North Atlantic is most data-consistent in PO4, followed by AERO, the latter of which also reproduces the reconstructed changes in the deep Atlantic. $\delta^{13}$C at intermediate depth in the North Atlantic might thus have been dominated by changing export fluxes and circulation changes,

while the latter dominated $\delta^{13}$C in the deep North Atlantic. In the deep Pacific and South Atlantic, accumulation of isotopically light $\delta^{13}$C as in simulation LAND likely dominated in the deep Pacific and South Atlantic. None of our simulations show lower marine benthic $\delta^{13}$C during the Eemian than during the Holocene, as reconstructed from marine sediments (Bengtson et al., 2021) and which may be linked to changes in weathering fluxes not considered here.



## 4.6 Sedimentary burial and $CO_3^{2-}$ concentrations

Changes in the simulated sedimentary burial fluxes result in net transfers of up to 2000 PgC between the carbon pools of
the ocean and sediments throughout a glacial cycle, while the net transfer between the ocean and atmosphere is roughly 200
PgC, and between the terrestrial biosphere and atmosphere on the order of 500-1000 PgC (Jeltsch-Thömmes et al., 2019). The
carbon cycle impact of glacial cycles was thus likely larger in the ocean than in the atmosphere (Roth et al., 2014; Buchanan
et al., 2016), due to changes in sedimentary carbon storage. How realistic are the simulated fluxes?

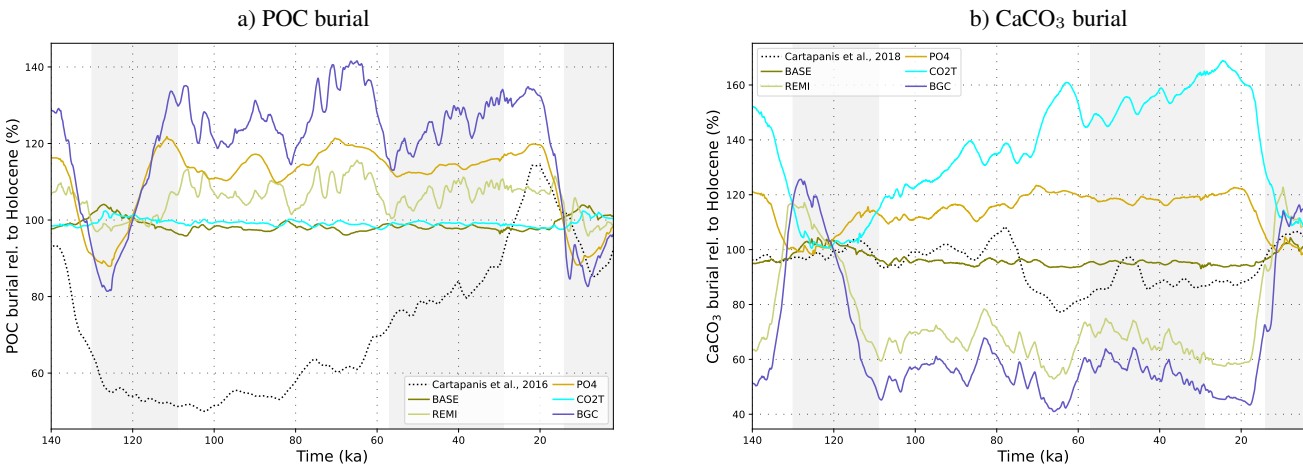

**Figure 10.** Simulated and reconstructed (Cartapanis et al., 2016, 2018) relative changes of global POC and CaCO$_3$ burial changes over the
last glacial cycle. Gray shading indicates odd isotope stages (MIS5e, MIS3 and MIS1).

440       Reconstructions of global POC burial flux changes over the last glacial cycle (Cartapanis et al., 2016) found that POC burial
was smallest during interglacials, and gradually rose during glacial phases until it peaked during the LGM (Fig. 10a, see also
simulated and reconstructed POC burial rates across the last glacial cycle in Figs S1, S3, S4, S5, S6). Simulations REMI
and PO4 are the only single-forcing simulations which produce higher POC burial fluxes during glacial maxima than during
interglacials. However, in these simulations, the simulated increase in POC burial already occurs during inception, such that the
highest burial rates persist throughout most of the glacial phase while in the reconstructions they remain below the Holocene
value until MIS2. PO4 also roughly reproduces the magnitude of POC burial changes during the last deglaciation but fails
to produce the reconstructed low POC burial rates at the end of the Eemian. The large reconstructed POC burial reduction
after the Eemian is only simulated when all biochemical forcings are combined (BGC), and also then is not sustained into
MIS4 as indicated by the reconstructions. The sedimentary record indicates that the glacial increase in POC burial occurred
predominantly in the Southern Ocean (Cartapanis et al., 2016). In the model, surface ocean cooling and sea ice expansion
reduce export production in the Southern Ocean during glacial inceptions, which reduces burial rates. External nutrient supply
to the glacial Southern Ocean increases the flux of organic particles to the sediments relative to the other forcings and can
counteract the climate-driven productivity reductions. Simulation PO4 is thus the simulation which most closely aligns with



the geographic pattern of reconstructed POC burial changes during glacial inceptions. In REMI, POC burial remains unchanged in the Southern Ocean and instead increased on continental margins in the Pacific and Indian Oceans. In either case, we have to recall that in our simulations the forcings are simplified to be globally uniform, which might affect the spatial pattern of simulated burial changes. During deglaciations, the reconstructions show a strong decrease of POC burial in the Southern ocean, while most other ocean areas see burial increases during deglaciations. The latter are captured in most simulations, and are a response to surface ocean warming and enhanced upwelling of nutrients that previously accumulated in the deep ocean during the glacial, but the changes in the Southern Ocean are again best reproduced in simulation PO4.

Reconstructions of global $CaCO_3$ burial changes over the last glacial cycle (Cartapanis et al., 2018) show that burial rates decreased in most ocean basins during glacial inception, while they increased in the Southern Ocean, resulting in only minor glacial-interglacial changes in the global average. Physical forcings result in roughly constant $CaCO_3$ burial rates during glacial inception, as suggested by the reconstruction, while simulations PO4 and CO2T display burial increases and all other simulations decreases between MIS5 and MIS3. The simulations of physical forcings also reproduces decreased $CaCO_3$ burial rates starting in MIS4, though at a smaller amplitude than reconstructed (Fig. 10b), while biochemical forcings produce much larger changes and mostly inreased burial. The error in amplitude under biochemical forcings is particularly large during terminations (see also Figs S1, S3, S4, S5, S6). However, biochemical forcings produce more data-consistent spatial patterns of $CaCO_3$ burial changes. Simulations PO4 and CO2T are the only setups which produced the reconstructed amplified $CaCO_3$ burial in the Southern Ocean between MIS5 and MIS3 (Cartapanis et al., 2018). In PO4, burial rates increase due to increased productivity, and in CO2T due to higher sedimentary preservation in the open ocean (e.g. due to less deposition on shelves or more alkalinity input from weathering). In addition, burial rates continue to rise through the LGM in the Southern Ocean in both simulations, while the reconstructions show that burial rates decreased from MIS 3. It is possible that constraining nutrient supply to the Southern Ocean, e.g. by assuming that meteoric iron fertilization was the dominant nutrient forcing, could change the spatial pattern and result in more data-consistent burial changes during glacial inceptions. The reconstructed burial rate decline between MIS 3 and the LGM, which is missing in PO4, would then require that one of the other forcings became dominant over changes in nutrient supply. In our simulations, the other forcings all caused Southern Ocean $CaCO_3$ burial rates to reach a minimum during the LGM, consistent with the reconstructions.



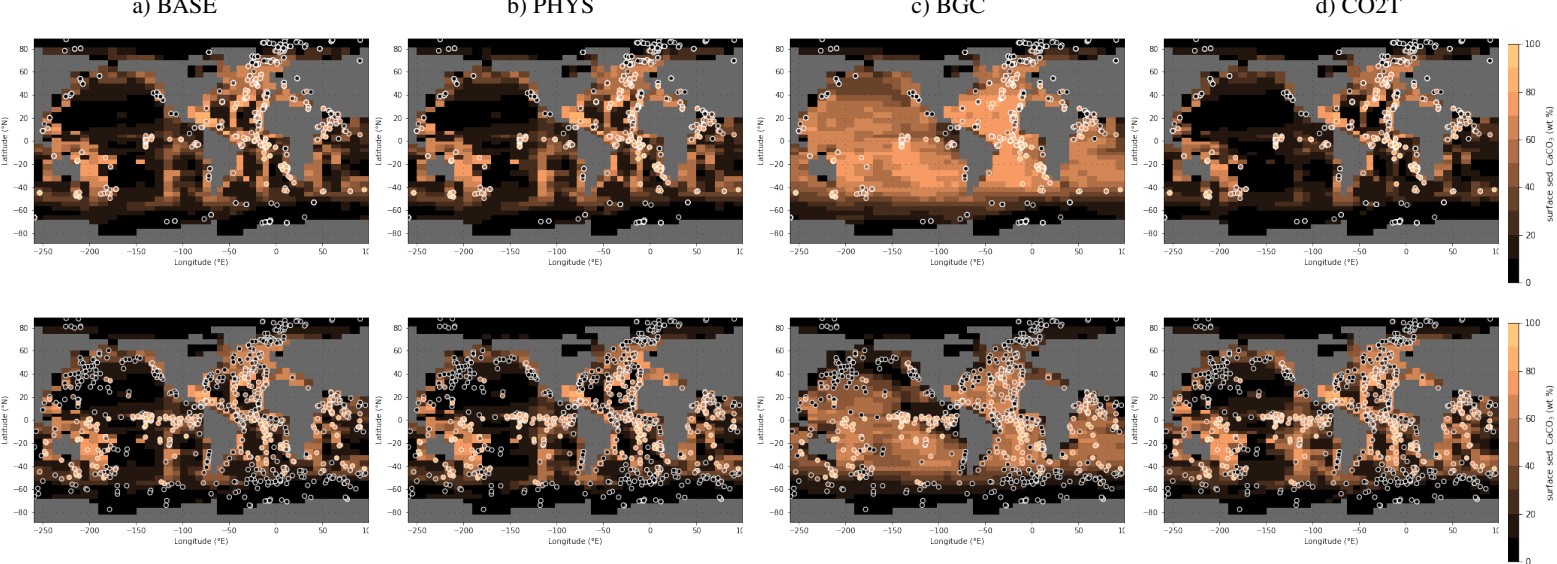

**Figure 11.** Sedimentary CaCO$_3$ fraction during the late Holocene (top row Cartapanis et al., 2018) and LGM (bottom row Wood et al., 2023) as reconstructed (circles) and in simulations BASE, PHYS, BGC and CO2T (underlying maps).

Model-data comparison for the CaCO$_3$ content of surface sediments is more challenging because of the simplified assump-
tion of globally uniform detrital fluxes to the sediment and the coarse model resolution which does not resolve sea mounts and
steep slopes. Yet, large-scale resemblance or mismatch of simulated and reconstructed sedimentary CaCO$_3$ fraction can still
offer some constraint on the reasonability of the simulated processes. For the LGM, the simulated distribution of sedimentary
CaCO$_3$ under the standard forcing (BASE) shows good agreement with the reconstructions by Wood et al. (2023) in most ocean
basins (Fig. 11). The low CaCO$_3$ content at the North Pacific margins is better simulated in REMI with slower remineralization
rates in the surface ocean. Yet, no simulation reproduces the reconstructed low CaCO$_3$ content in the North Atlantic and high
CaCO$_3$ content in the equatorial Pacific. The model has been tuned to the pre-industrial CaCO$_3$ distribution, however, in our
study late Holocene CaCO$_3$ contents are the result of almost 800 kyr of transient simulation. The model-data match for the
Holocene is relatively good for simulations with small sediment perturbations during the glacial cycle, especially simulations
BASE, KGAS, SOWI, AERO and LAND, but also PO4 which does not produce a CaCO$_3$ burial spike during deglaciations. In
simulations REMI and PIPO, however, CaCO$_3$ burial rates have not yet stabilised by the end of the simulations after the large
CaCO$_3$ burial event during the last deglaciation, resulting in much higher sedimentary CaCO$_3$ contents in the Holocene than re-
constructed. Simulation CO2T, on the other side, has less sedimentary CaCO$_3$ content during the Holocene than reconstructed.
This is the result of strong dissolution due to forced alkalinity removal from the ocean during deglaciations.





**Figure 12.** Evolution of $[CO_3^{2-}]$ in the tropical deep Pacific as simulated in simulations BASE, PHYS, BGC and ALL and reconstructed by Qin et al. (2018). The results of the other forcings are shown in Fig. S27.

Comparing our simulation results to reconstructed $[CO_3^{2-}]$ changes is another test of how realistic the simulated carbonate
495    system changes during glacial cycles are. Kerr et al. (2017) found a repeated pattern of low benthic $[CO_3^{2-}]$ in the tropical
Pacific and Indian Ocean during interglacials and high $[CO_3^{2-}]$ during glacials (difference of 20-55 mmol/m$^3$) across the last
500 kyrs. Qin et al. (2018) found that the same pattern extended over the last 700 kyrs. In our simulations with purely physical
forcings (BASE, KGAS, SOWI, AERO, PHYS) and PO4, $[CO_3^{2-}]$ is almost constant over glacial cycles (Fig. 12, S27). In
PIPO, LAND, REMI and CO2T the simulated $[CO_3^{2-}]$ changes are closer to the reconstructed magnitude. $[CO_3^{2-}]$ maxima
500    during glacial maxima are present in all simulations with interactive sediments, the glacial-interglacial difference ranging from





a few mmol/m$^3$ to 110 mmol/m$^3$, and is caused by invasion of $CO_2$ into the ocean and weathering-burial imbalances due to changes of the carbonate export flux and sediment dissolution (Fig. 12 and S27). In simulations without interactive sediments, this glacial $[CO_3^{2-}]$ maximum is not simulated in PIPO, REMI and LAND.

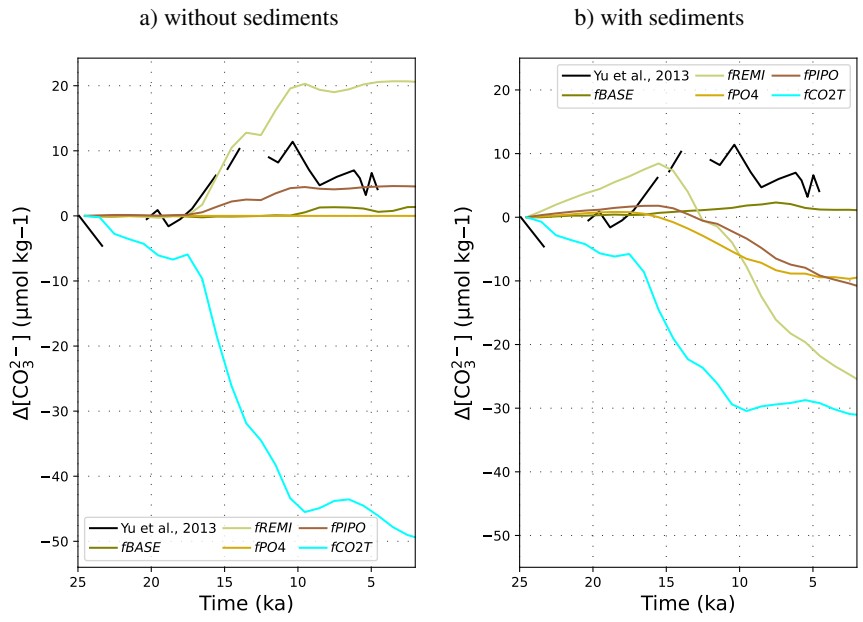

**Figure 13.** Comparison of simulated and reconstructed (Yu et al., 2013) $[CO_3^{2-}]$ changes in the deep (3500 m) West Equatorial Pacific with and without dynamic sediments.

During the last deglaciation, reconstructions of $[CO_3^{2-}]$ of the deep equatorial Pacific showed only small variability, which is solely simulated with purely physical forcings or in simulations without interactive sediments (Fig. 13). The offset from the reconstructed glacial values and the large changes during the deglaciation suggest that all applied changes to nutrient supply (PO4) and remineralization (REMI) cause too large weathering-burial imbalances of alkalinity. While net alkalinity removal from the ocean during deglaciation (e.g. by shallow deposition, coral reef growth or strongly reduced weathering inputs) can cause widespread dissolution which might diminish excess accumulation of $CaCO_3$ (see above), it cannot offset the strong $[CO_3^{2-}]$ decrease which is not part of the reconstructions but caused by the applied nutrient and remineralization forcings.

Insterestingly, during MIS13-MIS11 and the Mid-Brunhes transition, reconstructed $[CO_3^{2-}]$ changes in the deep equatorial Pacific were much larger than during the last glacial cycle (Fig. 12). Such variability of $[CO_3^{2-}]$ amplitude between glacial cycles is not reproduced by any of our simulations, but simulations CO2T and REMI, which produced larger-than-reconstructed $[CO_3^{2-}]$ changes over the last glacial cycle produced $[CO_3^{2-}]$ changes more similar to those reconstruced for MIS13-MIS11.





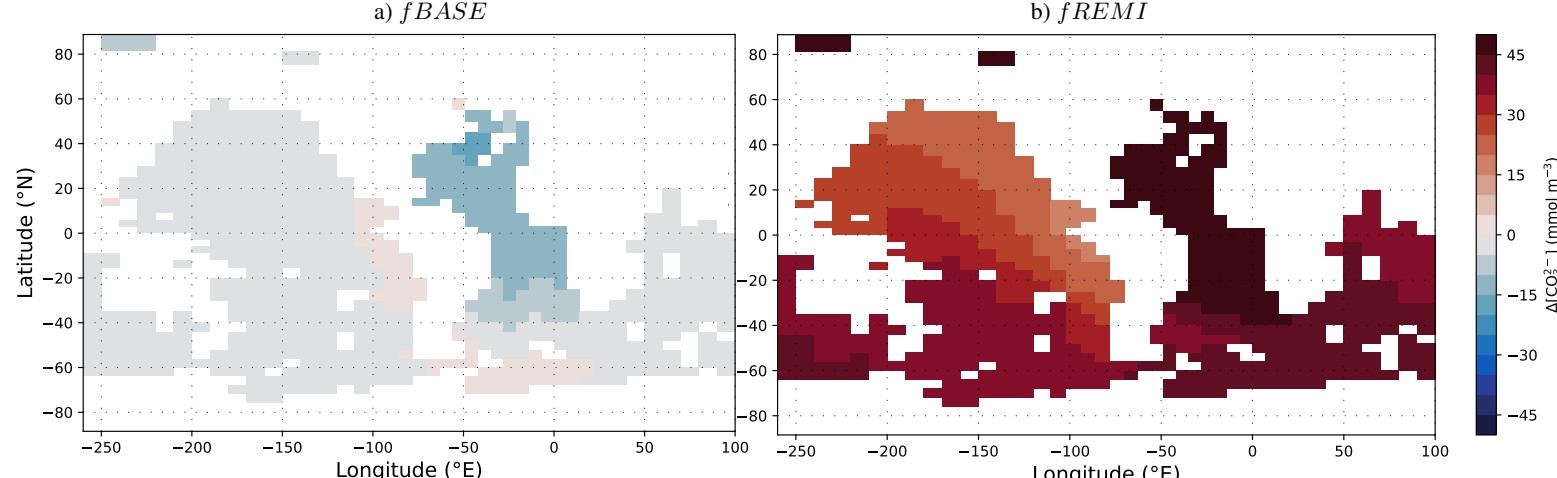

**Figure 14.** Selected factorial effects on simulated LGM-PI differences in deep $CO_3^{2-}$ (3500 m depth).

While the reconstructed deep ocean $[CO_3^{2-}]$ reservoir in the Pacific was relatively stable over the last deglaciation, a large $[CO_3^{2-}]$ increase was reconstructed for the deep Atlantic Qin et al. (2018); Yu et al. (2019). The different sensitivities of deep ocean $CO_3^{2-}$ in the two basins is also apparent in all of our simulations (see examples in Fig. 14 and S28) and is the result of larger circulation and productivity changes in the Atlantic than Pacific. However, circulation changes produce lower $[CO_3^{2-}]$ in the deep subpolar North Atlantic during the LGM, while reconstructions suggest higher $[CO_3^{2-}]$ (Yu et al., 2019). Higher deep Atlantic $[CO_3^{2-}]$ at the LGM requires increased nutrient supply (PO4), deeper remineralization (REMI) or a net alkalinity input (CO2T). These patterns appear with and without dynamic sediments in our simulations. Sediments mostly affect the amplitude and temporal evolution of deep $[CO_3^{2-}]$ changes, not their spatial pattern (not shown).

### 4.7 Imbalances of the geologic carbon cycle

The previous model-data comparison showed that weathering-burial imbalances increase the potential of many forcings to draw down $CO_2$ during glacial periods and buffer carbon cycle changes during deglaciations. We identified two processes which are most effective at raising atmospheric $CO_2$ during deglaciations: Alkalinity removal and organic carbon remineralization. In simulations PIPO, REMI, BGC and ALL the combination of high alkalinity at the end of glacial phases and increased $CaCO_3$ export production during deglaciation causes large transient $CaCO_3$ deposition events in the open ocean (Fig. S12) which remove the excess glacial alkalinity and thus drive a large but slow continuous $CO_2$ rise compared to the reconstruction. The marine DIC and alkalinity that built up over the previous glacial phase are too large to be removed instantly, and the resulting large deposition of $CaCO_3$ during the deglaciation persists far into the interglacial. In consequence, these simulations produce poorer model-data matches for Holocene $CaCO_3$ and marine $\delta^{13}C$ distributions than the initial PI spinup (see Fig. S36,S39). We also showed that the resulting $\delta^{13}C$ and $[CO_3^{2-}]$ changes in the deep Pacific are not consistent with reconstructions. In CO2T, alkalinity removal is forced to reproduce the reconstructed atmospheric $CO_2$ record, and can serve to study the effect



of alkalinity removal through means other than deep ocean $CaCO_3$ burial (e.g. shallow deposition, coral reef growth, reduced terrestrial input) on other proxy systems. In this simulation, deep ocean $CaCO_3$ dissolution occurs during the deglaciation and marine DIC increases, resulting in more proxy-consistent $\delta^{13}C$ changes in the deep Pacific but still a large mismatch in $[CO_3^{2-}]$. Alternatively, in simulation PO4, the deglacial $CO_2$ rise is largely caused by a reduction in export production and increased remineralization of sedimentary organic matter which accumulated during the previous glacial period. The resulting

$CO_2$ increase is of similar amplitude as that in simulation REMI but happens faster, more consistent with the reconstruction. In addition, deep Pacific $[CO_3^{2-}]$ is less perturbed by this process than in REMI or CO2T, yet deep ocean $\delta^{13}C$ is shifted in the wrong direction.

There are several ways, however, in which the amplitude or regional pattern of the simulated changes might be biased by our experiment design. Firstly, by design our forcings are smooth in time and spatially uniform, which is a stark simplification.

For example, the PO4 forcing ties nutrient supply to the $\delta^{18}O$ record. The correlation between dust (iron source to the open ocean) concentrations in the EPICA Dome C ice core and benthic $\delta^{18}O$ is of first order only and varies over the glacial cycle (Winckler et al., 2008). Several macro- and micronutrients were likely supplied to varying parts of the glacial ocean (Broecker, 1982b; Martin, 1990; Pollock, 1997; Deutsch et al., 2004) and while dust flux changes seem to correlate globally (Kukla et al., 1990; Winckler et al., 2008), the timings and rates of other nutrient fluxes might in reality have varied temporally and spatially.

Similarly, our other forcings might change more slowly over the deglaciation than the real processes they represent.

Another simplification in our experiment design is that the majority of our simulations assume temporally constant carbon inputs although in reality these fluxes are climate sensitive (Munhoven, 2002). It is unlikely that removing this simplification could resolve the bias in simulated global carbon fluxes and reservoir size changes because it is estimated that global weathering rate changes during glacial cycles were small despite large local variability, possibly because they canceled out in the global

mean (Jones et al., 2002; Von Blanckenburg et al., 2015; Frings, 2019; Börker et al., 2020). It was estimated that glacial-interglacial weathering flux changes altered atmospheric $CO_2$ by a maximum of 20 ppm (Köhler and Munhoven, 2020). Yet, the resulting $\delta^{13}C$ perturbation could be larger because a global balance in carbon flux changes does not imply a balance in carbon isotope fluxes (Jeltsch-Thömmes and Joos, 2023). Additionally, there might have been non-linear changes in isotopic input fluxes during the simulated time period.

Finally, the imbalance between weathering and burial fluxes is also shaped by the sedimentation rate. In our simulations, sedimentation rates vary due to changes in biogenic export, yet accumulation of non-biogenic material remained constant. This omission, however, does not seem to be the source of large errors. In a sensitivity experiment, we prescribed step-wise 30% increases and decreases of the non-biogenic flux in the PI steady state, which had only marginal effects on atmospheric $CO_2$, DIC and sedimentary accumulation of biogenic particles (Fig. S25).

Regardless of which process or combination of processes we simulated, no simulation reached a weathering-burial equilibrium by the late Holocene. It seems likely that carbon burial in marine sediments was still adjusting to interglacial conditions at the beginning of the industrial revolution. This is also indicated by the reconstructed changes in global sedimentary burial rates over the last glacial cycle, which suggest that burial fluxes changed throughout the Eemian and the current interglacial and only find apparent balances during the longer glacial phases (Fig S2, Cartapanis et al., 2016, 2018).



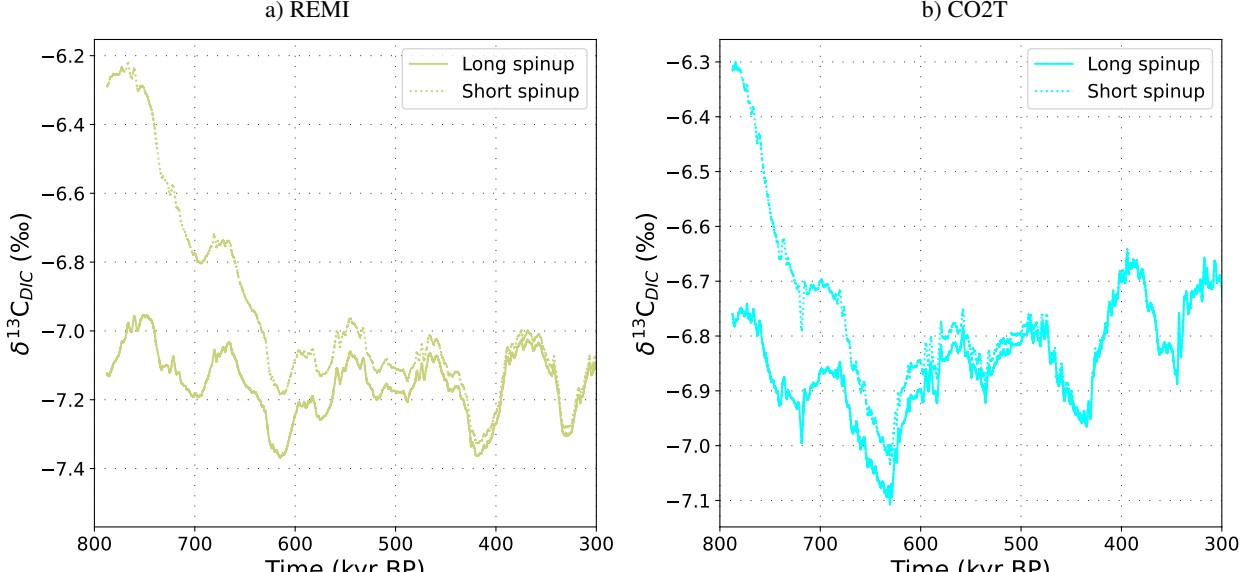

**Figure 15.** Comparison of simulated atmospheric $\delta^{13}$C in simulations REMI and CO2T when started from a 'short spinup', i.e. a 50 kyr PI spinup followed by a 2 kyr adjustment to MIS19 conditions, and a 'long spinup', i.e. the short spinup plus 215 kyr of transiently simulated MIS19-MIS15.

On a technical note, the long adjustment timescale in the geologic carbon cycle also presented an initialization problem, especially for carbon isotopes (Jeltsch-Thömmes and Joos, 2023). We started our experiments from MIS19, which was a colder interglacial than the the Holocene, and Holocene conditions were not reached during the lukewarm interglacials of the first 400 kyr of the simulations. In simulations with interactive sediments, the initial imbalance between weathering inputs derived from the pre-industrial spin-up and burial fluxes adjusting to the colder lukewarm interglacials and glacial states caused

$\delta^{13}$C drifts during the first glacial cycles (Fig. 15). Consequently, the simulated glacial-interglacial $\delta^{13}$C signal over this period is altered by the long-term adjustment of the geologic carbon cycle. We addressed this issue by transiently simulating two full glacial cycles before starting the experiments. The magnitude of the initial imbalance in the geologic carbon cycle, and hence isotopic drift, depended on the simulated forcing and was largest in simulations REMI, PIPO and CO2T. Importantly, the drift is a result of perturbing the sediment-weathering balance. The drift can therefore not be corrected for with a control simulation

without forcing, because it only appears in the perturbed system. Instead, to avoid a drift, the experiment needs to start from an isotopically balanced geologic carbon cycle, which most commonly will require a long spin-up with a fully-coupled, open system, ideally over several glacial cycles especially when simulating large changes of the biological pump or marine carbonate system. We suggest that the size of the transient imbalance of the geologic carbon cycle, and thus the length of the required spin-up, could be minimized by balancing the geologic carbon cycle not for an interglacial state but the mean burial fluxes over

a full glacial cycle.





## 5 Discussion

It is well established that cooling and circulation changes altered sea-air gas exchange and increased deep ocean carbon storage by isolating it from the surface during glacial phases (e.g. Brovkin et al., 2007). Combined, these effects contribute to changes in atmospheric $CO_2$ in our simulations that are comparable to Brovkin et al. (2012) (26 ppm compared to 30 ppm). Our

simulations also capture some transient features of of sea-air gas exchange variability over the last glacial cycle that have been simulated by Brovkin et al. (2012). For example, we found a brief atmospheric $CO_2$ increase between 110 and 90 ka due to circulation changes, but only in simulations in which AMOC has already weakened substantially by this point, i.e. those with feedbacks that amplify orbitally-driven cooling (e.g. increased aerosol load). Brovkin et al. (2012) showed that a higher vertical diffusivity in the Southern Ocean increases the sensitivity of the sea-air $CO_2$ flux to sea ice changes. Raising

the vertical diffusivity of ice-free areas by an order of magnitude in their model resulted in an additional $CO_2$ draw-down of 10 ppm during inception due to Southern Ocean cooling. We did not test the effect of such change in our simulations, but reducing the piston velocity in the Southern Ocean by up to 40 % in simulation KGAS caused an additional $CO_2$ draw-down of only ∼2 ppm. Isolating the deep Pacific through increased Southern Ocean wind forcing (simulation SOWI) caused a glacial $CO_2$ decline by ∼13 ppm, the biggest $CO_2$ draw-down on top of the effect orbital cooling of any isolated physical forcing that we

tested. Tschumi et al. (2011) showed that this effect also has the potential to cause larger $CO_2$ draw-down with sedimentary amplification than simulated here. The idealised, strong reductions in wind speeds over the Southern Ocean prescribed by Tschumi et al. (2011) as a tuning knob for producing old deep ocean waters are unrealistic, but other processes could have contributed to increased isolation of the deep Pacific. Bouttes et al. (2011) showed that during glacial stages enhanced brine rejection during sea ice formation can isolate abyssal waters and cause atmospheric $CO_2$ and $\delta^{13}C$ changes that are similar

to those reconstructed. Enhanced brine rejection could thus have provided an additional physical process that increased the glacial marine carbon storage. The strength of this process, however, is only poorly constrained, and Ganopolski and Brovkin (2017) showed that, at a sufficient strength to significantly affect deep ocean carbon storage, this process creates bigger $\Delta^{14}C$ anomalies in the deep ocean than reconstructed. Following (Menviel et al., 2011), they also argue that the timing of increased sea ice formation and atmospheric $CO_2$ changes during the last deglaciation (Roberts et al., 2016) are not entirely consistent

with a strong control of brine formation rates on marine carbon storage.

In further agreement with other modelling studies, e.g. Buchanan et al. (2016) and Morée et al. (2021), we find that changing the efficiency of the biological pumps is an efficient mechanism to achieve glacial-interglacial atmospheric $CO_2$ changes similar to those reconstructed from ice cores and to reproduce the reconstructed decreased preformed nutrient concentrations during glacial phases (Ito and Follows, 2005; Vollmer et al., 2022). However, because of its large effects on deep Pacific $[CO_3^{2-}]$ and

$CaCO_3$ accumulation during deglaciation it was unlikely the dominant carbon cycle change over the last glacial cycle. Several biochemical forcings in addition to reduced remineralization in the surface ocean (REMI) substantially reduce atmospheric $CO_2$, especially increased nutrient and alkalinity supply (PO4, CO2T). The low glacial DIC in the former simulations also shows that $CO_2$ removal from the atmosphere in theory does not need to result in increased DIC in the ocean. Instead, these



biochemical forcings cause sedimentary changes that can store large amounts of carbon in inorganic and organic sedimentary
matter.

A relevant role of marine sediments, particularly sedimentary $CaCO_3$, in glacial-interglacial carbon cycle dynamics has long
been discussed (e.g. Broecker, 1982b; Broecker and Peng, 1987; Opdyke and Walker, 1992; Archer and Maier-Reimer, 1994;
Raven and Falkowski, 1999) and shown in numerical experiments of differing physical and biochemical complexities (Ridgwell
et al., 2003; Joos et al., 2004; Tschumi et al., 2011; Menviel et al., 2012; Roth et al., 2014; Wallmann et al., 2016; Ganopolski
and Brovkin, 2017; Jeltsch-Thömmes et al., 2019; Köhler and Munhoven, 2020; Stein et al., 2020; Kobayashi et al., 2021). In
agreement with other studies (e.g. Ganopolski and Brovkin, 2017; Köhler and Munhoven, 2020), we find that changing marine
alkalinity can produce large $CO_2$ changes. Organic carbon storage is less often considered in modelling studies, although it
also showed significant changes across the last glacial cycle (Cartapanis et al., 2016). Out of the processes we tested, increased
nutrient supply during glacial phases (simulation PO4) produces temporal and regional organic carbon deposition changes that
were most consistent with the reconstructions. In this simulation, marine sediments turn into a strong carbon sink during cold
phases. The simulated increased organic carbon deposition during glacial phases reproduces the reconstructed long-term trends
in atmospheric and surface ocean $\delta^{13}C$ during glacials, but fails to simulate the reconstructed deep ocean $\delta^{13}C$ changes in the
Pacific and Atlantic. Thus, while sedimentary organic carbon burial could have provided a carbon sink during glacial phases,
it must have been small enough or be compensated by other isotopic forcings to not influence benthic $\delta^{13}C$. Interestingly, our
simulations with increased organic carbon burial during glacial phases show that some of the deposited organic carbon can be
returned to the ocean during deglaciations with a large potential to contribute to a fast post-glacial rise in atmospheric $CO_2$. In
addition to carbon, nutrients are also removed from the ocean when organic matter is buried (Roth et al., 2014). Tschumi et al.
(2011) demonstrated in their steady state experiments that increased organic nutrient burial reduces $CaCO_3$ export, increases
surface alkalinity and amplifies the $CO_2$ drawdown caused by the removal of organic carbon. In simulation REMI, this process
operates transiently. Given the reconstructed increased organic carbon burial rates during glacial maxima, this could have been
a relevant process over the last glacial cycles, though it might have been reduced in its efficiency due to reductions in the
PIC:POC of export production during glacial phases (Dymond and Lyle, 1985; Sigman and Boyle, 2000). Finally, sedimentary
organic carbon oxidation can also regulate marine alkalinity by affecting sedimentary $CaCO_3$ dissolution (Emerson and Bender,
1981; Sigman and Boyle, 2000). The only simulation in our set which shows increased alkalinity fluxes out of the sediments
during glacial phases is LAND, in which terrestrial carbon release causes sedimentary dissolution. In all others, especially those
which reproduce the reconstructed increase of organic carbon burial during glacial maxima, sedimentary remineralization rates
decrease during glacial times. Thus, remineralization-driven sedimentary dissolution does not contribute to the glacial $CO_2$
draw-down in our simulations.

In all of our simulations with interactive sediments, the DIC change over a glacial cycles is larger than the simultaneous
atmospheric $CO_2$ perturbation because of changes in carbon reservoirs in sediments and weathering-burial imbalances. In
some simulations, large DIC changes are produced by big sedimentary changes during glacials which cannot be restored
during deglaciations and cause interglacial carbonate preservation patterns that are not consistent with observations. While this
suggests that such a scenario is unrealistic, it does not generically preclude the possibility of large transient weathering-burial



imbalances. Testing a wider range of forcing magnitudes and combinations with the same model but different set-up, Morée et al. (2021) found a larger DIC change between the pre-industrial and LGM than simulated here (3900±550 GtC compared to a maximum of 1100±300 GtC in Fig. 5) that is consistent with carbonate system proxy constraints. Combinations of the tested forcings thus allow for larger transient weathering-burial imbalances than produced by our simulation ensemble that can still be reconciled with carbonate system proxies.

Regardless of the dominant carbon cycle change, like previous studies we find that $CO_2$ degassing from the ocean persisted throughout deglaciations and into interglacials (e.g. Brovkin et al., 2012), and that the carbon cycle does not reach a new equilibrium before the next glacial inception. In our simulations AMOC hysteresis, sedimentary changes and, delayed temperature responses e.g. due to ice shields (by scaling most forcings to the $\delta^{18}O$ record), introduce memory effects which buffer deglacial carbon cycle reorganizations and cause continued $CO_2$ rise throughout interglacials. For example, in PO4, BGC and ALL, the simulations which best align with the reconstructed glacial-interglacial organic carbon burial changes, not all glacial organic matter is remineralised and carbonate dissolution continued throughout the interglacials. It has long been suggested that sedimentary imbalances contributed to the observed interglacial sedimentary changes and $CO_2$ rises (Broecker et al., 1999; Ridgwell et al., 2003; Joos et al., 2004; Broecker and Stocker, 2006; Elsig et al., 2009; Menviel et al., 2012; Brovkin et al., 2016).

## 6 Conclusions

In response to different simulated carbon cycle forcings over the repeated glacial-interglacial cycles of the past 780 kyr in the Bern3D model, we found large sedimentary changes which substantially alter marine carbon and nutrient concentrations and spatial distributions. Our simulations show that biochemical forcings are required to perturb the sediments sufficiently to reproduce reconstructed burial changes and $CO_3^{2-}$ variations, yet other processes must have operated to reduce the buffering impact of this perturbation on the deglacial carbon cycle re-organization in order to match the speed of the associated carbon release. Our set of factorial simulations further leads to the following conclusions:

Firstly, ocean-sediment interactions and related weathering-burial imbalances, including fluxes of nutrients, alkalinity, organic and inorganic carbon, tend to amplify glacial-interglacial $CO_2$ change.

Secondly, changes in the ocean inventory of DIC do not scale with changes in atmospheric $CO_2$ but are strongly influenced by sediment fluxes. For example, the potential addition of phosphate from exposed continental shelves causes not only a decrease in atmospheric $CO_2$ but also in ocean carbon inventory. Factorial simulations yield an average DIC change between -1340 to +1400 GtC and $CO_2$ changes between -45 and 80 ppm (-96 and 180 GtC) over the last five deglaciations in response to individual prescribed physical and biogeochemical forcings. This suggests that approaches utilizing the relationship between radiocarbon and DIC from modern data to reconstruct the ocean's glacial DIC inventory and the postulated corresponding $CO_2$ change from glacial radiocarbon data may be biased.

Thirdly, ocean-sediment interactions strongly impact the evolution of important carbon cycle parameters such as $\delta^{13}C(DIC)$ and $\delta^{13}C_{CO_2}$, $CO_3^{--}$, export production, $CaCO_3$ and POM burial fluxes, preformed and remineralized nutrient concentrations,



and oxygen. The interpretation of the proxy records without consideration of weathering-burial imbalances and ocean-sediment interactions may lead to erroneous conclusions.

We also showed that the long timescales of ocean-sediment interactions and the weathering-burial cycle pose substantial challenges for model spin ups because imbalances in the geologic carbon cycle can cause isotopic drifts at the beginning of simulations and which are not present in a control run. Depending on the initial isotopic imbalance, it takes up to 200 kyr for the drift to subside and the signal of the applied forcing to dominate the simulated transient $\delta^{13}$C changes.

*Data availability.* All simulation output necessary to produce the figures in this manuscript are available at https://doi.org/10.5281/zenodo.11385608

*Author contributions.* FP and AJT designed the simulations. AJT ran the simulations. MA processed the model output and drafted the
manuscript. MA, AJT, FP, FJ and TFS interpreted the results and edited the manuscript.

*Competing interests.* The authors declare that they have no conflict of interest.

*Acknowledgements.* This research has been supported by the Schweizerischer Nationalfonds zur Förderung der Wissenschaftlichen Forschung (grant nos. 200020-200511 and 200020-200492) and Horizon 2020 (grant nos. 101023443 and 40 820970).



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
