# Peer review of "Sediment fluxes dominate glacial-interglacial changes in ocean carbon inventory: results from factorial simulations over the past 780,000 years"

_EGUsphere, 2024_

## Referee Comment (RC1)

Review of "Sediment fluxes dominate glacial-interglacial changes in ocean carbon inventory: results from factorial simulations over the past 780,000 years" by Adloff et al.

In this study, the authors use the Earth System Model of Intermediate Complexity Bern3D to simulate the carbon cycle dynamics of the last 800,000 years. They present multiple simulations, with or without interactive sediments, and with a diversity of simplified physical and/or biogeochemical forcings. Their results are evaluated using multiple proxy reconstructions. Thanks to this approach, they are able to assess to some extent the "potential and plausibility of major contributions" (L197) of several processes to the observed glacial-interglacial atmospheric $CO_2$ change, and notably the importance of the sediment fluxes. As such, it should be recognized that this study tackles an outstanding scientific question, and provides and analyzes long transient simulations (albeit with simplified forcings) of coupled climate-carbon variations during glacial-interglacial cycles, which are infamously difficult to simulate, especially considering an open carbon system.
I find the manuscript structured and well-written, although some clarifications are occasionnaly needed. Hopefully, some of my suggestions in the specific comments will be of use to the authors in this respect. Still, I would say that the main flaw of this ambitious study is that it is attempting too many things at once. A sign of that is that the main text is about 36 pages long, with additional analysis being provided in the first 14 pages of the Supplementary (before extra figures are included). As a result, and despite the clear language, it is often a dense and long read (some would even say tedious), with the reader having to navigate the large amount of information presented in the main text, in the main figures, in the Supplementary... which results from the great number of simulations and analyzed variables presented. In particular, it seems to me that there are instances in which the information contained in the Supplementary is critical to understand the simulations results, whereas some sections in the main text do not seem very central to the main messages. I would therefore encourage a reconsideration of the elements presented, and a reselection of some in the main text from/to the Supplementary (or a removal altogether), to a greater or lesser degree.  I am providing below a number of suggestions to hopefully help guide the authors in this direction, but other choices from the authors could be welcomed. Considering the manuscript length, this review also ended up rather lengthy.
In summary, I therefore consider that this palaeo-modelling effort is well worthy of publication and that it will be of interest to the Climate of the Past readership, and may especially pave the way for modellers using less computationnaly-efficient models. To my knowledge, it is the first to tackle coupled climate-carbon simulations of multiple glacial-interglacial cycles including an interactive sediment component, and examining the effects of various sensitivity tests using multiproxy analysis. Still, some of the presentation choices are not ideal and are shaping this study into a long, dense and comprehensive paper. I believe that some trimming (ranging from marginal changes to more substantial reorganization) may lead to a more focused, concise and therefore impactful paper.

**General comments**

**1. Making all connections explicit in order to spell out the knowledge gap, and linking it to the methodological choices**
The scientific reasoning is most of the time explicit to the reader, but there are a few instances in which all of these connections could be improved. I am pointing them out here.

- L5 : "Yet, it is unclear how much they affected carbon cycling during transient changes of repeated glacial cycles, and what role burial and release of sedimentary organic and inorganic carbon and nutrients played.". I believe the knowledge gap could be more clearly spelled out, including why you are considering specifically repeating cycles, and the role of sediments. A more explicit link to the

chosen scientific approach (justifying the choice of 1/ using "various" forcings, 2/ for simulations of the "last 780 kyr") could then be made.

- L45-49 : I think that the knowledge gap could be more clearly spelt out. As it is now constructed, the L45-47 sentence doesn't really justify the case for including sediments. Maybe you could explain that the increased marine carbon storage in closed atmosphere-ocean system is not sufficient to simulate the low glacial atmospheric $CO_2$ without carbonate compensation (e.g. Kobayashi et al., 2021), and cannot be of the right amplitude without the inclusion of land processes as well, as carbon is lost in the terrestrial vegetation. Without clearly showing the limitations of previous studies, the "little is know" and "it has long been assumed" formulations are not exactly impactful as a knowledge gap.

- L135-142 and L143-157: The choice of parameters to modify in the "simplified forcings" (wind stress, etc.) is not explicitly linked to the "glacial-interglacial carbon cycle drivers" described in the introduction. The "why" behind all of these simulations is not clear. I think some sentences should be added to make the connection (e.g. "as a modified wind stress significantly influences the deep ocean circulation in our model") with the mentioned drivers or to other relevant purpose (e.g. Southern Hemisphere westerly winds seems to have varied over glacial-interglacial cycles according to Gray et al., 2023). Alternatively, to keep it short, you could add a column "tested drivers" (e.g. ocean circulation) in Table 1. Also, in L139-142, I feel like it could be (1) mentioned which forcing was expected to behave like which delta and why; (2) acknowledged that this is a strong assumption, not allowing for threshold effects for e.g. nutrients supply or terrestrial carbon release. Finally, it could be acknowledged that your set of simulations do not tackle some of the drivers mentioned in the literature (e.g. sea ice). Likewise, it could be recognized that you are testing out phosphate input and not iron fertilisation effects because – I assume – your model does not include a representation of the iron cycle.

**2. Reorganizing elements from/to the Supplementary to make for a more standalone, understandable, and trimmed down main text**

- Additional sensitivity tests in L165-169: There are already a great number of simulations. I feel like these additional ones don't need to be part of the main text as they do not seem to closely relate to your main scientific question and message. Or do they?

- L193-196: It is odd to start the results section by refering to the extensive material presenting in the SI (which almost feels like a companion paper).
One sign is that the reader may struggle to understand the effects of individual forcings (see specific comments on Table 2, Fig. 3…) before seeing the results of individual simulations. In a way, to properly understand the paper, reading the first 14 pages of the Supplementary is necessary. This is where the processes explaining the changes in carbon storage in the different reservoirs are described. Yet it could be argued that at least one of the ΔC graphs in the SI should be part of the main text. Understanding the evolution of the distribution of carbon between the different pools seems important. But it seems that the weathering-burial imbalances are causing some issues at the end of transient runs (esp. SOWI, p.16), are they not?
Another sign is that despite the scientific question on the role of sediments, "a detailed analysis of the sedimentary changes" is part of the SI. I believe that since most readers won't read the SI, the authors should strive towards a more standalone main text.

- Please consider briefly introducing the proxy reconstructions used in this study in the start of the results sections or in the methods. At the start of the results section when they are already used right off the bat, the reader only knows that many are "available" (L205) and that some were "selected" (L207),

with has no idea how these reconstructions were made, if they are robust, how they can be compared to the model results (thanks to the isotope-enabled feature), and on which basis some were apparently selected…

- L230: The first mention of a figure in SI is numbered S17. We jump to S22 in L234. The numbering of the figures in SI therefore do not seem consistent. This doesn't help the back-and-forth between the main text and the SI which is too often required to understand the main text properly. Despite the amount of materials, some of this back-and-forth could be prevented. For example, for L230-234 the authors could consider only relying on Fig. 3b, or a modified Fig. 3b, making the main text standalone by removing nuances which can only be explained by the extra figures. These few sentences could be kept in the SI, along with Fig. S17 and S18, in a numbered section addressing a specific question (e.g. "What are the lags between CO2 and the forcings?"), with the main text only commenting "lags observed in Fig 1 and 3b depend on the choice of forcings and are more fully explained in section S1 of the SI". Alternatively, the main information of Fig. S17 could be shown within Fig. 1, as a zoom on the last termination.

- Figure 4 and L240-251: It is unclear to me what this whole section on the MBE actually brings to the table except for the message in L250-251. In the introduction, the MBE wasn't exactly part of the main scientific question.  While this is interesting, describing the lags between CO2 and the forcing in different sensitivity tests (previous comment), or the maximum amplitude of the interglacials before/after the MBE across different sensitivity tests seem like details when the main text has yet to explain which processes are behind the glacial-interglacial variations in these simulations. I would argue that this whole part could be moved to the SI (as a "please refer to section S2 "How is the MBE simulated?" in the SI for more.")

- L251-261: Some of the processes are here explained. However, they are just described and not *shown*. No figure is refered. Please consider whether some sort of modified Fig. S4 (incl. evolution curves for atmospheric CO2, DIC, alkalinity, CaCO3 deposition, POC deposition, and also the weathering-burial imbalance…) could be of use in the main text for the reader to properly understand the simulations. I was also curious to see organic carbon variations.

- Section 4.2: Fig. 5 is first shown but not described, only briefly refered to L298. The text goes on to explain the importance of the Southern Ocean (not shown, and we are wondering why we are starting by regional changes), then surface pH (shown, but in SI). Why are carbonate shifts described via pH changes and not alkalinity? So far, the link with DIC has not been made explicit. Perhaps the authors should consider transforming Figure 5a into a DIC-ALK diagram. Also, "The spatial patters of marine carbon uptake and release are thus driven by combinations of physical and chemical processes" (L282) are never shown, making the sentence feel out of place and unproven.

- Fig. 14: Is it really essential to include in the main text?

- Please note that L524-542 somehow already feels like a conclusion, and L543-559 like a discussion. Perhaps this should be acknowledged in the subsection title.

- L560-564: Is it really essential to include in the main text?

- Page 32, incl. Fig. 15: Is it really essential to include in the main text?

- L593-598: Is this discussion relevant? Increasing the vertical diffusivity is not what you did, and KGAS has only a minor effect.

- Fig S23-S24 + Fig S36-39: Considering your outcomes of Section 4.2 and the fact the REMI seems to strongly imprint on the direction of the nutrient changes during deglaciation in BGC and ALL, and on the model-data agreement for delta13C, I am wondering whether you have considered a BGC simulation excluding REMI?

**3. A couple of reccurent inadequate terms**

- In L2 and other instances, it is unclear why the reactive layers of sediments are considered "ocean sediments" whereas the inert layers are part of the "lithosphere". From a geological point of view, this is disputable. Other instances include L48-49, where "marine sediments and the lithosphere" are mentioned as a unique "fourth reservoir", which makes it confusing why this distinction is introduced in the first place.

- L171 and many other instances: The term "simulated process" is often used to designate the simplified forcings prescribed in the simulation. I would be careful with this term. Processes may refer to the mechanisms of the carbon cycle which contribute to explaining glacial-interglacial variations (e.g. changes in water mass distribution, carbonate compensation…). We can achieve process-understanding (to some extent at least) by examining the model's response to different forcings. But prescribed parameters are hardly processes by themselves. Same for e.g. L202 "We investigate the isolated processes."

- L192 and many other instances: I believe that the term "biochemical" refers to molecular biology and that "biogeochemical" should be used.

**Specific comments**

    **1. Abstract**

- L3 : I think that carbon transfers don't directly result from different "sensitivities" of the carbon reservoirs to the forcing, but result from the different responses to the forcing of the carbon reservoirs, which depend on their sensitivities ?

- L3 : It is unclear what "many of which" refers to.

- L10 : It is unclear what "associated" refers to.

- L10 : The importance of considering "isotopic shifts" is not introduced.

- L12-13 : "In our simulations the ocean inventory changed by 200-1400 GtC and the atmospheric inventory by 1-150 GtC over the last deglaciation." It is unclear which conclusion the reader should draw from these results.

- L18 : It is unclear what "needs to be considered" refers to. Does this refer to isotopic drifts?

    **2. Introduction**

- Introduction : Please consider mentioning the permafrost as a carbon reservoir of relevance.

- L24 : It is a bit unclear which variable is delayed with respect to the others.

- L39 : Although the "enhanced stratification due to brine rejection" is classified in "other physical processes", stratification closely relates to "changes in ocean circulation" (L34).

- L50 : I suggest changing "on continental shelves" by "considering continental shelves". "On" is slightly odd as the sentence refers to altered "seawater carbonate chemistry", but no seawater remains "on" continental shelves when they are emerged.

- L61-63 : A lot of new notions are introduced here, which could be hard to follow for non-specialists. Please consider briefly explaining (1) why carbon isotopes are used in model simulations, (2) what is the burial-nutrient feedback, (3) what you mean by dynamic sedimentary adjustment, and (4) the relevance of imbalances in weathering-burial fluxes when considering an open carbon system. It should also be mentioned whether this imbalances in fluxes refer to carbon fluxes, or also include nutrients. I note that sentences in L65-68 and L69-73 are much more explicit, so it is just the first occurrence of theses processes which I find not easily understandable. A small reorganization of these sentences could therefore be considered. To this end, I should also mention that the L62-64 about equilibration time feels out of place in a paragraph whose main message is to underline the importance of including sediment for its impact on the carbon cycle.

- L74-75 : So far, the main messages that I am getting are "Sediment fluxes are important to consider / they are difficult to simulate". I think that the long equilibration time ( L62-64) and memory effects now mentioned could be used to explicitly make the case for transient simulations of multiple cycles. As of now, the connection with the proposed study (L90-93) is not clearly made, as the main message from paragraph L76-89 seems to be "no one has done it in this way before".

- L84 : Please consider detailing what "partially" exactly means, especially if it helps make the case for dynamic sediments.

- L90-96. All the originalities of the scientific approach of this study should be justified. Here we have :
    - transient → justified by "to avoid biases resulting from steady state assumptions", even though it is unclear what those biases exactly are. I think that it is unclear what the precision "so that all carbon stores at the beginning of the last glacial cycle are achieved dynamically rather than being prescribed" brings to the table. You haven't said that the last glacial cycle is of specific interest and why ?
    - 780 kyr long → not explicitly justified. Why several cycles? Why not shorter/longer?
    - factorial simulations → justified by "to understand how various processes…"
    - "with and without sediments" → justified by "to distinguish the role of interactive sediments"
    - isotope-enabled → it is not said in the introduction that one strength of this study is that it will provide a multiproxy analysis, even though it is part of your main results (e.g. L16).

    **3. Methods**

- It might be worth mentioning whether the model simulations include dynamic land-sea mask and bathymetry change (and hence varying ocean volume).

- L107-108: I don't understand. Is this to correct a specific bias?

- L118-119: Is this compensation done uniformly at the surface of the ocean? It should also be recognized that weathering isn't explicitly represented in the model, but that the loss to sediments is compensated by an input flux which we assume could represent a source from weathering fluxes brought to the ocean by rivers, thus allowing for an equilibrium of whole-ocean inventories. In this respect, perhaps it is more relevant to speak of "terrestrial solute supply" as in L124.

- L122: "in three stages, sequentially coupling all modules". Please consider briefly mentioning why. Also, it is never clearly said how many years the model was spun-up?

- L125-126: "is prescribed to balance loss to the lithosphere over 50 kyr". I am not sure I understand properly. When is the loss to sediment calculated? Is it a each step over a 50 kyr spin-up? Is it done once at the end of the 50 kyr? Why us the input flux kept constant thereafter and not diagnosed over each cycle or so?

- L133: It should be explained  why your approach includes the design of "simplified forcings".

- Table 1: (1) It is never explained why no PO4 simulation without sediments was performed. (2) It is not explained why the forcing values (e.g. -40%, -2.5 W/m², etc.) were chosen. (3) Note that the combination simulations PHY, BGC and ALL, as well as CO2T, have not been explained in the text before reading the Table. (4) It could be interesting for the reader to see the remineralization profiles of BASE compared to REMI.

- Fig. 1: I think that the legend should mention (1) the placement of each subpanel with indications (e.g. top, bottom, or panel numbers), (2) the abbreviations (e.g. RF), (3) that the grey background indicates MIS.

- L143-144: "to achieve an older glacial deep ocean". Please consider mentioning right away that "younger deep water masses" are in disagreement in proxy reconstructions. It is mentioned but later, in L146.

- L148-149: I understand the rationale behind KGAS but considering the results (as I remember them now), it doesn't seem like this simulation bring much to the table…

- L160: "adjust external alkalinity fluxes": is it the same as the "terrestrial solute supply" which is alternatively refered to as "weathering"? Also, I am curious as to how your model is able to effectively restore $CO_2$ variations: how are you prescribing alkalinity fluxes which results in the right $CO_2$ change, on a practical level?

- L164 and L165: Both of these sentences somehow feel out of place. Maybe you should explain the rationale behind a different $CO_2$ in the radiative code and its connection to sensitivity test 1.

- L169 : Why do drifts occur in these simulations specifically?

- L178 : I feel like the interest of these simulations to determine non-linearities could have been justified earlier, in the introduction. As of now, these simulations feel like an afterthought.

- L186-187 : This also feels out of place. It could have been mentioned earlier in the Methods.

**4. Results**

- Figure 2: (1) The gray shading is barely visible in print format. (2) Please consider explaining or simplifying the data legend (I don't understand "cubic spline Bereiter 2kyr cop"). (3) I don't understand why some simulations are selected to be represented here (e.g. PO4, REMI) and not others (BGC, ALL, etc.). (4) It could be nice to add in a value of the different LGM drawdowns on this figure (as text annotations). (5) Figure 1 would fit better in subsection 4.1 than in the start of the results section, which otherwise contains clarifications of the approach of the study (L196-205).

- L189: Note that differences "their timings" are not explained here but much later in the text.

- L191-192: This seems to be true for all biogeochemical forcings except for LANDC though.

- Table 2: Many comments and questions. (1) Legend could be more precise: "summary model-data comparison" → "Quantified metrics of the carbon cycle according to reconstructions and our set of simulations with sediments". (2) Legend indicates "global preformed nutrient concentration" but the variable in the column is the change of [PO4, *reg*]. (3) Why are you only using carbonate ions reconstructions for the deep Pacific and not other basins? (4) Why are dates and anomalies (esp. "PI – 18 ka")  inconsistent with each other and with the legend ("LGM-Holocene") and in L.209? Are the same dates and anomalies used for simulations? (5) Why are the effect of individual forcings presented before/instead of the overall model-data comparison of different simulations? E.g. I can read that fLAND has an opposite effect on pCO2, but I only have a vague idea of the LGM CO2 of the LAND simulation (by adding up fLAND + fBASE). As such, this table isn't exactly a "summary of data-consistency" but rather a "summary of the model response to forcings", isn't it?
Note that I understand the choice on focusing on the model response to a specific forcing, which is justified in L196-205, but it is surprising to start out the results with that, when the reader doesn't know yet which simulation(s) are consistent with proxy reconstructions.

- L212: "non-linearities [are] still small compared to the effect of individual process" → This is arguable, as they seem of the same order of magnitude as fBASE…

- L215-217: Should this be discussed in terms of processes (e.g. physical pump)?
L220: "other processes": other than what? One could for example argue that the right processes are simulated but that the amplitude of their effect is wrong.

- L220-222: Consider justifying the choice of assessing CO2, export production, biological pump, sedimentary fluxes and carbon isotopes. Can't an explicit link be made to what is said in the introduction? It is also unclear why we are examining these variables in this specific order.

- Figure 3. (1) Again, by showing the effect of individual forcings, the reader doesn't really see e.g. the good match of simulation CO2T with the ice core data , and you are forced to add in L225 that the sum of fCO2T and fBASE is what is line with the represented reconstructions. It feels like a first graph should first show the simulation results in terms of glacial-interglacial CO2 variations as well, so that the reader clearly realises that adding up all the physical forcings in PHY is still not enough, whereas combinations of some biogeochemical forcings may produce a signal of the right amplitude. (2) Legend should mention that this is the "effect of individual forcings on atmospheric CO2 changes". (3) Why is it only for the last 5 glacial cycles? (4) "selected factors" : on which basis are the effects of some factors represented and not others? e.g. the timing of LAND is mention in L228 but not

represented. (5) Note that the difference between the two green-yellowish are not very visible on print format.

- L226: "produce the most consistent CO2 difference": with Fig. 3 only showing factorial effects, the reader can't see that and only gets the impression that fREMI, fPO4 and fPIPO "produce the largest CO2 difference", none of them being remotely close to producing a 80-100 ppm drawdown by themselves. Same for "is not necessary produced in our idealised simulations with simplified forcings (Fig. 3b)": the reader cannot directly visualise this affirmation.

- L238-239: I think that the authors should elaborate on this. Why is this the case? Is this a good thing? What does this imply for models which do not consider imbalances? Alternatively, they could refer to a section in which this is discussed.

- L228: You could refer to Fig. 2.

- L259: You could consider briefly explaining why POC dissolution (and also increased alkalinity) raise atmospheric CO2.

- L261-262: "good match with various features". This is rather vague. Do you mean in terms of amplitude? Timing?

- L263-265: You could refer back -to the Fig. where this can be observed.

- L266 : Please elaborate. Are you implying "and so we can rely on multiproxy analysis to disentangle the different mechanisms in play"?

- L280 : "This mirrors". It it unclear what "this" refers to, with REMI described in L278 but PO4 described in L279.
- L283-285: Does this mean that over a long period, we have a rough equilibrium, with total fluxes at zero, and the carbon fluxes which goes in the Southern Ocean compensating the carbon outgassed elsewhere?

- L292: Doesn't the fact that weathering is kept constant also causes imbalances?

- L294: I don't understand the causality link ('consequently').

- L301-302: I am not sure I understand properly why the DIC/CO2 lag in simulations with sediments causes uncertainties in DIC reconstructions. As for "do not necessarily imply a comparable CO2 drawdown", isn't this obvious already when considering the carbon reservoirs on land?

- L305: "additional proxy data". "additional" with respect to which data? I remember of no proxy data which was used in this section.

- L317-318: It is not obvious to me when observing Fig. 6 that AERO simulation has the best model-data agreement, and I expect it is because it is fAERO and fBASE (to add in mind then) which are represented again… Same for L324 "which is consistent with the reconstructions" (there seems to be many black points in these regions as well)

- L323: This had me wondering where the extra PO4 was put in the PO4 simulation. Is it uniformly distributed at the surface ocean?

- L417: What do you mean by "sensitivities"?

- L422: Note that this is the first occurrence of the formulation "alkalinity nudging" for simulation CO2T. You might want to consider to use consistent terminology in the Methods/Results.

- L435-437: According to model simulations or data?

- Figure 11: Please consider providing quantifications of the model-data agreement such as a RMSE (here and in other instances).

- L484: REMI results are commented but not shown in Fig. 11.

- L524: It is unclear which model-data comparison "previous" is refering to.

- L553: "the bias in simulated global carbon fluxes and reservoir size changes". Why bias do you mean? Or do you mean "biases" in a general sense?

- L558-559: as well as changes in alkalinity input fluxes, right?

- L565: Could these imbalances in weathering-burial (in terms of carbon, nutrients, alkalinity) be quantified or represented somewhere? Maybe this could lead to a more affirmative sentence than "it seems likely that".

**5. Discussion**

- L617: Why "in addition" and not "such as"?

- L635: "Our simulations with increase organic carbon burial". Consider citing the simulation names again.

- L683: Please explain how. Same for L645.

- L646: "especially those which reproduce the reconstructed increase of organic carbon burial during glacial maxima" → Looking again at Fig. 10, I would say none of the simulations do that…

- L651: "big sedimentary changes": Does this exclude the fraction lost to inert sediments which is compensated by weathering inputs?

- L652: You could refer to a figure again.

- L655-656: But didn't Morée et al. perform equilibrium runs? (I could be mistaken, I haven't checked the paper.)

**6. Conclusion**

- L673: "other processes must have operated". It is unclear what "other processes" are refering to. Do you mean physical processes (e.g. circulation changes)? Do you mean "compensating processes"?

- L678: This was not really shown in the text previously.

- L680: Do you mean "ocean carbon inventory" or "ocean inorganic carbon inventory"?

- L691-692: And so, what are the perspectives which can be infered? For example, would you be able to provide recommendation for less computationnaly-efficient models which cannot afford to run 200 kyr?

**Technical/small comments**

- L2 : Consider putting "atmosphere" last in the list.

- L2 : "… preserved biogeochemical evidence" → "… preserved indirect biogeochemical evidence"?

- L7 : "uncertainty" → "knowledge gap"

- L16 : Consider removing "likely".

- L21 : "Earth's carbon cycle" → "the Earth's carbon cycle" ?

- L32 : Consider quoting Kohfeld and Ridgwell, 2009.

- L33 : "last glacial maxiumum" → "Last Glacial Maximum"

- L37 : "added CO2 back" → "tend to counteract this effect by stimulating CO2 outgassing"

- L42 : "as well as increased nutrient supply" → "as well as increased biological pump from increased nutrient supply"?

- L81 : "combinations" or "combination" ?

- L83 : Why isn't "shallow water carbonate burial" not included in the list of biogeochemical processes with the others?

- L87 : "models" or "model"?

- L98 : "The Bern 3D 2.0 model"?

- L100 : "41x40" → why not provide the resolution in °?

- L130 : Please consider avoiding mentioning the simulation names before they are properly defined.

- L149 : "Finally" is inadequate as you are not describing the final simulation, merely the last simulation with different physical forcing.

- L152 : "added" → "mimicked", as it is not this terrestrial sink/source is not explicitly resolved by the model.

- L210 and L211 : "three" → "four"

- L257 : "under" → I think you mean "subsequently to", since forcings (e.g. PO4 supply) would be close to 0 during interglacials and we are seeing a memory effect.

- L261 : "processes" → "forcings"

- L269 : typo "productivity"

- L270 : "played" → "plays"

- L275 : I may be wrong, but I think "towards" indicates a spatial direction (and may not be adequate for a time direction). Same for L410.

- L278 : "lags behind" typo

- L284 : "40 °S" typo

- L303 : "marine carbon storage" → I think you mean DIC, to exclude organic carbon.

- L315 : invert "the LGM and the Holocene"

- L342 : typo citep not citet

- Figure 10 : Gray shading is barely visible in print format.

- L465 : typo "reproduce"

- L467 : typo "increased"

- Fig. 12 : Dotted light green-yellowish line is barely visible in print format.

- L516 : typo citet no citep

- L550 : "the real processes they represent" → "mimick"

- L661 : missing preposition

- L663 : typo "ice sheets"

- L686 : $CO_3^{2-}$

---

## Referee Comment (RC2)

**Review of egusphere-2024-1754**

**Glacial ocean cSediment fluxes dominate glacial-interglacial changes in ocean carbon inventory: results from factorial simulations over the past 780,000 years**

**by M. Adloff, A. Jeltsch-Thömmes, F. Pöppelmeier, T. F. Stocker, and F. Joos**

The authors conducted an investigation into the differences in the ocean carbon cycle response during glacial-interglacial cycles, comparing scenarios with and without sedimentation processes. By incorporating various idealized forcings based on ice core and sediment core records, they analyzed not only atmospheric $CO_2$ but also $\delta^{13}C$ in the atmosphere and ocean, oceanic DIC, regenerated DIC, and $CO_3^{2-}$, comparing these with variations reconstructed from geological records. The study identifies the dominant processes driving these changes, with the authors concluding that variations in oceanic DIC are more significant than those resulting from changes in carbon inventory driven by atmospheric $CO_2$, thereby highlighting sedimentation processes as the primary driver of DIC variations. These findings are robust and significant, making them well-suited for publication in Climate of the Past. However, as another reviewer has noted, the main text and supplementary materials are quite dense, often requiring the reader to refer back to the experimental setup. Additionally, it can be difficult to discern which figures correspond to specific descriptions. Enhancing the clarity of these elements would greatly benefit the overall communication of the study's results.

**General comments:**

The supplementary material touches on changes in deep ocean circulation, but before diving into a more detailed discussion, it is useful to first introduce what happens in the BASE scenario to provide a clearer context.

In the sensitivity experiments, different forcings are applied. However, while the LGM-PI amplitudes are determined for each experiment, it would be helpful to provide a more detailed explanation of the rationale behind these choices. For example, why was an amplitude of -40 chosen for SOWI rather than -30 or -50? A clearer justification for these specific values would help readers better understand the experimental design.

It would be beneficial to clearly identify what this model successfully captures and what it may be lacking, based on the results of these experiments.

The point that simple changes in the DIC inventory do not fully explain atmospheric $CO_2$ variations is particularly compelling and aligns well with my understanding.

The size of the figure captions, plots of sediment core data, and the contrast in the line plots may currently lack sufficient clarity, which could affect the overall readability.

**Specific comments:**

L30: It might be helpful to introduce "Last Glacial Maximum" as "LGM" when first mentioned, and then use the abbreviation in subsequent references throughout the text.

L46: "… but not necessarily in open systems": Based on the current results s, do you have any discussions related to these previous studies? It is interesting to note that, depending on the experiment, DIC inventory either increases or decreases during glacial periods. Clarifying how these findings relate to or contrast with previous studies could provide valuable insights.

L117: How are the dissolution processes of organic matter and calcium carbonate in the sediment model formulated? Specifically, regarding the burial dissolution of organic matter, since it is also mentioned later when explaining changes in oxygen concentration, it would be helpful to explicitly describe the dependence of these processes on oxygen concentration.

L125: "balance...kept constant thereafter,": Does this mean that during initialization the river input is set to balance the burial rate and this value is used consistently throughout? If so, could you also clarify the specific values of these rates?

L142: Could the results be largely different depending on which variation ($\delta D$ or $\delta^{18}O$) each experiment is concerned with?

L153: Is there a specific assumption or basis for the 30%?

L159: How was the alkalinity adjustment carried out?

L228: "However, … constant.": Which figure does this description correspond to?

L284: "40°CS" is a typo and should be corrected to "40°S."

L355: What is happening in the case of SOWI? Is the weakening of the AMOC leading to an increased accumulation of DIC in the deep ocean?

L370: As mentioned earlier, to clarify the relationship between oxygen depletion and the increase in organic carbon burial, could you provide the specific formulation used?

L393: "However … during the deglaciation." Which figure should be referenced to understand this description?

L433: "which may be linked to changes in weathering fluxes not considered here.": Does this mean that changes in weathering could lead to increased inputs of DIC with lower carbon values?

L457: "the reconstructions show…": Where can I find information on the changes in POC burial by region?

L484: "… better simulated in REMI": Can this be understood from Figure 11c (BGC)?

L494: Does this imply that alkalinity is being removed too quickly in order to reproduce $CO_2$ levels?

L497: Which region's sediment core does Qin et al. (2018) refer to? There appear to be other reconstructions of $[CO_3^{2-}]$ as well. Could you clarify why the comparison was made exclusively with this particular study?

L498: Why is it that, in Experiment BASE, $CO_2$ shows a significant change, yet there is little to no change in $CO_3^{2-}$?

L539: "… increased remieneralization of sedimentary organic matter …": Does this contribute to the rise in $CO_2$ by the slow decomposition of organic matter once it has accumulated in the sediment?

L592: How does the change in AMOC affect atmospheric $CO_2$ in this model? Based on this description, does a weakening of the AMOC lead to an increase in $CO_2$?

L598: Does "increased Southern Ocean wind forcing" refer to a weakening of the wind forcing?

L608: typo: it should be "Menviel et al. (2011)" rather than "(Menviel et al., 2011)."

L686: typo: it should be "$CO_3^{2-}$" rather than "$CO_3^{--}$."

Table2: Since only $CO_2$ reflects the difference between the Holocene and the glacial period, it might be worth considering aligning the other variables as well for consistency.

Figure 9: Does $\Delta\delta^{13}C$ represent the difference from the modern value? It would be helpful to clarify this in the footnote, as it is not currently specified.

Figure 15: It might also be helpful to change the color of the lines to make them easier to distinguish at a glance.

---

## Author Comment (AC1)

Glacial ocean cSediment fluxes dominate glacial-interglacial changes in ocean carbon inventory: results from factorial simulations over the past 780,000 years
by M. Adloff, A. Jeltsch-Thömmes, F. Pöppelmeier, T. F. Stocker, and F. Joos

The authors conducted an investigation into the differences in the ocean carbon cycle response during glacial-interglacial cycles, comparing scenarios with and without sedimentation processes. By incorporating various idealized forcings based on ice core and sediment core records, they analyzed not only atmospheric $CO_2$ but also $\delta^{13}C$ in the atmosphere and ocean, oceanic DIC, regenerated DIC, and $CO_3^{2-}$, comparing these with variations reconstructed from geological records. The study identifies the dominant processes driving these changes, with the authors concluding that variations in oceanic DIC are more significant than those resulting from changes in carbon inventory driven by atmospheric $CO_2$, thereby highlighting sedimentation processes as the primary driver of DIC variations. These findings are robust and significant, making them well-suited for publication in Climate of the Past. However, as another reviewer has noted, the main text and supplementary materials are quite dense, often requiring the reader to refer back to the experimental setup. Additionally, it can be difficult to discern which figures correspond to specific descriptions. Enhancing the clarity of these elements would greatly benefit the overall communication of the study's results.

We thank the reviewer for the fair assessment and constructive comments. We are largely revising and restructuring the main text to shorten it and clarify the main points of our analysis.

**General comments:**
The supplementary material touches on changes in deep ocean circulation, but before diving into a more detailed discussion, it is useful to first introduce what happens in the BASE scenario to provide a clearer context.
In the sensitivity experiments, different forcings are applied. However, while the LGM-PI amplitudes are determined for each experiment, it would be helpful to provide a more detailed explanation of the rationale behind these choices. For example, why was an amplitude of -40 chosen for SOWI rather than -30 or -50? A clearer justification for these specific values would help readers better understand the experimental design.
It would be beneficial to clearly identify what this model successfully captures and what it may be lacking, based on the results of these experiments.
The point that simple changes in the DIC inventory do not fully explain atmospheric $CO_2$ variations is particularly compelling and aligns well with my understanding.
The size of the figure captions, plots of sediment core data, and the contrast in the line plots may currently lack sufficient clarity, which could affect the overall readability.

We will largely rewrite the methods, results and discussion of our manuscript to improve clarity and focus on the most relevant points. We also change the figures to improve their readability.

**General Comments:**
L30: It might be helpful to introduce "Last Glacial Maximum" as "LGM" when first mentioned, and then use the abbreviation in subsequent references throughout the text.

We change all instances of this as suggested.

L46: "… but not necessarily in open systems": Based on the current results s, do you have any discussions related to these previous studies? It is interesting to note that, depending on the experiment, DIC inventory either increases or decreases during glacial periods. Clarifying how these findings relate to or contrast with previous studies could provide valuable insights.

We will add the following clarification to the introduction:

"In a (hypothetical) closed atmosphere-ocean system, the combination of these processes results in increased marine carbon storage during glacials, but not necessarily in the open Earth system because the carbon removed from the surface ocean and atmosphere by these processes could have been sequestered in the water column as DIC or particulate carbon but also in marine sediments. Carbon can also be transferred to the land. Constraints on glacial atmospheric $CO_2$ can be reconciled with increased and decreased marine DIC inventory in an open system (Jeltsch-Thömmes et al., 2019; Kemppinen et al., 2019), though reproducing reconstructed carbon isotopic changes in atmosphere and ocean seems to require elevated DIC at the LGM (Jeltsch-Thömmes et al., 2019)."

And the following discussion:
"Some of the tested forcings also show lower glacial than inter-glacial DIC ($fPO4$, $fCO2T$) showing that $CO_2$ removal from the atmosphere in theory does not need to result in increased DIC in the ocean. Instead, these biogeochemical forcings cause sedimentary changes that can store large amounts of carbon in inorganic and organic sedimentary matter. Kemppinen et al. (2019) and Jeltsch-Thömmes et al. (2019) previously showed and discussed the possibility of a negative glacial DIC anomaly due to increased sedimentary storage. As found by Jeltsch-Thömmes et al. (2019), organic carbon burial extensive enough to cause a negative glacial DIC anomaly (e.g. $fPO4$) produces large $\delta^{13}C$ signals of opposite sign than reconstructed, and thus seems unlikely. In the study by Jeltsch-Thömmes et al. (2019), a negative glacial DIC anomaly due to alkalinity-driven $CaCO_3$ accumulation is also inconsistent with the proxy record of the last 25 kyr. Consistently, we find that reconstructed deep Pacific $[CO_3^{2-}]$ changes make a large-scale alkalinity-driven ($fCO2T$) glacial $CaCO_3$ accumulation, which reduces atmospheric $CO_2$ while also reducing DIC, unlikely because it causes larger deep Pacific $[CO_3^{2-}]$ changes than reconstructed over the last deglaciation (Table 2). The isotopic signal of such large $CaCO_3$ deposition, however, is smaller than that of POC burial changes and could more likely be overprinted by other processes (e.g. terrestrial carbon release and export production changes) to yield proxy-consistent evolutions (Table 2)."

L117: How are the dissolution processes of organic matter and calcium carbonate in the sediment model formulated? Specifically, regarding the burial dissolution of organic matter, since it is also mentioned later when explaining changes in oxygen concentration, it would be helpful to explicitly describe the dependence of these processes on oxygen concentration.

We will add the following to the model description:

*"CaCO₃ dissolution rates in the sediments are determined from the pore water saturation state, and POC remineralisation is parameterised by a linear dependence on porewater $O_2$ (Heinze et al., 1999; Tschumi et al., 2011). "*

L125: "balance...kept constant thereafter,": Does this mean that during initialization the river input is set to balance the burial rate and this value is used consistently throughout? If so, could you also clarify the specific values of these rates?

That is correct, we diagnosed the burial fluxes at the end of the spin-up and kept the compensating continental inputs constant throughout the transient simulations. We will add a table with the specific fluxes to the SI.

L142: Could the results be largely different depending on which variation (δD or δ 18 O) each experiment is concerned with?

The only difference would be the timings of environmental shift due to lags between δD and $\delta^{18}O$ during deglaciations. The amplitudes would be similar because both forcings were normed.

L153: Is there a specific assumption or basis for the 30%?

There is no specific reason for the exact maximum magnitude of the forcings we apply, they were chosen to cause noticeable $CO_2$ or circulation shifts, informed by previous studies.

We will add the following to the description of the experiment design:

*"Data constraints on carbon cycle forcings are too sparse to know exact magnitudes and timings of the forcings that might have varied spatially and temporarily over the last eight glacial cycles. An inverse estimation of the forcings from the resulting proxy signals requires a different simulation ensemble and is beyond the scope of our study. Rather than trying to guess the most proxy consistent forcing amplitudes and patterns, we designed seven simplified forcings, each with one exemplary magnitude, to simulate the generic effects of processes that have been identified as glacial-interglacial carbon cycle drivers. Except for the orbital changes, which were calculated following Berger (1978); Berger and Loutre (1991) and the reconstructed $CO_2$, $N_2O$ and $CH_4$ curves (Loulergue et al., 2008; Joos and Spahni, 2008; Bereiter et al., 2015; Etminan et al., 2016), which we used to calculate the radiative forcing of greenhouse gas changes, the amplitudes of the forcings were set to cause noticeable $CO_2$ or circulation shifts, informed by previous studies (e.g. Tschumi et al., 2011; Menviel and Joos, 2012; Menviel et al., 2012; Jeltsch-Thömmes et al., 2019; Pöppelmeier et al., 2020)."*

L159: How was the alkalinity adjustment carried out?

We will add the following explanation to the description of our experiment design:

*"In addition we performed one run in which we let the model dynamically apply external alkalinity fluxes (in addition to the constant terrestrial solute supply applied in each simulation, see spin-up methodology) to restore the reconstructed atmospheric $CO_2$ curve*

*(CO2T). In this simulation, the model evaluates the difference between the simulated and reconstructed $CO_2$ at each time step and adds or removes the marine alkalinity required to cause the necessary compensatory air-sea carbon flux from the surface ocean. Alkalinity changes, e.g. due to changes in shallow carbonate deposition or terrestrial weathering, are an effective lever for atmospheric $CO_2$ change (e.g. Brovkin et al., 2007), and this additional run shows the long-term changes in marine biochemistry if this was the dominant driver of glacial-interglacial atmospheric $CO_2$ change."*

L228: "However, … constant.": Which figure does this description correspond to?
This statement was meant in relation to the $\delta$D record shown in Fig. 1. We will clarify this passage.

L284: "40°CS" is a typo and should be corrected to "40°S."

Will be amended.

L355: What is happening in the case of SOWI? Is the weakening of the AMOC leading to an increased accumulation of DIC in the deep ocean?

The increased isolation of the deep Pacific in SOWI causes increased preservation of POC and $CaCO_3$ in the sediments, and thus less DIC accumulation in the deep ocean than under the BASE forcing. AMOC weakening without wind stress changes (simulation AERO) leads to increased DIC in the deep Atlantic. We reworded the sentence for clarity.

L370: As mentioned earlier, to clarify the relationship between oxygen depletion and the increase in organic carbon burial, could you provide the specific formulation used?

We will extend the model description as stated above. For the specific equations, we refer to Appendix A of Tschumi et al., 2011 (equation A3).

L393: "However … during the deglaciation." Which figure should be referenced to understand this description?

This section was discussing the old Fig. 9. We will revise the section with clearer figure references.

L433: "which may be linked to changes in weathering fluxes not considered here.":
Does this mean that changes in weathering could lead to increased inputs of DIC with lower carbon values?

This is a speculation but one possibility is that the mean $\delta^{13}C$ of continental DIC reaching the ocean was different during the Eemian, i.e. because of different weathered lithologies or different DIC transport/transformations in terrestrial water ways.

L457: "the reconstructions show…": Where can I find information on the changes in POC burial by region?

We will add a map of POC burial to Fig. 10.

L484: "… better simulated in REMI": Can this be understood from Figure 11c (BGC)?

We will improve the figure and clarify the text, such that this point is now better understandable.

L494: Does this imply that alkalinity is being removed too quickly in order to reproduce CO 2 levels?

Yes, the alkalinity forcing required to drive the whole deglacial $CO_2$ change by ALK changes is too strong for sedimentary $CaCO_3$.

L497: Which region's sediment core does Qin et al. (2018) refer to? There appear to be other reconstructions of [CO 32- ] as well. Could you clarify why the comparison was made exclusively with this particular study?

Qin et al. (2018) present a record of the western tropical Pacific. Here, we are not focussing on site-specific details of the record but the general pattern of glacial-interglacial changes and their amplitudes. The most important question for us is the glacial-interglacial $[CO_3^{2-}]$ amplitude and how it compares across the last eight glacial cycles. Over the last glacial cycle, the low glacial-interglacial amplitude shown in Qin et al. (2018) is similar in other Pacific records (e.g. Yu et al., 2013, Kerr et al., 2017). We chose the Qin et al. (2018) record for Fig. 12 because it is the best resolved record we could find covering almost the entire simulation period. The resolution of a later and longer record (Qin et al., 2020) has a lower resolution over 800 ka. Over the deglaciation (Fig. 13), we also compared the simulation results to Yu et al. (2013).

L498: Why is it that, in Experiment BASE, CO 2 shows a significant change, yet there is little to no change in CO 32- ?

In BASE, the DIC increase is accompanied by acidification and reduced $CaCO_3$ accumulation, which means that DIC increases while $[CO_3^{2-}]$ stays almost constant.

L539: "… increased remieneralization of sedimentary organic matter …": Does this contribute to the rise in CO 2 by the slow decomposition of organic matter once it has accumulated in the sediment?

In addition, the re-ventilation of the ocean increases oxygen supply to the previously dysoxic sediments, which raises remineralization rates. We will clarify this in the text.

L592: How does the change in AMOC affect atmospheric CO 2 in this model? Based on this description, does a weakening of the AMOC lead to an increase in CO 2 ?

We agree, our formulation was unclear. In most cases, a strengthening of AMOC causes a $CO_2$ increase in our model. However, AMOC is only pushed into a stronger circulation state during this time interval if it had already transitioned into a weaker state previously. Otherwise the orbital forcing during the time window does not cause a shift in AMOC strength.

We will revise the sentence accordingly.

L598: Does "increased Southern Ocean wind forcing" refer to a weakening of the wind Forcing?

Yes, we will correct this.

L608: typo: it should be "Menviel et al. (2011)" rather than "(Menviel et al., 2011)."

Will be amended.

L686: typo: it should be "$CO_3^{2-}$ " rather than "CO 3-- ."

Will be changed.

Table2: Since only $CO_2$ reflects the difference between the Holocene and the glacial period, it might be worth considering aligning the other variables as well for Consistency.

We will change the table to show deglacial changes for all metrics.

Figure 9: Does $\Delta\delta^{13}C$ represent the difference from the modern value? It would be helpful to clarify this in the footnote, as it is not currently specified.

Apologies for the missing information, we will add it to the figure caption. The anomalies are with reference to 20 ka, or the closest available value.

Figure 15: It might also be helpful to change the color of the lines to make them easier to distinguish at a glance.

We will choose different line colours.

---

## Author Comment (AC2)

We thank the reviewer for the thorough discussion of our manuscript and the constructive comments. We will use this opportunity to rewrite and reorganise large parts of the manuscript to produce a more stand-alone, coherent and clear main text, as suggested by the reviewer. We will also remove several parts of the main text and figures that were not essential to the main message, including the discussions of primary production and remineralisation metrics which are less affected by interactive sediments, reducing the manuscript length by nine pages. Below are our responses (in blue) to the specific review comments (in black).

General comments
**1. Making all connections explicit in order to spell out the knowledge gap, and linking it to the methodological choices**
The scientific reasoning is most of the time explicit to the reader, but there are a few instances in which all of these connections could be improved. I am pointing them out here.
- L5 : "Yet, it is unclear how much they affected carbon cycling during transient changes of repeated glacial cycles, and what role burial and release of sedimentary organic and inorganic carbon and nutrients played.". I believe the knowledge gap could be more clearly spelled out, including why you are considering specifically repeating cycles, and the role of sediments. A more explicit link to the chosen scientific approach (justifying the choice of 1/ using "various" forcings, 2/ for simulations of the "last 780 kyr") could then be made.

We will rewrite the introduction and address this concern together with the next (see answer below)

- L45-49 : I think that the knowledge gap could be more clearly spelt out. As it is now constructed, the L45-47 sentence doesn't really justify the case for including sediments. Maybe you could explain that the increased marine carbon storage in closed atmosphere-ocean system is not sufficient to simulate the low glacial atmospheric CO 2 without carbonate compensation (e.g. Kobayashi et al., 2021), and cannot be of the right amplitude without the inclusion of land processes as well, as carbon is lost in the terrestrial vegetation. Without clearly showing the limitations of previous studies, the "little is know" and "it has long been assumed" formulations are not exactly impactful as a knowledge gap.

We agree that our introduction was missing clarity. The cited lines were meant to summarize existing knowledge rather than showing the knowledge gap, which we addressed in the last paragraph of the introduction. We will revise the wording throughout the introduction for clarity and clearly state the knowledge gap addressed in our study in the last paragraph:

"*Here we examine systematically how the transient built-up and dissolution of marine sediments on glacial-interglacial timescales affects the carbon cycle changes produced by the various processes suggested to be relevant on these timescales, a gap left by previous studies. Instead of searching for the most likely scenario that reconciles the vast proxy evidence, we attempt to gain a more complete process understanding and overview of the proxy-relevant signals that these processes cause in the presence of weathering-burial imbalances. With this goal, we extend factorial simulations of multiple simplified physical and biogeochemical forcings in a marine sediment and isotope-enabled intermediate complexity Earth system model over the last 780 kyr and compare the resulting carbon and carbon isotopic signals to reconstructions. The long timescale is chosen to avoid biases*"

*resulting from steady state assumptions and account for the possibility of memory effects under continuously varying climate and carbon cycle that could span multiple glacial cycles. Consequently, all carbon stores are achieved dynamically rather than being prescribed. We present two sets of simulations with and without interactive sediments to distinguish the role of interactive sediments in the carbon cycle changes caused by the tested forcings over reoccurring glacial cycles of the last 780 kyr.*"

- L135-142 and L143-157: The choice of parameters to modify in the "simplified forcings" (wind stress, etc.) is not explicitly linked to the "glacial-interglacial carbon cycle drivers" described in the introduction. The "why" behind all of these simulations is not clear. I think some sentences should be added to make the connection (e.g. "as a modified wind stress significantly influences the deep ocean circulation in our model") with the mentioned drivers or to other relevant purpose (e.g. Southern Hemisphere westerly winds seems to have varied over glacial-interglacial cycles according to Gray et al., 2023). Alternatively, to keep it short, you could add a column "tested drivers" (e.g. ocean circulation) in Table 1. Also, in L139-142, I feel like it could be (1) mentioned which forcing was expected to behave like which delta and why; (2) acknowledged that this is a strong assumption, not allowing for threshold effects for e.g. nutrients supply or terrestrial carbon release. Finally, it could be acknowledged that your set of simulations do not tackle some of the drivers mentioned in the literature (e.g. sea ice). Likewise, it could be recognized that you are testing out phosphate input and not iron fertilisation effects because – I assume – your model does not include a representation of the iron cycle.

We will revise the description of our forcings and further add details of their effects in the model. We will also add a rationale for the choice of whether to tie the forcing to the $\delta D$ or $\delta^{18}O$ record:

"*Specifically, we performed one 'base' run with orbital and radiative forcing only, one model run for different forcings, each added to the base forcing, and combinations of the individual forcings to study non-linear effects that appear when processes interact. All of these experiments are run once with and once without interactive sediments, to examine the effect of sediment perturbations on the results. The forcings and their rationale are described below. The experiments are summarized in Table 1. The application of the standard forcing in simulation BASE causes temperature changes associated with orbital, albedo, and greenhouse gas changes which affect solubility, sea ice and circulation, e.g. slightly weakening AMOC (by up to 4.5 Sv, Fig. S8) and resulting in younger deep water masses in the Atlantic and Pacific during the LGM than at the PI, which is inconsistent with proxy data and thus indicates that additional Earth system changes must have occurred (Pöppelmeier et al., 2020). To achieve an older glacial deep ocean (diagnosed with an ideal age tracer), we reduced the wind stress south of 48°S by a maximum of 40% (simulation SOWI) temporally changing proportionately to the $\delta D$ change because we assume that wind strength over the Southern Ocean evolved without temporal lags to Antarctic temperature. As a result, the South Pacific downwelling is strengthened by up to 1.5 Sv locally in glacials, AMOC strength is further reduced by up to 1 Sv and the simulated deep ocean age is ~100 years older in the LGM than in the PI, close to published model estimates (Schmittner, 2003). In this set-up, changing wind stress only affects the circulation, not the piston velocity of gas exchange, which is forced by a wind-speed climatology. For an independent assessment of the effect of wind speed changes on sea-air gas exchange, we performed a*

*simulation in which we decreased the piston velocity in the Southern Ocean by a maximum of 40% (KGAS), also following the evolution of $\delta D$. Next, we tested an additional negative radiative forcing due to increased dust loads in the glacial atmosphere (e.g. Claquin et al., 2003) by reducing the total radiative forcing by a maximum of 2.5 W/m$^2$ during the LGM to test the effects of stronger AMOC weakening (AERO), modulated by the $\delta^{18}O$ record based on the reconstructed correlation between dust and $\delta^{18}O$ (Winckler et al., 2008, similar to the study of long-term circulation changes in Adloff et al. (2024)). Under this forcing, the AMOC weakens by up to 12 Sv relative to PI during glacial maxima (the model behaviour to this forcing is described more extensively in Adloff et al., 2024) and water mass age rises to up to 1000 years in the deep North Atlantic as glacial deep water formation now only occurs in the Southern Ocean. In terms of biogeochemical forcings, we mimicked a terrestrial carbon sink/source by removing/adding 500 PgC during deglaciation/ice age inception (LAND Jeltsch-Thömmes et al., 2019) and increased the marine phosphate inventory by 30% during the glacial maxima by a globally-uniform supply of phosphate into the surface ocean (PO4). The timeseries of both forcings are proportional to $\delta^{18}O$ changes, because we assume that both are lagging behind temperature changes due to continental ice-sheets and changing terrestrial environments. Effectively, our nutrient forcing reduces nutrient limitation globally. Rather than simulating the effects of different nutrient inputs in different regions (e.g. iron in the Southern Ocean, phosphate at shelves), we decided to group all these in one simulation with a global forcing because their net effect, increased export production, would be the same in our model, just in different regions. This is the only forcing that we did not apply to the model without interactive sediments because, while nutrients can be added to the surface ocean periodically, there is no simple way of artificially extracting nutrients from the ocean in return. We also reduced the speed of aerobic organic matter remineralization in the ocean by transitioning between the standard, pre-industrial Bern3D particle profile (Martin scaling) during interglacials and a linear profile in the first 2000 m of the water column (REMI, Fig. S9), following the $\delta D$ record, since we assume that remineralization changes happened synchronously with temperature change. Next, we reduced the PIC:POC rain ratio by 33% in the LGM (PIPO) and similarly modulated the forcing timeseries with the $\delta D$ record. In addition we performed one run in which we let the model dynamically apply external alkalinity fluxes (in addition to the constant terrestrial solute supply applied in each simulation, see spin-up methodology) to restore the reconstructed atmospheric $CO_2$ curve (CO2T). In this simulation, the model evaluates the difference between the simulated and reconstructed $CO_2$ at each time step and adds or removes the marine alkalinity required to cause the necessary compensatory air-sea carbon flux from the surface ocean. Alkalinity changes, e.g. due to changes in shallow carbonate deposition or terrestrial weathering, are an effective lever for atmospheric $CO_2$ change (e.g. Brovkin et al., 2007), and this additional run shows the long-term changes in marine biochemistry if this was the dominant driver of glacial-interglacial atmospheric $CO_2$ change."*

**2. Reorganizing elements from/to the Supplementary to make for a more standalone, understandable, and trimmed down main text**

- Additional sensitivity tests in L165-169: There are already a great number of simulations. I feel like these additional ones don't need to be part of the main text as they do not seem to closely relate to your main scientific question and message. Or do they?

We will agree and hence remove the additional sensitivity experiments from the manuscript.

- L193-196: It is odd to start the results section by refering to the extensive material presenting in the SI (which almost feels like a companion paper). One sign is that the reader may struggle to understand the effects of individual forcings (see specific comments on Table 2, Fig. 3…) before seeing the results of individual simulations. In a way, to properly understand the paper, reading the first 14 pages of the Supplementary is necessary. This is where the processes explaining the changes in carbon storage in the different reservoirs are described. Yet it could be argued that at least one of the ΔC graphs in the SI should be part of the main text. Understanding the evolution of the distribution of carbon between the different pools seems important. But it seems that the weathering-burial imbalances are causing some issues at the end of transient runs (esp. SOWI, p.16), are they not? Another sign is that despite the scientific question on the role of sediments, "a detailed analysis of the sedimentary changes" is part of the SI. I believe that since most readers won't read the SI, the authors should strive towards a more standalone main text.

We will restructure the main text of the manuscript and start with a description of the simulated carbon fluxes before continuing with the model-data comparison. We will move some paragraphs from the SI into the main text and add a new summary. We will keep additional text in the SI for interested readers but the main text will contain all relevant information to understand the simulations without reading the SI.

- Please consider briefly introducing the proxy reconstructions used in this study in the start of the results sections or in the methods. At the start of the results section when they are already used right off the bat, the reader only knows that many are "available" (L205) and that some were "selected" (L207),with has no idea how these reconstructions were made, if they are robust, how they can be compared to the model results (thanks to the isotope-enabled feature), and on which basis some were apparently selected…

We will explicitly state that we exemplarily select records that show prominent features that point at carbon cycle changes, and that we do not attempt a thorough data compilation or to explain any record in detail. Rather, we use these records exemplarily to show how the coupling interactive sediments affects the interpretation of the records. For the reliability of the records, we rely on the uncertainty assessment done in the original studies. Detailed assessment of the proxy records is beyond the scope of this study and not necessary for the points we want to make.

- L230: The first mention of a figure in SI is numbered S17. We jump to S22 in L234. The numbering of the figures in SI therefore do not seem consistent. This doesn't help the back-and-forth between the main text and the SI which is too often required to understand the main text properly. Despite the amount of materials, some of this back-and-forth could be prevented. For example, for L230-234 the authors could consider only relying on Fig. 3b, or a modified Fig. 3b, making the main text standalone by removing nuances which can only be explained by the extra figures. These few sentences could be kept in the SI, along with Fig. S17 and S18, in a numbered section addressing a specific question (e.g. "What are the lags between CO2 and the forcings?"), with the main text only commenting "lags observed in Fig 1 and 3b depend on the choice of forcings and are more fully explained in section S1 of the SI". Alternatively, the main information of Fig. S17 could be shown within Fig. 1, as a zoom on the last termination.

We will reorganise the SI such that the sequence of the SI figures follows their first mention in the main text. We will also move some SI figures into the main text and remove some discussion of details that are not essential for the main text.

- Figure 4 and L240-251: It is unclear to me what this whole section on the MBE actually brings to the table except for the message in L250-251. In the introduction, the MBE wasn't exactly part of the main scientific question. While this is interesting, describing the lags between CO2 and the forcing in different sensitivity tests (previous comment), or the maximum amplitude of the interglacials before/after the MBE across different sensitivity tests seem like details when the main text has yet to explain which processes are behind the glacial-interglacial variations in these simulations. I would argue that this whole part could be moved to the SI (as a "please refer to section S2 "How is the MBE simulated?" in the SI for more.")

We will follow the suggestion and removed this part from the main text, keeping a short reference to the Figure moved to the SI.

- L251-261: Some of the processes are here explained. However, they are just described and not shown. No figure is referred. Please consider whether some sort of modified Fig. S4 (incl. evolution curves for atmospheric CO2, DIC, alkalinity, CaCO3 deposition, POC deposition, and also the weathering-burial imbalance…) could be of use in the main text for the reader to properly understand the simulations. I was also curious to see organic carbon variations.

We will add a figure with the simulated fluxes of inorganic and organic carbon, to summarize the model behaviour in response to the tested forcings.

- Section 4.2: Fig. 5 is first shown but not described, only briefly refered to L298. The text goes on to explain the importance of the Southern Ocean (not shown, and we are wondering why we are starting by regional changes), then surface pH (shown, but in SI). Why are carbonate shifts described via pH changes and not alkalinity? So far, the link with DIC has not been made explicit. Perhaps the authors should consider transforming Figure 5a into a DIC-ALK diagram. Also, "The spatial patters of marine carbon uptake and release are thus driven by combinations of physical and chemical processes" (L282) are never shown, making the sentence feel out of place and unproven.

We will restructure the section on DIC and remove part of the discussion that was not central to the main points of the manuscript.

- Fig. 14: Is it really essential to include in the main text?

We will move Fig. 14 to the SI.

- Please note that L524-542 somehow already feels like a conclusion, and L543-559 like a discussion. Perhaps this should be acknowledged in the subsection title.

We agree and move these sections to the beginning of the discussion.

- L560-564: Is it really essential to include in the main text?

We will remove this section from the main text and add it to the SI.

- Page 32, incl. Fig. 15: Is it really essential to include in the main text?

Yes, we think this is relevant. The long-lasting $\delta^{13}C$ drifts are a feature that we expect to occur in all models with weathering-burial imbalances, questioning how transient isotopic signals in short simulations should be interpreted.

- L593-598: Is this discussion relevant? Increasing the vertical diffusivity is not what you did, and KGAS has only a minor effect.

We will remove these sentences.

- Fig S23-S24 + Fig S36-39: Considering your outcomes of Section 4.2 and the fact the REMI seems to strongly imprint on the direction of the nutrient changes during deglaciation in BGC and ALL, and on the model-data agreement for delta13C, I am wondering whether you have considered a BGC simulation excluding REMI?

We agree that additional combinations of the tested forcings would be interesting to study. However, adding new simulations is not feasible at this point. Each simulation takes several months to run, and then time to analyse. We are confident that the existing set of simulations provides sufficient insight for this manuscript.

We will add this point to our discussion of the limitations of our study (lines SI 20-23): *"A more detailed analysis of non-linear interactions between the tested forcings would require an additional simulation ensemble that tests all possible forcing combinations and ideally also with varying forcing magnitudes."*

**3. A couple of reccurent inadequate terms**
- In L2 and other instances, it is unclear why the reactive layers of sediments are considered "ocean sediments" whereas the inert layers are part of the "lithosphere". From a geological point of view, this is disputable. Other instances include L48-49, where "marine sediments and the lithosphere" are mentioned as a unique "fourth reservoir", which makes it confusing why this distinction is introduced in the first place.

We will change all instances and differentiate between reactive sediments and sediment burial now.

- L171 and many other instances: The term "simulated process" is often used to designate the simplified forcings prescribed in the simulation. I would be careful with this term. Processes may refer to the mechanisms of the carbon cycle which contribute to explaining glacial-interglacial variations (e.g. changes in water mass distribution, carbonate compensation…). We can achieve process- understanding (to some extent at least) by examining the model's response to different forcings. But prescribed parameters are hardly processes by themselves. Same for e.g. L202 "We investigate the isolated processes."

We agree with the reviewer and will revise our word choices in the text, changing 'process' to 'forcing' where necessary.

- L192 and many other instances: I believe that the term "biochemical" refers to molecular biology and that "biogeochemical" should be used.

Will be amended.

**Specific comments**
**1. Abstract**
- L3 : I think that carbon transfers don't directly result from different "sensitivities" of the carbon reservoirs to the forcing, but result from the different responses to the forcing of the carbon reservoirs, which depend on their sensitivities?

We will change 'sensitivities' to 'responses'.

- L3 : It is unclear what "many of which" refers to.

We will change the sentence structure and now write of 'poorly understood responses'.

- L10 : It is unclear what "associated" refers to.

We will write of 'carbon fluxes resulting from the forcings'.

- L10 : The importance of considering "isotopic shifts" is not introduced.

We will add that isotopic changes can serve as proxy for carbon fluxes: 'and the associated isotopic shifts that could serve as proxy data'.

- L12-13 : "In our simulations the ocean inventory changed by 200-1400 GtC and the atmospheric inventory by 1-150 GtC over the last deglaciation." It is unclear which conclusion the reader should draw from these results.

We will restructure the sentences to provide clearer lessons:

*"In our simulations, the forcings cause sedimentary perturbations that have large effects on marine and atmospheric carbon storage. Dissolved Inorganic Carbon (DIC) changes differ by a factor of up to 28 between simulations with and without interactive sediments, while $CO_2$ changes in the atmosphere are up to four times larger when interactive sediments are simulated. The relationship between simulated DIC (-1800–1400 GtC) and atmospheric $CO_2$ change (-170–190 GtC) over the last deglaciation is strongly setup-dependent, highlighting the need for considering multiple carbon reservoirs and multi-proxy analyses to more robustly quantify global carbon cycle changes during glacial cycles"*

- L18 : It is unclear what "needs to be considered" refers to. Does this refer to isotopic drifts?

Yes, we will split the sentence to make it clearer:

*" Finally, initiating transient simulations with an interglacial geologic carbon cycle balance causes isotopic drifts that require several 100 kyr to overcome. These model drifts need to be considered when designing spin-up strategies"*

**2. Introduction**
- Introduction : Please consider mentioning the permafrost as a carbon reservoir of relevance.

Will be done.

- L24 : It is a bit unclear which variable is delayed with respect to the others.

We will clarify this: *"and is lagged by ice sheet extent"*

- L39 : Although the "enhanced stratification due to brine rejection" is classified in "other physical processes", stratification closely relates to "changes in ocean circulation" (L34).

Yes, we will discuss different processes that caused circulation changes. To clarify, we will first list "changes in ocean circulation due to lower temperatures", and then add that brine rejection can cause additional stratification.

- L50 : I suggest changing "on continental shelves" by "considering continental shelves". "On" is slightly odd as the sentence refers to altered "seawater carbonate chemistry", but no seawater remains "on" continental shelves when they are emerged.

Will be done.

- L61-63 : A lot of new notions are introduced here, which could be hard to follow for non-specialists. Please consider briefly explaining (1) why carbon isotopes are used in model simulations, (2) what is the burial-nutrient feedback, (3) what you mean by dynamic sedimentary adjustment, and (4) the relevance of imbalances in weathering-burial fluxes when considering an open carbon system. It should also be mentioned whether this imbalances in fluxes refer to carbon fluxes, or also include nutrients. I note that sentences in L65-68 and L69-73 are much more explicit, so it is just the first occurrence of theses processes which I find not easily understandable. A small reorganization of these sentences could therefore be considered. To this end, I should also mention that the L62-64 about equilibration time feels out of place in a paragraph whose main message is to underline the importance of including sediment for its impact on the carbon cycle.

We will follow the reviewer's suggestion and reorganize the sentences, merging this paragraph with the previous and following ones. The new text about sediments during glacial cycles will be:

*"It is very probable that changing sedimentary carbonate and particulate organic carbon (POC) burial played a relevant role in glacial-interglacial carbon cycle changes by altering seawater carbonate chemistry, carbonate ion concentrations, carbon isotope ratios, and oxygenation. Particularly, continental shelves have emerged from the ocean during glacial sea level low stands and provided new reef habitats and carbonate deposition environments*

*during deglaciations and interglacials (e.g. Broecker,1982b; Opdyke and Walker, 1992; Ridgwell et al., 2003; Brovkin et al., 2007; Menviel and Joos, 2012). Additionally, carbonate burial changes in the open ocean have been considered as amplifiers of marine carbon uptake (e.g. Archer and Maier-Reimer,1994; Kohfeld and Ridgwell, 2009; Schneider et al., 2013; Roth et al., 2014; Kerr et al., 2017; Kobayashi et al., 2021). Organic carbon burial is also prone to vary in response to changes in the rain rate of POC sinking to the sea floor and altered oxygenation. Previous model simulations, that included POC burial, showed that interactive sediments greatly affect atmospheric $CO_2$ and carbon isotope variations through the burial-nutrient feedback, whereby enhanced burial of organic-bound carbon and nutrients reduces export production (Tschumi et al., 2011; Roth et al., 2014; Jeltsch-Thömmes et al., 2019; Jeltsch-Thömmes and Joos, 2023). Reconstructions of marine burial changes over the last glacial cycle suggest a reduction in globally-integrated inorganic carbon burial (Cartapanis et al., 2018; Wood et al., 2023) during the last glacial period, but increased organic (Cartapanis et al., 2016) sedimentary carbon burial. The extents of both changes are uncertain due to the spatial heterogeneity of sedimentary burial and the inherently local nature of marine archives, but possibly of comparable magnitude to terrestrial carbon stock changes (Cartapanis et al., 2016, 2018). These findings demonstrate that organic and inorganic sedimentary changes and imbalances with weathering fluxes need to be considered when quantifying carbon reservoir changes of the ocean, atmosphere, and land and interpreting the reconstructed changes in $CO_2$, carbonate ion concentrations, isotopes, and nutrients over glacial cycles.*

*Model-based estimates of carbon and carbon isotope inventory differences between glacial and interglacial periods are complicated by temporal carbon cycle imbalances during the continuously evolving climate of glacial cycles. This is particularly challenging when simulating dynamic elemental cycling in and burial from reactive marine sediments and the input of elements by weathering and volcanic outgassing because of long-lasting re-equilibration and memory effects in carbon and nutrient fluxes and particularly isotopic changes (Tschumi et al., 2011; Jeltsch-Thömmes and Joos, 2020). Dynamic sedimentary adjustment, i.e. the equilibration of sedimentary dissolution and remineralization to changes in bottom water which slowly diffuse into sedimentary porewater, and imbalances between the supply (weathering) and loss (sedimentary burial) of carbon and nutrients also increase the equilibration time of atmospheric $CO_2$ by a factor of up to 20 to several tens of thousands of years and the resulting $\delta^{13}C$ perturbations take hundreds of thousands of years to recover (Roth et al., 2014; Jeltsch-Thömmes et al., 2019; Jeltsch-Thömmes and Joos, 2023). Importantly, the equilibration time scales are longer than typical interglacials in the late Pleistocene, which opens up the possibility for memory effects that span several glacial cycles."*

- L74-75 : So far, the main messages that I am getting are "Sediment fluxes are important to consider / they are difficult to simulate". I think that the long equilibration time ( L62-64) and memory effects now mentioned could be used to explicitly make the case for transient simulations of multiple cycles. As of now, the connection with the proposed study (L90-93) is not clearly made, as the main message from paragraph L76-89 seems to be "no one has done it in this way before".

Yes, we will change the main message accordingly:

*"A caveat of several modeling studies attempting to quantify carbon reservoir sizes at the LGM is that they assume a steady state carbon cycle in a closed (atmosphere-ocean only) system and do not account for the history of environmental changes that pre-dated the LGM but could have introduced long-lasting memory effects."*

*"The long timescale is chosen to avoid biases resulting from steady state assumptions and account for the possibility of memory effects under continuously varying climate and carbon cycle that could span multiple glacial cycles. Consequently, all carbon stores are achieved dynamically rather than being prescribed."*

- L84 : Please consider detailing what "partially" exactly means, especially if it helps make the case for dynamic sediments.

We will change the sentence to: *"Yet, shallow water carbonate burial was prescribed and POC burial not included in the simulations, which begs the question how the effect of the considered processes on glacial-interglacial atmospheric $CO_2$ and carbon isotopic ratios changes if the sediments are dynamically calculated"*

- L90-96. All the originalities of the scientific approach of this study should be justified. Here we have : - transient → justified by "to avoid biases resulting from steady state assumptions", even though it is unclear what those biases exactly are. I think that it is unclear what the precision "so that all carbon stores at the beginning of the last glacial cycle are achieved dynamically rather than being prescribed" brings to the table. You haven't said that the last glacial cycle is of specific interest and why ?
- 780 kyr long → not explicitly justified. Why several cycles? Why not shorter/longer?
- factorial simulations → justified by "to understand how various processes…"
- "with and without sediments" → justified by "to distinguish the role of interactive sediments"
- isotope-enabled → it is not said in the introduction that one strength of this study is that it will provide a multiproxy analysis, even though it is part of your main results (e.g. L16).

We will follow the reviewer's suggestions and revise the paragraph as follows:

*"Here we examine systematically how the transient built-up and dissolution of marine sediments on glacial-interglacial timescales affects the carbon cycle changes produced by the various processes suggested to be relevant on these timescales, a gap left by previous studies. Instead of searching for the most likely scenario that reconciles the vast proxy evidence, we attempt to gain a more complete process understanding and overview of the proxy-relevant signals that these processes cause in the presence of weathering-burial imbalances. With this goal, we extend factorial simulations of multiple simplified physical and biogeochemical forcings in a marine sediment and isotope-enabled intermediate complexity Earth system model over the last 780 kyr and compare the resulting carbon and carbon isotopic signals to reconstructions. The long timescale is chosen to avoid biases resulting from steady state assumptions and account for the possibility of memory effects under continuously varying climate and carbon cycle that could span multiple glacial cycles. Consequently, all carbon stores are achieved dynamically rather than being prescribed. We present two sets of simulations with and without interactive sediments to distinguish the role of interactive sediments in the carbon cycle changes caused by the tested forcings over reoccurring glacial cycles of the last 780 kyr."*

**3. Methods**

- It might be worth mentioning whether the model simulations include dynamic land-sea mask and bathymetry change (and hence varying ocean volume).

We will add the missing information: "The (pre-industrial) land-sea mask and bathymetry are fixed throughout the spin-up and simulations."

- L107-108: I don't understand. Is this to correct a specific bias?

Yes, this is a correction for a bias in air-sea carbon fluxes that cause mismatches with observed $\Delta^{14}C$. We will add this information to the manuscript:

*"The global mean sea-air gas exchange was then reduced by 19% to achieve agreement with pre-bomb testing radiocarbon distribution estimates and 20th century observations (Müller et al., 2008). This is a standard adjustment in Bern3D and accounts for the fact that $\Delta^{14}C$ in the surface ocean is overestimated by the gas transfer velocities calculated from wind speed."*

- L118-119: Is this compensation done uniformly at the surface of the ocean? It should also be recognized that weathering isn't explicitly represented in the model, but that the loss to sediments is compensated by an input flux which we assume could represent a source from weathering fluxes brought to the ocean by rivers, thus allowing for an equilibrium of whole-ocean inventories. In this respect, perhaps it is more relevant to speak of "terrestrial solute supply" as in L124.

We will follow the reviewer's advice and change the sentence to: "*At the end of this stage, the solute input flux required to balance sedimentary burial is diagnosed (Table S1) and kept constant for the rest of the spin up procedure and throughout our transient experiments*"

- L122: "in three stages, sequentially coupling all modules". Please consider briefly mentioning why. Also, it is never clearly said how many years the model was spun-up?

We will add both pieces of information: "for computational efficiency" and "The total length of the spin-up to this point was 72 kyr."

- L125-126: "is prescribed to balance loss to the lithosphere over 50 kyr". I am not sure I understand properly. When is the loss to sediment calculated? Is it a each step over a 50 kyr spin-up? Is it done once at the end of the 50 kyr? Why us the input flux kept constant thereafter and not diagnosed over each cycle or so?

We will revise the sentence to: "In the next step, the sediment module is coupled and terrestrial solute supply (phosphate, alkalinity, DIC, DI13C and Si) to the ocean is set to dynamically balance the loss through sedimentary burial for 50 kyr. At the end of this stage, the solute input flux required to balance sedimentary burial is diagnosed"

We chose to keep the solute flux constant because of the large uncertainty about actual solute flux changes over glacial cycles (Jeltsch-Thömmes et al., 2023). The important

feature for marine sedimentary changes is the difference between input and burial, which varies in our simulations and has an uncertain amplitude because of supply and burial rates.

One lesson from our study for the experiment design of future studies is to choose a different time window for diagnosing this flux, as suggested by the reviewer (lines 428-434).

*"The drift can therefore not be corrected for with a control simulation without forcing, because it only appears in the perturbed system. Instead, to avoid a drift, the experiment needs to start from an isotopically balanced geologic carbon cycle, which most commonly will require a long spin-up with a fully-coupled, open system, ideally over several glacial cycles especially when simulating large changes of the biological pump or marine carbonate system. We suggest that the size of the transient imbalance of the geologic carbon cycle, and thus the length of the required spin-up, could be minimized by balancing the geologic carbon cycle not for an interglacial state but for the mean burial fluxes over a full glacial cycle."*

- L133: It should be explained why your approach includes the design of "simplified forcings".

We will expand our explanation of the experiment design:

*"Data constraints on carbon cycle forcings are too sparse to know exact magnitudes and timings of the forcings that might have varied spatially and temporarily over the last eight glacial cycles. An inverse estimation of the forcings from the resulting proxy signals requires a different simulation ensemble and is beyond the scope of our study. Rather than trying to guess the most proxy consistent forcing amplitudes and patterns, we designed seven simplified forcings, each with one exemplary magnitude, to simulate the generic effects of processes that have been identified as glacial-interglacial carbon cycle drivers. Except for the orbital changes, which were calculated following Berger (1978); Berger and Loutre (1991) and the reconstructed $CO_2$, $N_2O$ and $CH_4$ curves (Loulergue et al., 2008; Joos and Spahni, 2008; Bereiter et al., 2015; Etminan et al., 2016), which we used to calculate the radiative forcing of greenhouse gas changes, the amplitudes of the forcings were set to cause noticeable $CO_2$ or circulation shifts, informed by previous studies (e.g. Tschumi et al., 2011; Menviel and Joos, 2012; Menviel et al., 2012; Jeltsch-Thömmes et al., 2019; Pöppelmeier et al., 2020). We produced timeseries of these forcings by defining a maximum forcing amplitude for the LGM, a minimum for the Holocene and then modulating this amplitude by reconstructed relative changes in the temporal evolution of either Antarctic ice core $\delta D$ (Jouzel et al., 2007) or benthic $\delta^{18}O$ (Lisiecki and Raymo, 2005) for each year (Fig. 1). The choice of the isotope record for calculating the instantaneous forcing depends on whether we expect the forcing to evolve synchronously with temperature like $\delta D$ or have a time lag similar to $\delta^{18}O$ (see section SI.5 for a discussion of the limitations). In all simulations, we prescribed the radiative effect of $CO_2$ in the atmosphere, so that all simulations have the same radiative forcing from greenhouse gases despite differences in simulated $CO_2$."*

- Table 1: (1) It is never explained why no PO4 simulation without sediments was performed. (2) It is not explained why the forcing values (e.g. -40%, -2.5 W/m², etc.) were chosen. (3) Note that the combination simulations PHY, BGC and ALL, as well as CO2T, have not been explained in the text before reading the Table. (4) It could be interesting for the reader to see the remineralization profiles of BASE compared to REMI.

(1) We will add an explanation: *"This is the only forcing that we did not apply to the model without interactive sediments because, while nutrients can be added to the surface ocean periodically, there is no simple way of artificially extracting nutrients from the ocean in return."*

(2) We will explain explicitly that the forcing magnitudes are chosen exemplarily, to achieve noticeable impacts, informed by previous studies (lines 151-166, see new text in the previous answer).

(3) We will change the order of text and table to introduce all experiments properly before Table 1 is shown.

(4) We will add a figure comparing the two profiles (Fig. S9).

- Fig. 1: I think that the legend should mention (1) the placement of each subpanel with indications (e.g. top, bottom, or panel numbers), (2) the abbreviations (e.g. RF), (3) that the grey background indicates MIS.

We will make the suggested changes.

- L143-144: "to achieve an older glacial deep ocean". Please consider mentioning right away that "younger deep water masses" are in disagreement in proxy reconstructions. It is mentioned but later, in L146.

Will be done.

- L148-149: I understand the rationale behind KGAS but considering the results (as I remember them now), it doesn't seem like this simulation bring much to the table…

We agree that the carbon cycle impacts are not large but we think that this is a result worth showing.

- L160: "adjust external alkalinity fluxes": is it the same as the "terrestrial solute supply" which is alternatively refered to as "weathering"? Also, I am curious as to how your model is able to effectively restore CO 2 variations: how are you prescribing alkalinity fluxes which results in the right CO 2 change, on a practical level?

We will add the requested additional information:

*"In addition we performed one run in which we let the model dynamically apply external alkalinity fluxes (in addition to the constant terrestrial solute supply applied in each simulation, see spin-up methodology) to restore the reconstructed atmospheric $CO_2$ curve (CO2T). In this simulation, the model evaluates the difference between the simulated and reconstructed $CO_2$ at each time step and adds or removes the marine alkalinity required to cause the necessary compensatory air-sea carbon flux from the surface ocean. Alkalinity changes, e.g. due to changes in shallow carbonate deposition or terrestrial weathering, are an effective lever for atmospheric $CO_2$ change (e.g. Brovkin et al., 2007), and this additional run shows the long-term changes in marine biochemistry if this was the dominant driver of glacial-interglacial atmospheric $CO_2$ change."*

- L164 and L165: Both of these sentences somehow feel out of place. Maybe you should explain the rationale behind a different CO 2 in the radiative code and its connection to sensitivity test 1.

These sentences were misplaced and we will remove them. We will also remove the sensitivity experiments from the manuscript.

- L169 : Why do drifts occur in these simulations specifically?

The shifts occur because of imbalances in terrestrial input and sedimentary burial during glacial phases, which cannot be undone during the relatively short interglacials. We will refer to the discussion of these drifts at the end of our results section when they are first mentioned in the method section.

- L178 : I feel like the interest of these simulations to determine non-linearities could have been justified earlier, in the introduction. As of now, these simulations feel like an afterthought.

We will move the description of our motivation for these runs higher up in the subsection.

- L186-187 : This also feels out of place. It could have been mentioned earlier in the Methods.

We will move the introduction of experiments with and without interactive sediments higher up in the subsection as suggested.

**4. Results**
- Figure 2: (1) The gray shading is barely visible in print format. (2) Please consider explaining or simplifying the data legend (I don't understand "cubic spline Bereiter 2kyr cop"). (3) I don't understand why some simulations are selected to be represented here (e.g. PO4, REMI) and not others (BGC, ALL, etc.). (4) It could be nice to add in a value of the different LGM drawdowns on this figure (as text annotations). (5) Figure 1 would fit better in subsection 4.1 than in the start of the results section, which otherwise contains clarifications of the approach of the study (L196-205).

We will move Fig. 2 into the subsection discussing $CO_2$ changes, addressing (5) of the reviewer's concern. This subsection also contains quantifications of the LGM-PI $CO_2$ differences (4).

We will make the suggested changes to the figure itself: (1) We will darken the gray shading, (2) We will simplify the legend. (3) We will reduce the displayed results to BASE, PHYS and BGC, because these are the runs that are discussed in the related paragraph

- L189: Note that differences "their timings" are not explained here but much later in the text.

By moving Fig. 2 and the corresponding text to the $CO_2$ subsection, it is also closer to the discussion of the differences in timings of $CO_2$ changes.

- L191-192: This seems to be true for all biogeochemical forcings except for LANDC though.

We will replace 'amplify' by 'alter', because the glacial-interglacial $CO_2$ difference is affected by interactive sediments under all biogeochemical forcings, just not in the same direction.

- Table 2: Many comments and questions. (1) Legend could be more precise: "summary model-data comparison" → "Quantified metrics of the carbon cycle according to reconstructions and our set of simulations with sediments". (2) Legend indicates "global preformed nutrient concentration" but the variable in the column is the change of [PO4, reg]. (3) Why are you only using carbonate ions reconstructions for the deep Pacific and not other basins? (4) Why are dates and anomalies (esp. "PI –18 ka") inconsistent with each other and with the legend ("LGM-Holocene") and in L.209? Are the same dates and anomalies used for simulations? (5) Why are the effect of individual forcings presented before/instead of the overall model-data comparison of different simulations? E.g. I can read that fLAND has an opposite effect on pCO2, but I only have a vague idea of the LGM CO2 of the LAND simulation (by adding up fLAND + fBASE). As such, this table isn't exactly a "summary of data- consistency" but rather a "summary of the model response to forcings", isn't it? Note that I understand the choice on focusing on the model response to a specific forcing, which is justified in L196-205, but it is surprising to start out the results with that, when the reader doesn't know yet which simulation(s) are consistent with proxy reconstructions.

We agree with the reviewer that Table 2 was not at an optimal position in the text. We will move it into the discussion, addressing the last concern.

(1) We will change the legend as suggested.

(2) It should be 'regenerated phosphate'. We will remove the discussion of regenerated phosphate to shorten the manuscript.

(3) We cannot compare our simulation results to all reconstructions because there are simply too many. Instead, we decided to focus on reconstructions in parameters and locations where the tested forcings show identifiable signals. Deep Pacific $CO_3^{2-}$ is interesting because its small variability over the last glacial cycle is seemingly at odds with the large global carbon cycle perturbation over the same time interval. This is discussed in the $CO_3^{2-}$ section, which will come before Table 2 is discussed. We hope this clarifies the selection of variables.

(4) We will use the same reference years where possible to calculate differences in proxies and simulations, however for some proxies they will be slightly different because of the temporal resolution and length of the underlying records. We will add a note of this to the table caption. We provided the PI-LGM difference for $CO_2$ and the LGM-PI differences for the other variables because that felt more intuitive. Since it turned out to be confusing, we will change it.

(5) We will add the suggested wording change to the table caption. The factorial results of the simulations are the focus of our discussion, so we prefer to show these in the table.

- L212: "non-linearities [are] still small compared to the effect of individual process" → This is

arguable, as they seem of the same order of magnitude as fBASE…

We agree with the reviewer and will change the wording, stating that non-linearities are smaller than the effects of individual biogeochemical forcings, while they are sometimes of similar magnitude as effects of physical forcings.

- L215-217: Should this be discussed in terms of processes (e.g. physical pump)?

Yes. We will move this paragraph to the discussion section, where we discuss the underlying processes in more detail.

-L220: "other processes": other than what? One could for example argue that the right processes are simulated but that the amplitude of their effect is wrong.

Yes, we will amend the sentence to say that the mismatch means that either some processes are missing (e.g. changed efficiency of biological pump, circulation changes, nutrient supply changes etc.) or that the simulated processes are not strong enough because of inadequate sensitivities to environmental changes.

- L220-222: Consider justifying the choice of assessing CO2, export production, biological pump, sedimentary fluxes and carbon isotopes. Can't an explicit link be made to what is said in the introduction? It is also unclear why we are examining these variables in this specific order.

We will add a sentence at the beginning of our results section to guide our results section:

"We discuss this question first by focusing directly on changes in the carbon stored as sedimentary organic and inorganic matter and changes in the benthic carbonate system, before studying their effects on four essential carbon cycle metrics: deep ocean $CO_2^{-3}$ atmospheric $CO_2$, marine DIC, and $\delta^{13}C$."

- Figure 3. (1) Again, by showing the effect of individual forcings, the reader doesn't really see e.g. the good match of simulation CO2T with the ice core data , and you are forced to add in L225 that the sum of fCO2T and fBASE is what is line with the represented reconstructions. It feels like a first graph should first show the simulation results in terms of glacial-interglacial CO2 variations as well, so that the reader clearly realises that adding up all the physical forcings in PHY is still not enough, whereas combinations of some biogeochemical forcings may produce a signal of the right amplitude. (2)
Legend should mention that this is the "effect of individual forcings on atmospheric CO2 changes". (3) Why is it only for the last 5 glacial cycles? (4) "selected factors" : on which basis are the effects of some factors represented and not others? e.g. the timing of LAND is mention in L228 but not represented. (5) Note that the difference between the two green-yellowish are not very visible on print format.

(1) We show the factorial results because they are essential for understanding the underlying processes. The goal of our simulations was not to match proxy records exactly but to test which signals in the records are caused by what processes. Therefore, we focus on comparing the factorial results rather than absolute results to the proxy records. In

reaction to the reviewer, we tried options to include absolute results in the main text but found it confusing to switch between absolute and factorial results, especially in a shortened manuscript. Hence, we decided to stick with showing the factorial results. The SI contain absolute results of the simulations (e.g. Figs S2-S7), to which we add a new figure with absolute carbon fluxes in each experiment (Fig. S10). We change the wording to remove the confusing reference to the absolute experiment output: *"By design, $CO_2$ restoring causes marine carbon uptake that fills the gap between dynamic atmospheric $CO_2$ changes in fBASE and reconstructions".*

(2) Will be done.

(3) We average over the last five deglaciations because those before the Mid-Brunhes event are less comparable since they had colder interglacials. We will state this in the caption.

(4) We will revisit the plotted results and make sure that all results discussed in the text are visualised.

(5) We will  remove the semi-transparent colours from the figure.

- L226: "produce the most consistent CO2 difference": with Fig. 3 only showing factorial effects, the reader can't see that and only gets the impression that fREMI, fPO4 and fPIPO "produce the largest CO2 difference", none of them being remotely close to producing a 80-100 ppm drawdown by themselves. Same for "is not necessary produced in our idealised simulations with simplified forcings (Fig. 3b)": the reader cannot directly visualise this affirmation.

We will change Fig. 2 to show absolute experiment results, and change the text to refer to Fig. 2, where appropriate, or focus on factorial rather than total results. We will also change the text to point out that the individual forcings smooth out transient features in the applied forcing due to memory effects in the sediments.

- L238-239: I think that the authors should elaborate on this. Why is this the case? Is this a good thing? What does this imply for models which do not consider imbalances? Alternatively, they could refer to a section in which this is discussed.

We will revise the discussion of deglacial $CO_2$ rises to focus more on the underlying processes:

*"The weathering-burial disequilibrium, which builds up over the glacial phase under these forcings, amplifies the deglacial $CO_2$ rise, particularly in $f REM I$ and $f P IP O$. In both cases, sedimentary accumulation of $CaCO_3$ spikes during deglaciation, due to increased $CaCO_3$ export as the forcings wane (Fig. 2). The corresponding ALK reduction expels more $CO_2$ from the surface ocean into the atmosphere. In the case of $f REM I$, this is further enhanced by a reduction in sedimentary POC accumulation during the deglaciation, which reduces the C loss to the sediments. In both cases the sedimentary processes that amplify the deglacial $CO_2$ rise also reduce its speed and smooth out transient features of the $\delta D$ record which are translated into transient atmospheric $CO_2$ changes in simulations without interactive sediments (Fig 6). These time lags are caused by the strengthened export production, which*

*counteracts C degassing, and a large build-up of alkalinity and DIC during the glacial phase (amplified by interactive sediments, Fig. S17) which is only gradually reduced by enhanced CaCO₃ burial during deglaciations (Fig. S18). If instead export production and sedimentary C accumulation decrease during the deglaciations due to increased nutrient limitation (ƒ P O4), the C previously incorporated into biogenic matter is outgassed from the surface ocean and no lag between CO₂ rise and the forcing emerges. Weathering-burial imbalances have a smaller effect on circulation-driven deglacial CO₂ degassing (ƒ SOW I, ƒ AERO), regarding both amplitude and timing. However, CO₂ also lags temperature in ƒ AERO (with and without interactive sediments), due to the hysteresis of the AMOC. Enhanced Southern Ocean wind stress (ƒ SOW I) is the only forcing in our simulation set that is able to create fast, transient CO₂ releases despite weathering-burial imbalances. In all simulations except LAND, the lowest CO₂ values occur during the coldest interval of glacial cycles, the glacial maxima (Fig. 5, S16). In all simulations in which the deglacial CO₂ rise lags that of temperature, CO₂ keeps rising throughout the Holocene"*

- L228: You could refer to Fig. 2.

Will be done.

- L259: You could consider briefly explaining why POC dissolution (and also increased alkalinity) raise atmospheric CO2.

We will add the suggested explanations as follows:

*"During deglaciations, sedimentary POC deposited during glacials is remineralized, which raises DIC and further reduces ALK, both contributing to enhanced CO₂ outgassing. We explore the forcing-specific differences in more detail by focusing exemplarily on the last deglaciation."*

- L261-262: "good match with various features". This is rather vague. Do you mean in terms of amplitude? Timing?

We will remove this expression and now discuss explicitly which forcings produce similarities in timing and which in amplitude.

- L263-265: You could refer back -to the Fig. where this can be observed.

Will be added.

- L266 : Please elaborate. Are you implying "and so we can rely on multiproxy analysis to disentangle the different mechanisms in play"?

Multi-proxy analysis is necessary to understand carbon cycle dynamics but we do not have sufficient proxy data to actually disentangle all mechanisms. Hence, we will be more cautious with our formulation now and "highlight the need for considering multiple carbon reservoirs and multi-proxy analyses to more robustly quantify global carbon cycle changes during glacial cycles." (lines 17-18)

- L280 : "This mirrors". It it unclear what "this" refers to, with REMI described in L278 but PO4 described in L279.

This sentence is part of the text that we will remove to shorten and focus the manuscript.

- L283-285: Does this mean that over a long period, we have a rough equilibrium, with total fluxes at zero, and the carbon fluxes which goes in the Southern Ocean compensating the carbon outgassed elsewhere?

Yes, net carbon uptake or release by the ocean occurs through regional imbalances between in- and outgassing which are small compared to the overall air-sea gas exchange.

- L292: Doesn't the fact that weathering is kept constant also causes imbalances?

Yes, the imbalances that appear in the simulations depend on the experimental design. Here, we meant to say that sedimentary burial is the only dynamic part in our experiments that can change the weathering-burial balance. Hence, imbalances can only occur when sedimentary burial changes. We will rephrase the sentence for clarity:

*"Secondly, sedimentary mass accumulation, dissolution, and remineralization rates control sedimentary burial, the only permanent sink for carbon and nutrients in our simulations and the only mechanism by which environmental change can create imbalances with the prescribed constant solute flux from land."*

- L294: I don't understand the causality link ('consequently').

We will reformulate our point:

*"Carbon fluxes from the sediments directly affect the ocean, but not the atmosphere, which causes different amplitudes in the simulated DIC and atmospheric $CO_2$ changes and different timings of carbon accumulation in ocean and atmosphere"*

- L301-302: I am not sure I understand properly why the DIC/CO2 lag in simulations with sediments causes uncertainties in DIC reconstructions. As for "do not necessarily imply a comparable CO2 drawdown", isn't this obvious already when considering the carbon reservoirs on land?

Yes, land carbon reservoirs can cause differences between DIC and $CO_2$ changes, but they are often considered in glacial carbon budgets. We underlined this point here because carbon exchange with the sediments can decouple DIC changes from atmospheric $CO_2$ changes (even causing low DIC and low $CO_2$ at the LGM), a possibility which is rarely considered when reconstructed DIC changes are quantitatively converted into $CO_2$ changes e.g. in Yu et al., 2010, Farmer et al. 2019, Vollmer et al., 2022.

We will move this sentence to the discussion section, where we also address the proxy-consistency of simulations with low DIC and low atmospheric $CO_2$ during the LGM:

"A close relationship between DIC and $\Delta^{14}C(DIC)$ is found in modern deep ocean waters and this relationship has been used to reconstruct past DIC changes from radiocarbon reconstructions (Sarnthein et al., 2013). Sedimentary carbon fluxes can de-couple deep ocean $\Delta^{14}C$ from DIC (Dinauer et al., 2020) and change DIC without altering sea-air carbon transfer, meaning that DIC changes do not necessarily imply a comparable $CO_2$ change in the atmosphere. In all of our simulations with interactive sediments, the DIC inventory change over a glacial cycles is larger than the simultaneous atmospheric $CO_2$ inventory perturbation because of changes in carbon reservoirs in sediments and weathering-burial imbalances. Changes in the simulated sedimentary burial fluxes result in net transfers of up to 2000 PgC between the carbon pools of the ocean and sediments throughout a glacial cycle, while the net loss of atmospheric C to reproduce the reconstructed glacial $CO_2$ is roughly 200 PgC (Sigman and Boyle, 2000; Yu et al., 2010), and the net loss of terrestrial C is on the order of 500-1000 PgC (Jeltsch-Thömmes et al., 2019). The carbon cycle impact of glacial cycles was thus likely larger in the ocean than in the atmosphere (Roth et al., 2014; Buchanan et al., 2016), due to changes in sedimentary carbon storage. In some of our simulations, large DIC changes are produced by big sustained weathering-burial imbalances during glacials that cannot be compensated during the relatively short deglaciations and cause interglacial carbonate preservation patterns that are not consistent with observations (Fig. S11, S12). While this suggests that such a scenario is unrealistic, it does not generically preclude the possibility of large transient weathering-burial imbalances. Testing a wider range of forcing magnitudes and combinations with the same model but different set-up, Jeltsch-Thömmes et al. (2019) (the DIC results of which are published in the Appendix of Morée et al. (2021)) found a larger DIC change between the pre-industrial and LGM than simulated here (3900±550 GtC compared to a maximum of 1100±300 GtC in Fig. 7) that is consistent with carbonate system proxy constraints. Combinations of the tested forcings thus allow for larger transient weathering-burial imbalances than produced by our simulation ensemble that can still be reconciled with carbonate system proxies. Some of the tested forcings also show lower glacial than inter-glacial DIC ($fPO4$, $fCO2T$) showing that $CO_2$ removal from the atmosphere in theory does not need to result in increased DIC in the ocean. Instead, these biogeochemical forcings cause sedimentary changes that can store large amounts of carbon in inorganic and organic sedimentary matter. Kemppinen et al. (2019) and Jeltsch-Thömmes et al. (2019) previously showed and discussed the possibility of a negative glacial DIC anomaly due to increased sedimentary storage. As found by Jeltsch-Thömmes et al. (2019), organic carbon burial extensive enough to cause a negative glacial DIC anomaly as due to $fPO4$, produces large $\delta^{13}C$ signals of opposite sign than reconstructed, and thus seem unlikely. In the study by Jeltsch-Thömmes et al. (2019), a negative glacial DIC anomaly due to alkalinity-driven $CaCO_3$ accumulation is also inconsistent with the proxy record of the last 25 kyr. Consistently, we find that reconstructed deep Pacific $[CO_3^{2-}]$ changes make a large-scale alkalinity-driven ($fCO2T$) glacial $CaCO_3$ accumulation, which reduces atmospheric $CO_2$ while also reducing DIC, unlikely because it causes larger deep Pacific $[CO_3^{2-}]$ changes than reconstructed over the last deglaciation (Table 2). The isotopic signal of such large $CaCO_3$ deposition, however, is smaller than that of POC burial changes and could more likely be overprinted by other processes (e.g. terrestrial carbon release and export production changes) to yield proxy-consistent evolutions (Table 2)."

- L305: "additional proxy data". "additional" with respect to which data? I remember of no proxy data which was used in this section.

We meant proxies in addition to reconstructed $CO_2$ changes, by which simulations with more glacial than interglacial DIC cannot be differentiated from simulations with less glacial than interglacial DIC. We will remove this sentence to avoid confusion and for brevity.

- L317-318: It is not obvious to me when observing Fig. 6 that AERO simulation has the best model-data agreement, and I expect it is because it is fAERO and fBASE (to add in mind then) which are represented again… Same for L324 "which is consistent with the reconstructions" (there seems to be many black points in these regions as well)

We agree that the global comparison of the simulated characteristics to proxy constraints on export production changes requires a more in-depth discussion. However, since export production changes are less affected by dynamic sediments, we will remove this section from the manuscript to shorten it by focussing on the effects of sediments.

- L323: This had me wondering where the extra PO4 was put in the PO4 simulation. Is it uniformly distributed at the surface ocean?

Yes, we will spell this out more clearly in the description of the experiment design.

- L417: What do you mean by "sensitivities"?

We will change the wording to "Since the processes that affect $\delta^{13}C_{CO2}$ and $\delta^{13}C_{DIC}$ are different, and $\delta^{13}C_{DIC}$ varies between ocean basins, the forcings which best reproduce reconstructed evolution of $\delta^{13}C$ also vary between atmosphere and ocean, and specific water masses (Oliver et al., 2010)."

- L422: Note that this is the first occurrence of the formulation "alkalinity nudging" for simulation CO2T. You might want to consider to use consistent terminology in the Methods/Results.

Yes, this was wrong terminology. We will correct it to "$CO_2$-restoring alkalinity fluxes".

- L435-437: According to model simulations or data?

We change the wording.

- Figure 11: Please consider providing quantifications of the model-data agreement such as a RMSE (here and in other instances).

We will revise Fig. 11 and provide RMSE to quantify the spatial error.

- L484: REMI results are commented but not shown in Fig. 11.

We will add the results for all individual simulations to the SI.

- L524: It is unclear which model-data comparison "previous" is refering to.

Yes, this is actually referring just to the shown simulation results, not the model-data comparisons. We will correct this.

- L553: "the bias in simulated global carbon fluxes and reservoir size changes". Why bias do you mean? Or do you mean "biases" in a general sense?

Yes, we meant the plural, in a general sense. We will change the wording to "would substantially alter the simulated carbon fluxes" for clarity.

- L558-559: as well as changes in alkalinity input fluxes, right?

Yes, we will change the wording higher up from "carbon fluxes" to "terrestrial solute fluxes".

- L565: Could these imbalances in weathering-burial (in terms of carbon, nutrients, alkalinity) be quantified or represented somewhere? Maybe this could lead to a more affirmative sentence than "it seems likely that".

We will add the ranges of weathering-burial imbalances at the end of our simulations.

The 'it seems likely' statement refers to the real world, and we cannot be more affirmative based on our simplified and factorial simulations.

**5. Discussion**
- L617: Why "in addition" and not "such as"?

We wrote 'in addition' because REMI was implicitly discussed in the section on the more efficient biological pump. We will rewrite parts of the discussion and the sentence will not appear anymore in its previous form.

- L635: "Our simulations with increase organic carbon burial". Consider citing the simulation names again.

Will be done.

- L683: Please explain how. Same for L645.

We will expand both sentences:

"Tschumi et al. (2011) demonstrated in their steady state experiments that increased organic nutrient burial enhances nutrient limitation on export production and reduces $CaCO_3$ export, which increases surface alkalinity and amplifies the $CO_2$ drawdown caused by the increased burial of organic carbon."

" Finally, sedimentary organic carbon oxidation can also regulate marine alkalinity by affecting sedimentary $CaCO_3$ dissolution (Emerson and Bender, 1981; Sigman and Boyle, 2000)."

- L646: "especially those which reproduce the reconstructed increase of organic carbon burial during glacial maxima" → Looking again at Fig. 10, I would say none of the simulations do that…

No simulation matches the reconstruction in amplitude but some match in sign, which we focus on here. We will reword the sentence to clarify.

- L651: "big sedimentary changes": Does this exclude the fraction lost to inert sediments which is compensated by weathering inputs?

We will rephrase this and write "large, sustained weathering-burial imbalances".

- L652: You could refer to a figure again.

Will be done.

- L655-656: But didn't Morée et al. perform equilibrium runs? (I could be mistaken, I haven't checked the paper.)

Yes, the main text focuses on equilibrium runs but in the Appendix they present DIC results of transient simulations. We will expand the citation to clarify that we refer to the Appendix.

**6. Conclusion**
- L673: "other processes must have operated". It is unclear what "other processes" are refering to. Do you mean physical processes (e.g. circulation changes)? Do you mean "compensating processes"?

Yes, circulation changes, shallow carbonate deposition or terrestrial solute flux changes could be such processes. We will use the suggested term 'compensating processes' and provide examples.

- L678: This was not really shown in the text previously.

Yes, we discussed in the DIC section how DIC changes are not linearly related to $CO_2$ changes across the different forcings. We will rephrase the sentence to clarify.

- L680: Do you mean "ocean carbon inventory" or "ocean inorganic carbon inventory"?

Yes, we will use 'DIC' instead.

- L691-692: And so, what are the perspectives which can be infered? For example, would you be able to provide recommendation for less computationaly-efficient models which cannot afford to run 200 Kyr?

We do not have a technical solution for this, other than minimising the weathering-burial imbalance, with regard to carbon and carbon isotopes. Still, however, spinning up $\delta^{13}C$ in an open system requires very long simulation times. In the absence of strategies to achieve

sufficient spin up times, we expect $\delta^{13}C$ drifts to be present, which at least would need to be corrected for with control simulations. We will add this text to address this:

*" The magnitude of the initial imbalance in the geologic carbon cycle, and hence isotopic drift, depended on the simulated forcing and was largest in simulations REMI, PIPO and CO2T. Importantly, the drift is a result of perturbing the sediment-weathering balance. The drift can therefore not be corrected for with a control simulation without forcing, because it only appears in the perturbed system. Instead, to avoid a drift, the experiment needs to start from an isotopically balanced geologic carbon cycle, which most commonly will require a long spin-up with a fully-coupled, open system, ideally over several glacial cycles especially when simulating large changes of the biological pump or marine carbonate system. We suggest that the size of the transient imbalance of the geologic carbon cycle, and thus the length of the required spin-up, could be minimized by balancing the geologic carbon cycle not for an interglacial state but for the mean burial fluxes over a full glacial cycle."*

**Technical/small comments**
- L2 : Consider putting "atmosphere" last in the list.

Will be done.

- L2 : "… preserved biogeochemical evidence" → "… preserved indirect biogeochemical evidence"?

Will be done.

- L7 : "uncertainty" → "knowledge gap"

Will be done.

- L16 : Consider removing "likely".

Will be done.

- L21 : "Earth's carbon cycle" → "the Earth's carbon cycle" ?

Will be done.

- L32 : Consider quoting Kohfeld and Ridgwell, 2009.

Will be done.

- L33 : "last glacial maxiumum" → "Last Glacial Maximum"

Will be done.

- L37 : "added CO2 back" → "tend to counteract this effect by stimulating CO2 outgassing"

Will be done.

- L42 : "as well as increased nutrient supply" → "as well as increased biological pump from increased nutrient supply"?

We will add "increased export production due to increased nutrient supply".

- L81 : "combinations" or "combination" ?

We will change the sentence to use 'a combination'.

- L83 : Why isn't "shallow water carbonate burial" not included in the list of biogeochemical processes with the others?

We will add 'shallow water carbonate burial' to the list of the other biogeochemical forcings.

- L87 : "models" or "model"?

We will choose 'a model' as suggested.

- L98 : "The Bern 3D 2.0 model"?

We use Bern 3D 2.0s because it includes sediments.

- L100 : "41x40" → why not provide the resolution in °?
The grid is irregular, we will add the range of resolutions in °.

- L130 : Please consider avoiding mentioning the simulation names before they are properly defined.

We prefer to keep all information on the spin-up procedure in one paragraph but will add a note that the experiment names will be explained in the next subsection.

- L149 : "Finally" is inadequate as you are not describing the final simulation, merely the last simulation with different physical forcing.

We will replace 'finally' with 'next'.

- L152 : "added" → "mimicked", as it is not this terrestrial sink/source is not explicitly resolved by the Model.

We will use the suggested term.

- L210 and L211 : "three" → "four"

Will be done.

- L257 : "under" → I think you mean "subsequently to", since forcings (e.g. PO4 supply) would be close to 0 during interglacials and we are seeing a memory effect.

Yes, we will use the suggested wording.

- L261 : "processes" → "forcings"

Will be done.

- L269 : typo "productivity"

Will be corrected.

- L270 : "played" → "plays"

Will be done.

- L275 : I may be wrong, but I think "towards" indicates a spatial direction (and may not be adequate for a time direction). Same for L410.

We will replace 'towards' in both cases.

- L278 : "lags behind" typo

Will be done.

- L284 : "40 °S" typo

Will be done.

- L303 : "marine carbon storage" → I think you mean DIC, to exclude organic carbon.

Will be done.

- L315 : invert "the LGM and the Holocene"

Will be removed from the new manuscript version.

- L342 : typo citep not citet

Will be removed from the new manuscript version.

- Figure 10 : Gray shading is barely visible in print format.

We will darken the shadings in all figures.

- L465 : typo "reproduce"

Will be done.

- L467 : typo "increased"

Will be done.

- Fig. 12 : Dotted light green-yellowish line is barely visible in print format.

We will thicken the lines to increase their visibility

- L516 : typo citet no citep

Will be done.

- L550 : "the real processes they represent" → "mimick"

Will be done.

- L661 : missing preposition

Will be done.

- L663 : typo "ice sheets"

Will be done.

- L686 : CO 32-

Will be done.

References

Farmer, J.R., Hönisch, B., Haynes, L.L., Kroon, D., Jung, S., Ford, H.L., Raymo, M.E., Jaume-Seguí, M., Bell, D.B., Goldstein, S.L. and Pena, L.D., 2019. Deep Atlantic Ocean carbon storage and the rise of 100,000-year glacial cycles. *Nature Geoscience*, *12*(5), pp.355-360.

Vollmer, T.D., Ito, T. and Lynch-Stieglitz, J., 2022. Proxy-Based Preformed Phosphate Estimates Point to Increased Biological Pump Efficiency as Primary Cause of Last Glacial Maximum CO2 Drawdown. *Paleoceanography and Paleoclimatology*, *37*(11), p.e2021PA004339.

Yu, J., Broecker, W.S., Elderfield, H., Jin, Z., McManus, J. and Zhang, F., 2010. Loss of carbon from the deep sea since the Last Glacial Maximum. *Science*, *330*(6007), pp.1084-1087.

---

## Referee Report (RR1)

I thank the authors for their great and diligent work in revising the paper while taking into account our comments. They did not hesitate to trim down the text quite drastically. I think the manuscript is greatly improved as it makes for a smoother, more focused and therefore more impactful read. I only have minor comments left, most of which related to subideal language.

**General comments**

**1.** L20, end of result section, end of conclusion : All of these critical parts of the paper end on a rather anecdotic statement. The spin-up strategy is rather a minor part of the study, with (arguably) minor implications. I think the authors could be less humble and find a stronger statement to end on an meaningful note to the community.

**2.** Statements regarding terrestrial carbon are rather cryptic :
- L51-52 : 'carbon can also be transferred on land'. It is unclear what is the point of this argument, since this study is not running simulations with a land carbon model.
- L86 : '(atmosphere-ocean only)'. Same, I don't understand the emphasis on previous modelling studies using atmosphere-ocean system, as many models include a land carbon model.
- L388 : I am confused by the mention of the '4-box land biosphere'.

**3.** Page 8 : The choices of all physical forcings (+ CO2T) is justified by a clear link to the processes identified as potential contributors to glacial-interglacial CO2 changes. This is not the case of the biogeochemical forcings (PO4, REMI, PIPO). A short but explicit link/scientific reasoning would be welcomed so that this part doesn't read as a list ('we tried that… and that, and that…') without the reader knowing why you tested these specific things.

**4.** C, ALK, DIC : Please check that all abreviations are defined at first instance.

**Specific comments**

L4 : 'proved the potential'. In my mind (non-native English speaker there), the concept of 'proof' is strong and contrast with 'potential'. I would use a more neutral verb, like 'demonstrate'.

L5 : Please clarify what 'they' refers to. 'These processes' ? 'These glacial conditions'?

L10-11 : This sentence is long and convoluted. First, there is a repetition of 'due to these different forcings' and 'resulting from these forcings' which feels unnecessary. Second, the following proposition 'and the associated isotopic shifts' is missing a verb, and it is unclear whether it is supposed to echo 'assessing' or 'gaining a better understanding of'. Third, the same idea ('transient', 'continuously perturbed' and 'non-equilibrium glacial cycles') is repeated three times, which I think is more than enough for the reader to get it.

L44-48 : First, this sentence is too long and hard to follow. Second, why is nitrate not mentioned in addition to phosphate ? Third, the construction of the last part (with 'counteract the effect') makes it difficult to understand the exact effect of 'changes in Southern Ocean dynamics'. It is unclear in which direction these variables are varying, so it could be specified what 'changes' we are talking about, as well as whether the 'effect of colder temperatures' is enhancing or dampening export production.

L51 : 'could have been sequestered in the water column'. Yes, but this is also true of a closed atmosphere-ocean system. I recommend using the phrasing 'in marine sediments as well as/in addition to DIC in the water column' to clarify.

L63 : 'previous model simulations, that included POC burial, showed that interactive sediments'. Too many commas (very german), a smoother phrasing could be considered ('previous model simulations showed that interactive sediments including POC burial…').

L81 : 'also'. I don't see the first argument to which 'also' implicitly refers to. This sentence reads to me as a precision of the previous one.

L97-98 : the in-text question feels a bit convoluted to me. What about : 'which begs the question : what are the effects of the considered processes on glacial-interglacial atmospheric CO2 and carbon isotopic ratios when the sediments are dynically calculated?'

L183-186 : This sentence is long and convoluted. The verb 'test' is used twice (but with different things following : AMOC changes versus radiative changes resulting to dust changes) and the Adloff 2024 paper is quoted twice. I think that all mentions of the resulting circulation changes could be kept for the next sentence to reduce the weight of this one. It could also be (very briefly) explained why the radiative changes have such an effect on the AMOC, and whether this is a full collapse (as most readers won't look for that information in the quoted paper).

L194 : Why isn't this simulation named 'NUT' if the forcing indirectly encompass the effects of different nutrient inputs? (I may have not understood this specific experimental design well.)

L231 : It could be mentioned at the first occurence of 'weathering input fluxes' that this is what you are calling the 'terrestrial solute supply' made to compensate loss to sedimentary burial in the following. A simple '(thereafter named weathering input)' in Section 2.2 would also do the trick.

Fig. 2 : What does the Delta mean ? Maybe the legend is not precise enough, for I was confused in the direction of the signals when reading through the next paragraph. Also, Fig. S10 look identical to Fig. 2 to me, so looking for absolute changes did not help.

L241-242 : It could be briefly mentioned why we are observing these variations.

L249 : This sentence looks unnecessary to me.

L255 : 'occurred simultaneously in reality'. The phrasing is not ideal, as we are talking of idealized forcings which did not 'occur in reality'.

L265 : I am confused as to the interpretation of the effects of physical forcings. It is said that 'Reconstructions […] show that burial rates decreased […] during glacial inception […]'. fSOWI shows constant CaCO3 balance on Fig. 2, so why is it said that 'Physical forcings do not affect burial rates during glacial inception', but later that 'However, the physical forcings fail to decrease burial rates during MIS3 and MIS2'?

Fig. 4 legend : It is a bit unclear why the Qin et al (2018) data in particular was chosen for reference. Is this the only core which spans a long enough time interval? Why are you showing the time series for the deep Pacific only (and not e.g. deep Atlantic, especially after the L305-306 mention)?

L319-320 : Having mentions to different periods ('during interglacials', 'during glacial phases') in the same sentence is a bit confusing.

L375 : It feels like a verb is missing. Do you mean 'required to compensate the prescribed solute fluxes'?

L382 : 'long-term trend of lower atmospheric d13C during the Eemian than the Holocene'. Why use this convoluted phrasing and not simply 'long-term trend of increasing atmospheric d13C during the last glacial cycle'?

Fig. 8 : Since fLAND causes an increase and fPO4 a decrease, I would be curious to see the absolute effect of the BGC simulation, cumulating the two.

L563 : 'the buffering impact of this perturbation on the deglacial carbon re-organization'. This phrasing doesn't read easily to me, perhaps because it is unclear what 'this perturbation' refers to.

**Technical comments**

L7 : 'of' → 'using' or 'with'

L68 : 'extents' → 'amplitude'

L95-96 : misplaced (

L145 : 'our results section' → using section numbering is better

L149 : Does that mean Section 5 of SI?

Fig 1 legend : typo 'gasses'

L172 and 175 : please use insecable spaces so that the units appear on the same line.

Page 8 : wall of text. I recommend a line jump between the description of physical and biogeochemical forcings. I also think that replacing 'Next, we tested...' with some type of numbering would make it easier to follow (e.g. 'A third simulation tested…', 'Thirdly, ...')

L191 : missing comma

L228 : 'sediments' → 'sediment fluxes'?

Fig. S9 : typo 'interglacial'

L257 : incomplete sentence without verb or majuscule.

L267 : 'in Fig. 2'. In this instance (and the whole paragraph), it would be quicker for the reader to find where to look at on the graph if the top panels where numbered and you could refer to 'Fig. 2a,b,c,d'.

Fig. 3 legend : References look misplaced to me. They would be better placed after 'reconstructed', with a mention of which is for POC, and which is for CaCO3.

Fig. S12 and Table S2 : It is unclear what simulation CACO is.

L298 and L302 : extra comma

L304 : unfinished sentence ('but').

L316 : 'causes' → 'cause'

L318 and L443 : typo 'biogeochemical'

Fig. 6 : Since there are mentions in the text of the range of effects in GtC, it could be helpful if a second axis in GtC (on top of the one in ppm) is added. Also, the legend in c) could add the references to the black lines.

L333 : 'Fig. 6' → 'Fig. 6c'

L385 : 'size' → 'amplitude'

L344 : typo majuscule

L392 : typo extra space

L386 : 'geologic' → 'geological'?

L458 : typo 'deglaciations'

L514 : missing space

L544 : 'unlikely' → 'which is unlikely'

L553 : missing comma

---

## Author Response (AR2)

Point-by-point response to comments by anonymous Referee #1

I thank the authors for their great and diligent work in revising the paper while taking into account our comments. They did not hesitate to trim down the text quite drastically. I think the manuscript is greatly improved as it makes for a smoother, more focused and therefore more impactful read. I only have minor comments left, most of which related to subideal language.

We thank the reviewer for their positive evaluation of our revisions and the additional comments.

General comments
1. L20, end of result section, end of conclusion : All of these critical parts of the paper end on a rather anecdotic statement. The spin-up strategy is rather a minor part of the study, with (arguably) minor implications. I think the authors could be less humble and find a stronger statement to end on an meaningful note to the community.

We agree that the discussion of the spin up strategy is not the main outcome of the study, so we re-arranged the passages to mention this before the other outcomes. However, we consider the long isotopic drifts in our simulations an important outcome with large implications for the discussion of how to set up and interpret isotope-enabled model simulations as well as d13C records. We added a sentence to our conclusions to express this:

" These results have implications for model experiment design and the interpretation of $\delta$13C proxy data: We showed that the long timescales of ocean-sediment interactions and the weathering burial cycle pose substantial challenges for model spin up because imbalances in the geologic carbon cycle can cause isotopic drifts at the beginning of simulations and which are not present in a control run. Depending on the initial isotopic imbalance, it takes up to 200 kyr for the drift to subside and the signal of the applied forcing to dominate the simulated transient $\delta$13C changes. Further studies are needed to test whether $\delta$13C can be spun up in more computationally-expensive models by combining them with lower-complexity models. In the absence of such a spin up strategy, open system simulations of glacial $\delta$13C are likely strongly affected by these initial drifts severely hampering interpretation of results. These long adjustment timescales also pose challenges for separating long-term from short-term signals in the proxy records."

2. Statements regarding terrestrial carbon are rather cryptic :
- L51-52 : 'carbon can also be transferred on land'. It is unclear what is the point of this argument, since this study is not running simulations with a land carbon model.

We made this statement in the introduction for completeness because increased land carbon storage has been discussed as a process lowering glacial atmospheric CO2. Here we indeed do not test this hypothesis and only consider the land as a possible C source during glacial phases. We therefore removed this reference to avoid confusion.

- L86 : '(atmosphere-ocean only)'. Same, I don't understand the emphasis on previous modelling studies using atmosphere-ocean system, as many models include a land carbon model.

Our intention here was to point out the lack of dynamic weathering-sediment burial imbalances in many simulations, not comment on the inclusion or exclusion of terrestrial carbon dynamics. We revised our formulation and now mention explicitly that we address the issue of sediment dynamics.

- L388 : I am confused by the mention of the '4-box land biosphere'.

This was a misplaced technical note specific to our model. We removed it.

3. Page 8 : The choices of all physical forcings (+ CO2T) is justified by a clear link to the processes identified as potential contributors to glacial-interglacial CO2 changes. This is not the case of the biogeochemical forcings (PO4, REMI, PIPO). A short but explicit link/scientific reasoning would be welcomed so that this part doesn't read as a list ('we tried that… and that, and that…') without the reader knowing why you tested these specific things.

We added brief reasonings to each forcing as suggested.

4. C, ALK, DIC : Please check that all abreviations are defined at first instance.

Done

Specific comments

L4 : 'proved the potential'. In my mind (non-native English speaker there), the concept of 'proof' is strong and contrast with 'potential'. I would use a more neutral verb, like 'demonstrate'.

Done

L5 : Please clarify what 'they' refers to. 'These processes' ? 'These glacial conditions'?

Done

L10-11 : This sentence is long and convoluted. First, there is a repetition of 'due to these different forcings' and 'resulting from these forcings' which feels unnecessary. Second, the following proposition 'and the associated isotopic shifts' is missing a verb, and it is unclear whether it is supposed to echo 'assessing' or 'gaining a better understanding of'. Third, the same idea ('transient', 'continuously perturbed' and 'non-equilibrium glacial cycles') is repeated three times, which I think is more than enough for the reader to get it.

We simplified this passage by deleting repetitions.

L44-48 : First, this sentence is too long and hard to follow. Second, why is nitrate not mentioned in addition to phosphate ? Third, the construction of the last part (with 'counteract the effect') makes it difficult to understand the exact effect of 'changes in Southern Ocean dynamics'. It is unclear in which direction these variables are varying, so it could be specified what 'changes' we are talking about, as well as whether the 'effect of colder temperatures' is enhancing or dampening export production.

We split the sentence into two, added a mention of N, and clarified the end.

L51 : 'could have been sequestered in the water column'. Yes, but this is also true of a closed atmosphere-ocean system. I recommend using the phrasing 'in marine sediments as well as/in addition to DIC in the water column' to clarify.

Done as suggested

L63 : 'previous model simulations, that included POC burial, showed that interactive sediments'. Too many commas (very german), a smoother phrasing could be considered ('previous model simulations showed that interactive sediments including POC burial…').

Done as suggested

L81 : 'also'. I don't see the first argument to which 'also' implicitly refers to. This sentence reads to me as a precision of the previous one.

We deleted 'also'.

L97-98 : the in-text question feels a bit convoluted to me. What about : 'which begs the question : what are the effects of the considered processes on glacial-interglacial atmospheric CO2 and carbon isotopic ratios when the sediments are dynically calculated?'

Done as suggested.

L183-186 : This sentence is long and convoluted. The verb 'test' is used twice (but with different things following : AMOC changes versus radiative changes resulting to dust changes) and the Adloff 2024 paper is quoted twice. I think that all mentions of the resulting circulation changes could be kept for the next sentence to reduce the weight of this one. It could also be (very briefly) explained why the radiative changes have such an effect on the AMOC, and whether this is a full collapse (as most readers won't look for that information in the quoted paper).

We simplified this passage.

L194 : Why isn't this simulation named 'NUT' if the forcing indirectly encompass the effects of different nutrient inputs? (I may have not understood this specific experimental design well.)

This is a good point. It is called PO4 because technically we remove nutrient limitation by adding phosphate, the only export-limiting nutrient in our model set-up. We now clarified this in the text.

L231 : It could be mentioned at the first occurence of 'weathering input fluxes' that this is what you are calling the 'terrestrial solute supply' made to compensate loss to sedimentary burial in the following. A simple '(thereafter named weathering input)' in Section 2.2 would also do the trick.

Done as suggested.

Fig. 2 : What does the Delta mean ? Maybe the legend is not precise enough, for I was confused in the direction of the signals when reading through the next paragraph. Also, Fig. S10 look identical to Fig. 2 to me, so looking for absolute changes did not help.

We revised the figure caption of Fig. 2 and the following text. Differences between Fig. 2 and Fig. S10 are small because they only differ in whether the small effects of the standard forcing in BASE are subtracted or not.

L241-242 : It could be briefly mentioned why we are observing these variations.

We added this info to the text.

L249 : This sentence looks unnecessary to me.

We removed it.

L255 : 'occurred simultaneously in reality'. The phrasing is not ideal, as we are talking of idealized forcings which did not 'occur in reality'.

We now rephrased this sentence.

L265 : I am confused as to the interpretation of the effects of physical forcings. It is said that 'Reconstructions […] show that burial rates decreased […] during glacial inception […]'. fSOWI shows constant CaCO3 balance on Fig. 2, so why is it said that 'Physical forcings do not affect burial rates during glacial inception', but later that 'However, the physical forcings fail to decrease burial rates during MIS3 and MIS2'?

Our formulation was wrong. The reconstructions show no global CaCO3 burial change during glacial inception (before MIS3), which is reproduced by the physical forcings. However, CaCO3 burial was lower in the reconstructions during MIS3 and MIS2, which is not reproduced by these simulations. We adjusted the text to reflect this.

Fig. 4 legend : It is a bit unclear why the Qin et al (2018) data in particular was chosen for reference. Is this the only core which spans a long enough time interval? Why are you showing the time series for the deep Pacific only (and not e.g. deep Atlantic, especially after the L305-306 mention)?

Yes, to our knowledge, Qin et al (2018) is the only record of that length. We added this to the text. We added a time series of changes in the deep Atlantic to the SI as suggested.

L319-320 : Having mentions to different periods ('during interglacials', 'during glacial phases') in the same sentence is a bit confusing.

We rephrased the sentence.

L375 : It feels like a verb is missing. Do you mean 'required to compensate the prescribed solute fluxes'?

Yes, we corrected the sentence as suggested.

L382 : 'long-term trend of lower atmospheric d13C during the Eemian than the Holocene'. Why use this convoluted phrasing and not simply 'long-term trend of increasing atmospheric d13C during the last glacial cycle'?

We rephrased the sentence as suggested.

Fig. 8 : Since fLAND causes an increase and fPO4 a decrease, I would be curious to see the absolute effect of the BGC simulation, cumulating the two.

We added the results for BGC to the figure.

L563 : 'the buffering impact of this perturbation on the deglacial carbon re-organization'. This phrasingdoesn't read easily to me, perhaps because it is unclear what 'this perturbation' refers to.

We clarified the sentence.

Technical comments

L7 : 'of' → 'using' or 'with'
L68 : 'extents' → 'amplitude'
L95-96 : misplaced (
L145 : 'our results section' → using section numbering is better
L149 : Does that mean Section 5 of SI?
Fig 1 legend : typo 'gasses'
L172 and 175 : please use insecable spaces so that the units appear on the same line.
Page 8 : wall of text. I recommend a line jump between the description of physical and biogeochemical forcings. I also think that replacing 'Next, we tested...' with some type of numbering would make it easier to follow (e.g. 'A third simulation tested…', 'Thirdly, ...')
L191 : missing comma
L228 : 'sediments' → 'sediment fluxes'?
Fig. S9 : typo 'interglacial'
L257 : incomplete sentence without verb or majuscule.

L267 : 'in Fig. 2'. In this instance (and the whole paragraph), it would be quicker for the reader to find where to look at on the graph if the top panels where numbered and you could refer to 'Fig. 2a,b,c,d'.

Fig. 3 legend : References look misplaced to me. They would be better placed after 'reconstructed', with a mention of which is for POC, and which is for CaCO3.

Fig. S12 and Table S2 : It is unclear what simulation CACO is.

L298 and L302 : extra comma

L304 : unfinished sentence ('but').

L316 : 'causes' → 'cause'

L318 and L443 : typo 'biogeochemical'

Fig. 6 : Since there are mentions in the text of the range of effects in GtC, it could be helpful if a second axis in GtC (on top of the one in ppm) is added. Also, the legend in c) could add the references to the black lines.

L333 : 'Fig. 6' → 'Fig. 6c'

L385 : 'size' → 'amplitude'

L344 : typo majuscule

L392 : typo extra space

L386 : 'geologic' → 'geological'?

L458 : typo 'deglaciations'

L514 : missing space

L544 : 'unlikely' → 'which is unlikely'

L553 : missing comma

We made all the requested changes, except for putting the references for the forcings in Fig. 6 into the figure caption rather than the figure legend due to limited space.